# Integrated transcriptome landscape of ALS identifies genome instability linked to TDP-43 pathology

Oliver J. Ziff [1,2,3] ✉, Jacob Neeves[1,2], Jamie Mitchell[1,2], Giulia Tyzack [1,2], Carlos Martinez-Ruiz [4], Raphaelle Luisier[5], Anob M. Chakrabarti [1], Nicholas McGranahan [4], Kevin Litchfield[4], Simon J. Boulton [1], Ammar Al-Chalabi [6], Gavin Kelly [1], Jack Humphrey [7] & Rickie Patani [1,2,3] ✉

Amyotrophic Lateral Sclerosis (ALS) causes motor neuron degeneration, with 97% of cases exhibiting TDP-43 proteinopathy. Elucidating pathomechanisms has been hampered by disease heterogeneity and difficulties accessing motor neurons. Human induced pluripotent stem cell-derived motor neurons (iPSMNs) offer a solution; however, studies have typically been limited to underpowered cohorts. Here, we present a comprehensive compendium of 429 iPSMNs from 15 datasets, and 271 post-mortem spinal cord samples. Using reproducible bioinformatic workflows, we identify robust upregulation of p53 signalling in ALS in both iPSMNs and post-mortem spinal cord. p53 activation is greatest with *C9orf72* repeat expansions but is weakest with SOD1 and FUS mutations. TDP-43 depletion potentiates p53 activation in both post-mortem neuronal nuclei and cell culture, thereby functionally linking p53 activation with TDP-43 depletion. ALS iPSMNs and post-mortem tissue display enrichment of splicing alterations, somatic mutations, and gene fusions, possibly contributing to the DNA damage response.

ALS is a fatal neurodegenerative disease caused by death of motor neurons[1]. However, there is substantial heterogeneity in both clinical presentation and prognosis[2]. For instance, patients with limb-onset ALS tend to progress more slowly than those with bulbar-onset ALS who typically succumb within two years of diagnosis[1]. The pathological hallmark of ALS is TDP-43 proteinopathy, which is observed in 97% of cases and is characterised by the mislocalisation and aggregation of TDP-43 in the cytoplasm of neurons[3]. More than 20 gene mutations have been established to cause ALS, the most common being in *C9orf72, SOD1, TARDBP* and *FUS*. However, in ~80% of cases, no pathogenic mutation is identified[4,5]. Nonetheless, there is still genetic

susceptibility in these cases with heritability estimates of ~50%, and genome-wide association studies have identified over 15 variants associated with ALS susceptibility (for example *UNC13A, TBK1, ATXN2, NEK1*)[6–8]. This clinical and genetic heterogeneity has made it challenging to identify the pathogenic mechanisms across the spectrum of ALS[7].

A major hurdle in identifying the causes of ALS is the inaccessibility of patient motor neurons. Although post-mortem ALS tissue has revealed important insights, it represents the end-stage of the disease, with few surviving motor neurons[9–12]. Human induced pluripotent stem cell (iPSC)-derived motor neurons (iPSMNs) offer a potential

[1]The Francis Crick Institute, 1 Midland Road, London NW1 1AT, UK. [2]Department of Neuromuscular Diseases, Queen Square Institute of Neurology, University College London, London WC1N 3BG, UK. [3]National Hospital for Neurology and Neurosurgery, University College London NHS Foundation Trust, London WC1N 3BG, UK. [4]Cancer Research UK Lung Cancer Centre of Excellence, University College London Cancer Institute, London, UK. [5]Genomics and Health Informatics Group, Idiap Research Institute, Martigny, Switzerland. [6]Maurice Wohl Clinical Neuroscience Institute, Department of Basic and Clinical Neuroscience, Institute of Psychiatry, Psychology and Neuroscience, King's College London, London, UK. [7]Nash Family Department of Neuroscience & Friedman Brain Institute, Icahn School of Medicine at Mount Sinai, New York, NY, USA. ✉e-mail: o.ziff@ucl.ac.uk; rickie.patani@ucl.ac.uk

solution. iPSMNs recapitulate pathological features of ALS, enabling exploration of the functional consequences of genetic variants on motor neurons during the initial phases of the disease[13–15]. Since iPSMNs can be generated from any individual irrespective of their genetic background, they enable sporadic ALS to be modelled, which is not possible with animal models[16]. However, iPSMN cultures are expensive and labour-intensive, and many studies have been limited to three or fewer patients[17–21]. Despite this, there has been a recent expansion of ALS iPSMN biobanks with initiatives such as neuroLINCS[22] and Answer ALS[23], offering a unique opportunity to identify generalisable motor neuron perturbations across ALS genetic backgrounds.

Here, we report a robust analytical framework to identify unifying transcriptomic aberrations underlying motor neuron dysfunction in ALS, providing the largest resource of ALS iPSMNs and post-mortem tissue to date. We identify an accumulation of somatic mutations and a heightened DNA damage response in ALS, most strikingly in cases with TDP-43 proteinopathy. This demonstrates the importance of transcriptomic data in understanding how diverse genetic backgrounds contribute to ALS and in mapping the landscape of ALS-related RNA changes.

## Results

### iPSC-derived motor neuron resource

Our database search strategy identified 16 ALS iPSMN bulk RNA-sequencing (RNA-seq) datasets, of which 15 passed quality control (Fig. 1, Supplementary Fig. 1). iPSMN differentiation protocols for each dataset were extracted and found to follow generally similar procedures between datasets; however, there were notable differences in the duration of cultures, which ranged between 12–42 days in vitro (mean 31 days; Supplementary Data 1). All samples underwent extensive quality control, and principal component analysis was used to investigate the effects of sequencing and culture batch confounding variables (Supplementary Data 2). This revealed two global clusters of iPSMNs separated by PC1 and PC2 according to poly(A) or total Ribo-Zero RNA library preparation (Supplementary Fig. 2-3). Principal component gene loadings confirmed that the separation was driven by histone and small nucleolar encoding genes, which represent non-polyadenylated genes (Supplementary Fig. 2a).

iPSMNs showed high expression of neuronal markers across datasets, except for one HB9-reporter dataset that also exhibited different RNA library preparations between ALS (Ribo-Zero) and control (polyA) samples and was therefore excluded (Supplementary Fig. 4)[24]. Although the expression of post-mitotic dorso-ventral motor neuron domain markers varied between datasets (e.g. *CHAT, MNX1 [HB9], LHX3, FOXP1, ALDH1A2, ISL1*), there was a strong expression of rostro-caudal markers (*HOX1-8*) across datasets, which is consistent with hindbrain, cervical, and thoracic spinal cord specification (Supplementary Fig. 5–7).

For the integrated analysis, we included 15 datasets comprising 429 iPSMNs, of which 323 were from ALS patients and 106 were from non-ALS controls. ALS iPSMNs carried pathogenic mutations in 10 different genes, including *C9orf72*[22,23,25–28] ($n = 60$), *SOD1*[18,19,21–23] ($n = 20$), *FUS*[20,23,25,29,30] ($n = 14$), and *TARDBP*[23,26,31,32] ($n = 10$); whilst 208 (64.2%) were from patients without an identifiable ALS mutation, which we refer to here as sporadic ALS (Table 1).

### ALS iPSMNs activate the DNA damage response

To investigate the underlying mechanisms of motor neuron degeneration across all genetic subtypes of ALS, we first focused on identifying pan-ALS gene expression changes. We performed an integrated analysis comparing all 323 ALS versus 106 control iPSMNs, accounting

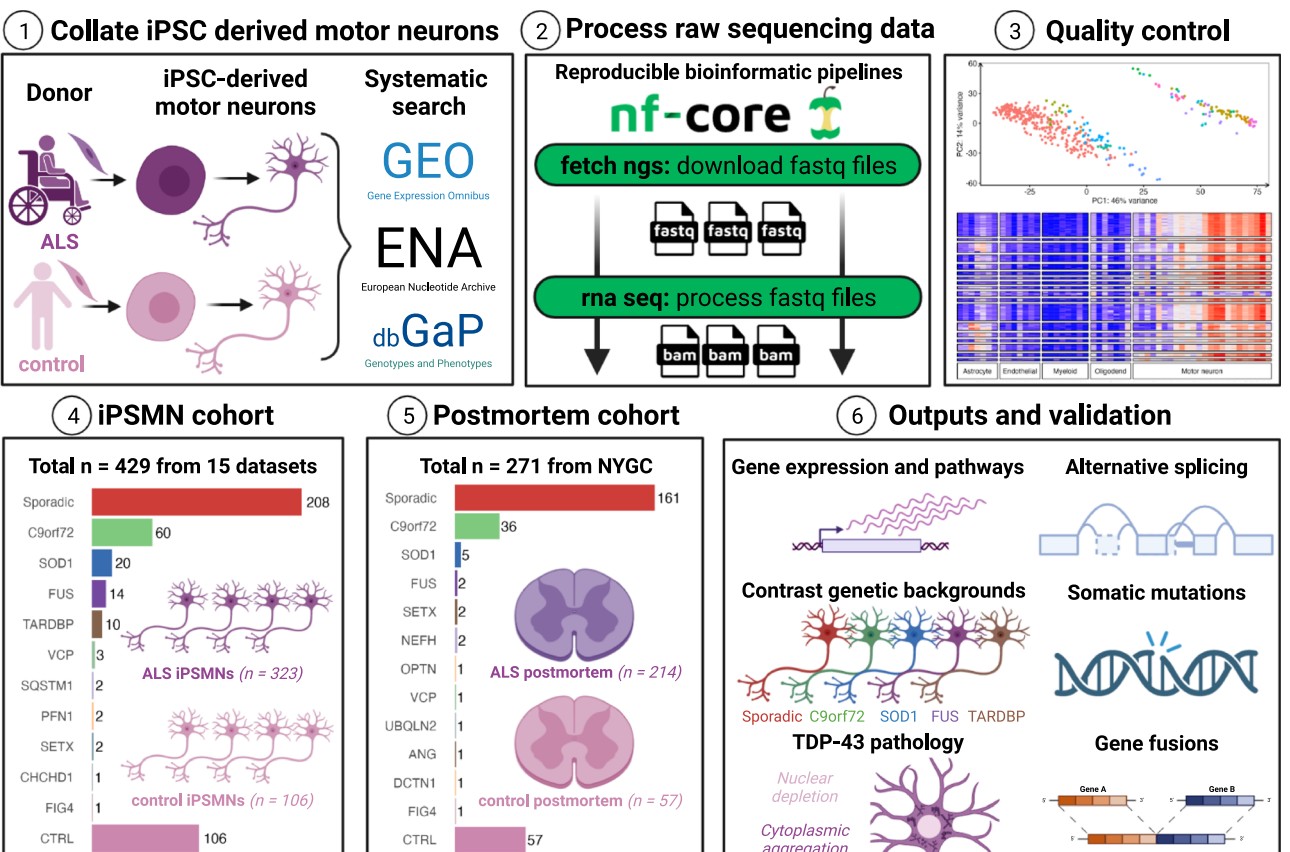

**Fig. 1 | Study overview.** Schematic summarising our analytic framework using iPSC-derived motor neurons (iPSMNs) and post-mortem tissue to interrogate perturbations across the spectrum of ALS. Made with BioRender.

**Table 1 iPSMN datasets included in the integrated analysis**

| Reference | Accession # | Mutation | ALS n | Control n | Library type | Layout |
|---|---|---|---|---|---|---|
| Sareen et al. 2013 | GSE52202 | C9orf72 | 4 | 4 | polyA | Single |
| Kiskinis et al. 2014 | GSE54409 | SOD1 | 2 | 3 | polyA | Paired |
| Kapeli et al. 2016 | GSE77702 | FUS | 3 | 2 | polyA | Single |
| Wang et al. 2017 | GSE95089 | SOD1 | 2 | 2 | polyA | Paired |
| De Santis et al. 2017 | GSE94888 | FUS | 3 | 3 | Ribo-zero | Paired |
| Bhinge et al. 2017 | PRJNA361408 | SOD1 | 2 | 2 | Ribo-zero | Single |
| Luisier et al. 2018 | GSE98290 | VCP | 3 | 3 | polyA | Single |
| Abo-Rady et al. 2020 | GSE143743 | C9orf72 | 3 | 3 | polyA | Single |
| Dafinca et al. 2020 | GSE139144 | C9orf72 | 4 | 8 | polyA | Paired |
| | GSE147544 | TARDBP | 6 | 4 | | |
| Catanese et al. 2021 | GSE168831 | C9orf72 | 6 | 6 | polyA | Paired |
| | | FUS | 6 | | | |
| Smith et al. 2021 | PRJEB47567 | TARDBP | 3 | 2 | polyA | Paired |
| Hawkins et al. 2022 | GSE203168 | FUS | 2 | 2 | Ribo-zero | Single |
| Sommer et al. 2022 | GSE201407 | C9orf72 | 6 | 6 | polyA | Paired |
| NeuroLINCS, 2022 | phs001231.v2.p1 | sporadic | 8 | 14 | Ribo-zero | Paired |
| | | SOD1 | 6 | | | |
| | | C9orf72 | 16 | | | |
| Answer ALS, 2022 | AnswerALS data portal | sporadic | 200 | 42 | Ribo-zero | Paired |
| | | C9orf72 | 21 | | | |
| | | SOD1 | 8 | | | |
| | | 6 other ALS mutations | 9 | | | |
| Total | | | 323 | 106 | | |

for batch effects between datasets and sex (see Methods). We found 43 differentially expressed genes in pan-ALS versus control iPSMNs, with 20 upregulated and 23 downregulated in ALS iPSMNs (false discovery rate [FDR] < 0.05, Fig. 2a, Supplementary Data 3). Amongst differentially expressed genes most increased in ALS was the endoribonuclease *RNase L (RNASEL)* which regulates the decay of cytoplasmic RNA and localisation of RNA binding proteins (RBPs)[33].

Functional over-representation analysis revealed that upregulated genes in ALS were enriched in the DNA damage response (hypergeometric FDR = $2.2 \times 10^{-5}$; *SESN1, RRM2B, TNFRSF10B*) and p53 signalling (FDR = $2.7 \times 10^{-5}$; *CDKN1A [p21], TP53TG3E, FBXO22*) whereas downregulated genes were overrepresented by DNA-binding transcription factor activity (FDR = 0.003; *MYOG, TBX5, POU5F1*) and ventral spinal cord development (FDR = 0.004; *LMO4, OLIG2, FOXN4*; Fig. 2b). Gene Set Enrichment Analysis (GSEA) identified significant up-regulation of the p53 signal transduction gene set (GO:0072331, *n* = 264) in ALS iPSMNs (normalised enrichment score [NES]+ 1.44, enrichment *p* = $4.9 \times 10^{-4}$; Fig. 2c). Since p53 signalling and DNA damage response are large pathways, we next explored more specific gene sets by examining their daughter pathways. This revealed strong upregulation of genes involved with the mitotic G1 DNA damage checkpoint and intrinsic apoptosis signalling (Supplementary Fig. 8).

To further understand how signalling pathways are activated in ALS iPSMNs, we performed a Signalling Pathway RespOnsive GENes (PROGENy)[34] analysis which leverages perturbation experiments to infer pathway activity changes, weighting genes based on their responsiveness. PROGENy revealed that the most substantial pathway activity increase in ALS iPSMNs was in p53 (NES + 13.0, *p* < 0.001), followed by Mitogen-Activated Protein Kinase (MAPK; NES + 5.6, *p* < 0.001), whilst the greatest decrease was observed in WNT (NES −2.5, *p* = 0.03; Fig. 2d). Examining each gene in the p53 pathway according to its p53 weighting in PROGENy revealed that the genes with the strongest responsiveness in p53 activity in ALS iPSMNs included *CDKN1A, SESN1, RRM2B, MDM2, C2orf66, ZNF561* and *ZMAT3* (Fig. 2e).

We next inferred the activities of 429 transcription factors (TFs) from their regulon expression within the DoRothEA database[34]. Remarkably, this revealed that TP53 was the TF with the greatest increase in activity in ALS (NES + 7.62, *p* < 0.001) followed by ZNF274 and ATF4. The strongest TF decreases in ALS were in PRDM14, ZNF263, and SIX5 (Fig. 2f; Supplementary Data 4). Interrogating individual genes constituting the TP53 TF regulon revealed the greatest increases in ALS iPSMNs in *TNFRSF10B, SESN1, RRM2B, CDKN1A, ZMAT3* and *MDM2*. These integrated analysis results can be easily explored in the interactive web application at https://oliverziff.shinyapps.io/als_genome_instability/.

Although our statistical design adjusts for dataset batch effects, it is plausible that changes between ALS and control groups were confounded by imbalances between total Ribo-Zero and poly(A) libraries. To address this, we performed a subgroup analysis in poly(A) datasets (10 datasets; 48 ALS, 43 control iPSMNs) and total Ribo-Zero (5 datasets; 275 ALS, 63 control iPSMNs) separately (Supplementary Fig. 9a, b). Comparing ALS versus control iPSMNs revealed that in poly(A) datasets there were 69, and in total Ribo-Zero datasets there were 12 differentially expressed genes (Supplementary Fig. 9c, d). Overlapping differentially expressed genes between analyses revealed that RNase L was significantly increased in ALS iPSMNs in both library preparation analyses independently. Furthermore, we confirmed significant increases in p53 pathway activity in ALS with both library preparations (polyA datasets: NES + 11.4, *p* < 0.001; Ribo-Zero datasets: NES + 7.3, *p* < 0.001; Supplementary Fig. 9e, f). Likewise, TP53 TF activity was significantly increased in ALS iPSMNs in both library preparation groups (polyA datasets: NES + 6.54, *p* < 0.001; Ribo-Zero datasets: NES + 5.12, *p* < 0.001; Supplementary Fig. 9g, h). We further investigated changes between ALS and control samples in each dataset separately. This revealed substantial heterogeneity between datasets in ALS versus control gene expression changes (Supplementary Fig. 10a, b). Despite this, both TP53 transcription factor and p53 signalling were independently upregulated in ALS iPSMNs in 11 of

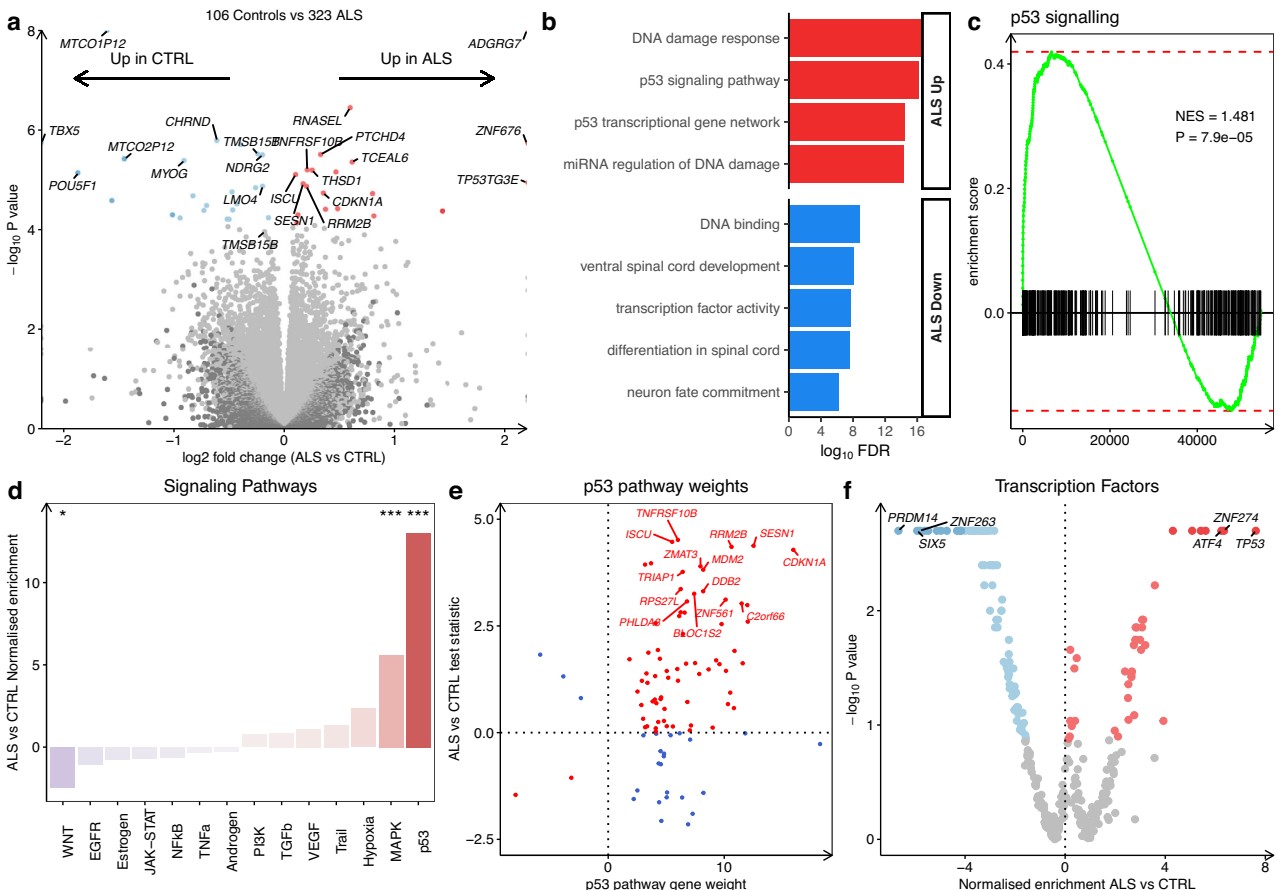

**Fig. 2 | Differential gene expression in ALS versus control iPSMNs. a** Volcano plot of differential gene expression in ALS versus control iPSMNs using the Wald test. **b** Functionally overrepresented terms in up-regulated (red) and down-regulated (blue) differentially expressed genes using the hypergeometric test. **c** GSEA of signal transduction by p53 (GO:0072331, $n = 264$) in ALS versus control using the permutation test. NES, normalized enrichment score. **d** PROGENy signalling pathway activities in ALS versus control using the weighted mean method. Pathways increased in ALS are red and pathways decreased are blue. *** represents

$P < 0.0001$ and $^{*}P < 0.05$ (p53 $p < 0.001$, MAPK $p < 0.001$, WNT $p = 0.03$). **e** Expression changes of p53 signalling pathway genes in ALS versus control according to their PROGENy weights. Genes increasing p53 activity in ALS are red whilst genes decreasing p53 activity in ALS are blue. **f** Activities of 429 transcription factors in DoRothEA inferred from their regulon expression changes in ALS versus control. The normalised enrichment score in ALS versus control (x-axis) is plotted according to the enrichment test p-value (y-axis).

17 datasets (Supplementary Fig. 10c, d). This indicates that neither library preparation nor dataset batch effects were responsible for the DNA damage response gene expression changes observed in ALS iPSMNs.

To identify how gene expression changes relate to protein expression changes in ALS iPSMNs, we interrogated mass spectrometry data in Answer ALS, which includes 204 iPSMNs (ALS $n = 171$, controls $n = 33$). Across the whole proteome, no proteins were significant at FDR < 0.05 in ALS compared to control, however, 276 were significantly different at unadjusted $p < 0.05$ (Supplementary Fig. 11a; Supplementary Data 5). Amongst these were the p53 pathway components RBBP7, CSNK2B, PRMT1, CNOT9, which were each increased in ALS iPSMNs. Functional over-representation analysis revealed that proteins increased in ALS were enriched in protein metabolism (e.g. PSDMD9, NAE1, PSMB5), RNA metabolism (e.g. APP, CSTF1, CIRBP, SNRPD2), protein binding (e.g. RRBP1, RPS29, EIF1), as well as other processes established in ALS pathophysiology (stress response, cholesterol synthesis, nucleocytoplasmic transport) whilst proteins decreased in ALS, were enriched in Golgi transport (Supplementary Fig. 11b). Enrichment analysis of the p53 pathway protein set revealed a nonsignificant increase in ALS iPSMNs (NES + 0.98, $p = 0.5$; Supplementary Fig. 11c). Comparing changes in mRNA expression with protein expression revealed a weak inverse correlation (R = −0.13;

Supplementary Fig. 11d), consistent with previous reports showing poor correlations between mRNA and protein[35,36].

## p53 activation across ALS genetic backgrounds

To identify how ALS iPSMN changes compare between ALS genetic backgrounds, we next examined the effect of each genetic subgroup on gene expression separately. Of the 15 datasets, 7 included C9orf72 mutants (comprising 60 C9orf72 iPSMNs and 83 control iPSMNs), 5 included SOD1 mutants (20 SOD1, 63 controls), 5 included FUS mutants (14 FUS, 55 controls), 3 included TARDBP mutants (10 TARDBP, 48 controls) and 2 included sporadic iPSMNs (208 sporadic, 56 controls). Controls from each dataset were utilised only if the dataset had samples from the relevant genetic background.

Although we found 3,547 differentially expressed genes (adjusted $P < 0.05$) in *TARDBP* mutants, the other subgroups showed more modest changes: *FUS* (239), *C9orf72* (161), and *SOD1* (7). Despite sporadic ALS being the most well-powered, with 208 iPSMNs, only 4 genes were differentially expressed (Fig. 3a–e). Correlating transcriptome-wide gene expression changes between ALS genetic backgrounds revealed weak associations, with the strongest correlation between *SOD1* and sporadic lines (Pearson R = +0.38, $p < 2.2 \times 10^{-16}$) and the weakest between *SOD1* and *TARDBP* (R = −0.14, $p < 2.2 \times 10^{-16}$; Fig. 3f, Supplementary Fig. 12a). To identify whether

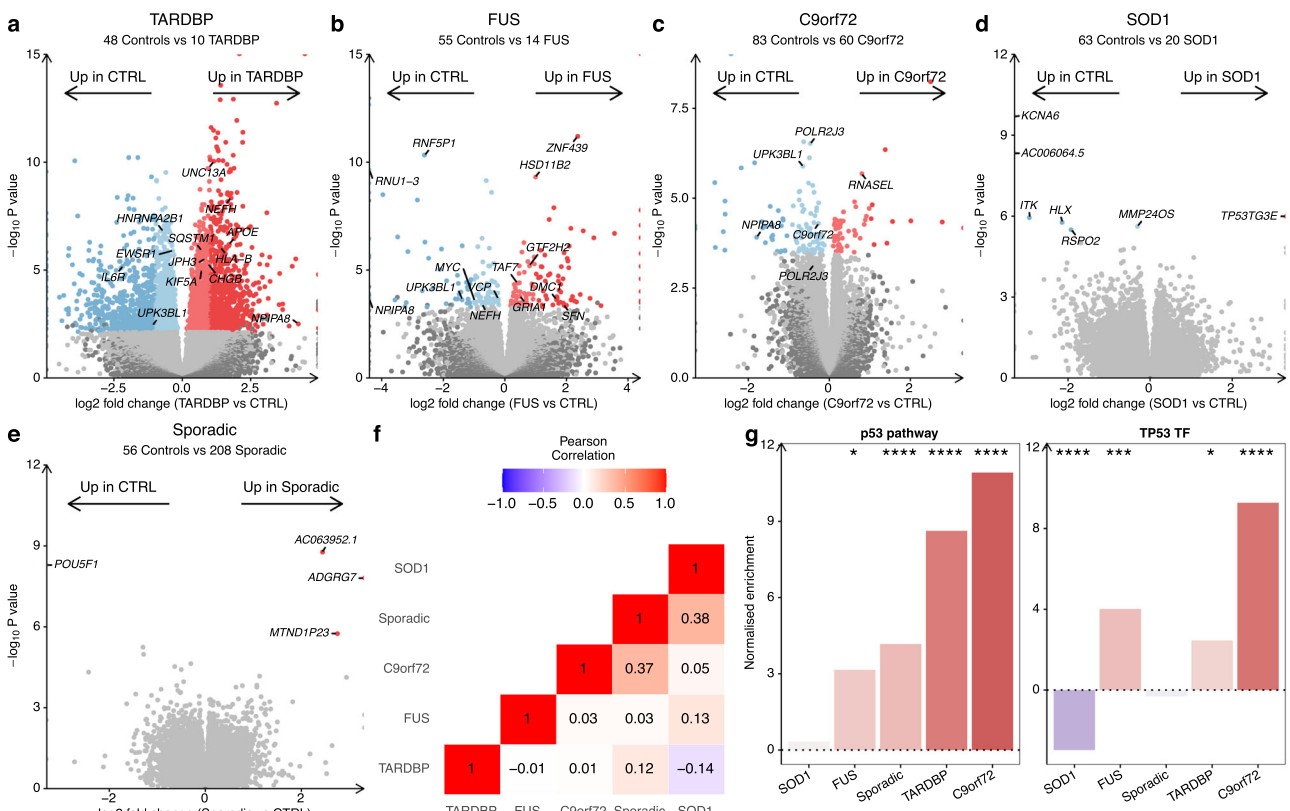

**Fig. 3 | Gene expression changes in each ALS genetic background. a–e** Volcano plots comparing ALS iPSMNs to controls in each ALS genetic background. Genes coloured red are significantly increased in the ALS subgroup and genes coloured blue are decreased in the ALS subgroup using the Wald test. **f** Heatmap showing the Pearson's correlation coefficient for transcriptome-wide changes between each genetic background. **g** PROGENy p53 signalling pathway (left) and Dorothea TP53 transcription factor regulon (right) activities amongst each of the genetic backgrounds independently using the weighted mean method. **** represents $P < 0.0001$, ***$P < 0.001$, **$P < 0.01$, *$P < 0.05$.

different ALS genetic backgrounds exhibit differential expression within the same genes, we overlapped genes significantly changed in expression (Supplementary Data 6). Although no genes were significantly changed in expression across all ALS genetic backgrounds, Uroplakin *UPK3BL1* and nuclear pore complex interacting protein *NPIPA8* were changed in the *C9orf72, TARDBP* and *FUS* subgroups (Supplementary Fig. 12b).

Functional over-representation analysis of differentially expressed genes in each genetic subgroup revealed that *C9orf72* mutants upregulated genes involved with p53 (hypergeometric FDR = 0.003) and the DNA damage response (FDR = 0.02) whilst downregulating cytoskeleton (FDR = 0.01) and microtubule genes (FDR = 0.02). *FUS* mutants showed upregulation of genes involved with transcription (FDR = $1.5 \times 10^{-8}$) and DNA-binding (FDR = $2.4 \times 10^{-8}$) and down-regulation of synaptic signalling genes (FDR = 0.02). Conversely, *TARDBP* mutants upregulated neuronal (FDR = $7.8 \times 10^{-29}$) and synaptic genes (FDR = $1.2 \times 10^{-32}$) and downregulated genes involved in the cell cycle (FDR = $3.1 \times 10^{30}$) and RNA splicing (FDR = $2 \times 10^{-5}$, Supplementary Fig. 12c-e). There were no functional terms enriched amongst SOD1 or sporadic ALS differentially expressed genes.

Examining PROGENy pathway activities in each genetic subgroup revealed that apart from *SOD1* (NES + 0.33, $p = 0.16$), the p53 pathway activity was significantly increased in each of *C9orf72* (NES + 10.9, $p < 0.001$), *TARDBP* (NES + 8.6, $p < 0.001$), sporadic (NES + 4.2, $p < 0.001$) and *FUS* (NES + 3.2, $p = 0.018$; Fig. 3g). Examining the other signalling pathways revealed that hypoxia, VEGF and MAPK were also increased across most genetic backgrounds, whilst WNT and PI3K tended to be decreased (Supplementary Fig. 13a). Examining TF regulon activity revealed significantly increased TP53 activity in *C9orf72*

(NES + 9.3, $p < 0.001$), *FUS* (NES + 4.0, $p = 0.004$) and *TARDBP* (NES + 2.5, $p = 0.01$) but decreased activity in *SOD1* (NES −3.0, $p < 0.001$) whilst sporadic (NES −0.31, $p = 0.25$) was non-significantly changed (Fig. 3g). Observing the other TF activity changes between genetic backgrounds revealed 5 TFs that were significantly changed in the same direction across 4 of 5 genetic backgrounds, of which ZNF274 was increased whilst GATA3, MAZ, TAL1 and TEAD4 were decreased in the ALS subgroups (Supplementary Fig. 13b). Taken together, despite transcriptome-wide heterogeneity between genetic backgrounds, these data suggest that p53 signalling activation is observed across the ALS spectrum in iPSMNs.

## ALS post-mortem tissue shows p53 activation
To identify whether iPSMN ALS gene expression signatures are also found in post-mortem tissue, we compared our findings with post-mortem spinal cord RNA-seq from the NYGC ALS cohort, consisting of tissue from 214 ALS patients and 57 controls[12,37]. We found 14,064 differentially expressed genes in post-mortem ALS versus control spinal cord samples, with 6575 upregulated and 7489 downregulated in ALS (FDR < 0.05; Fig. 4a). *CHIT1, GPNMB* and *LYZ* were the most strongly upregulated genes in ALS spinal cord, consistent with a recent report[12]. Functional over-representation analysis revealed that upregulated genes were enriched in mitochondria (FDR = $5.0 \times 10^{-74}$), stress response (FDR = $3.9 \times 10^{-42}$), programmed cell death (FDR = $1.8 \times 10^{27}$), and the p53 DNA damage response (FDR = $1.0 \times 10^{-5}$), whilst down-regulated genes were enriched in neuronal functions (Fig. 4b). GSEA of the p53 signal transduction gene set in ALS post-mortem spinal cord confirmed significant up-regulation (NES + 1.58, $p = 9.0 \times 10^{-6}$; Fig. 4c). As with iPSMNs, in ALS post-mortem spinal cord, the most strongly

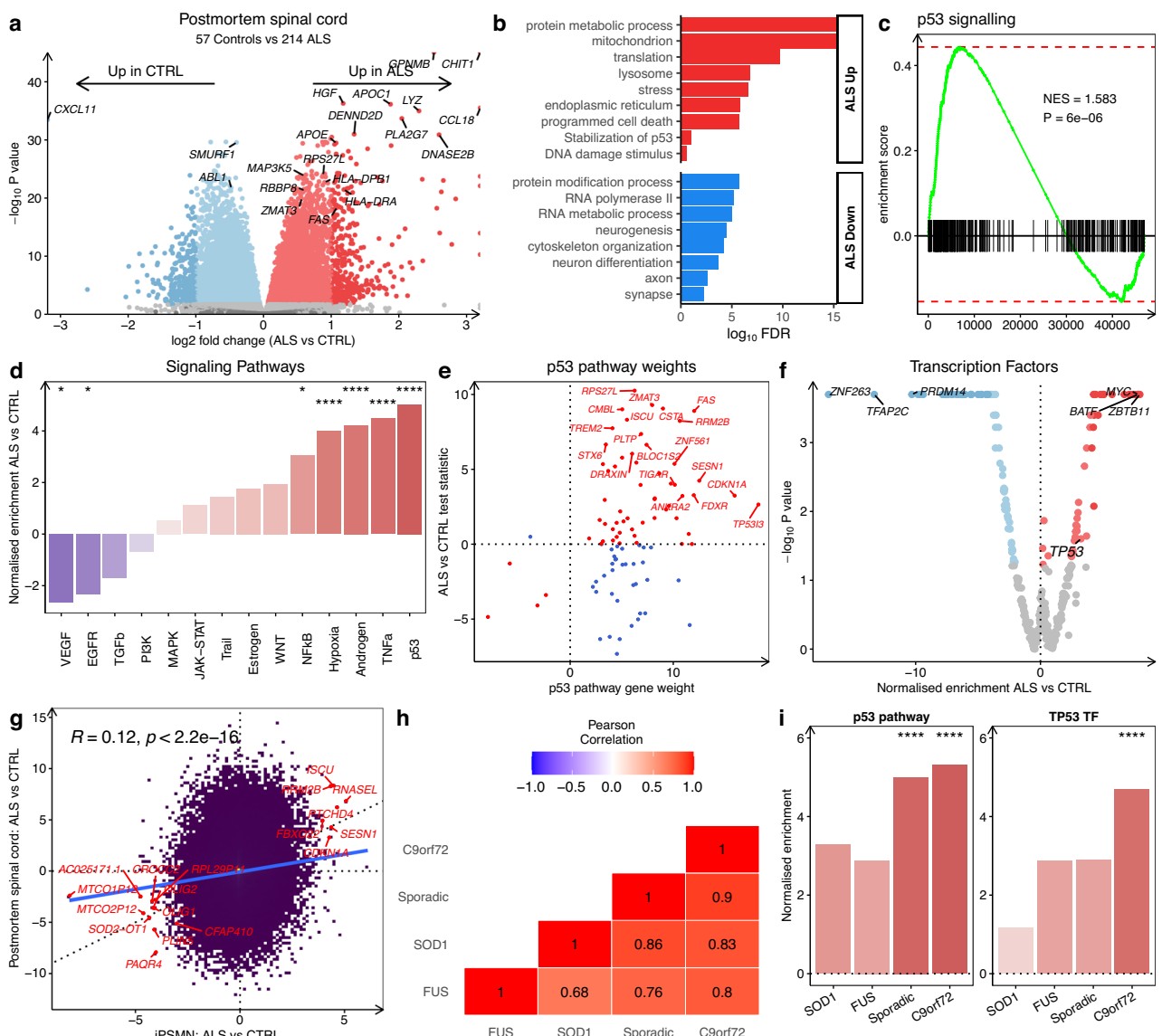

**Fig. 4 | Post-mortem spinal cord shows p53 activation. a** Volcano plot of differential gene expression in ALS versus control post-mortem spinal cord using the Wald test. **b** Functionally enriched terms in up-regulated (red) and down-regulated (blue) differentially expressed genes using the hypergeometric test. **c** GSEA for signal transduction by p53 in ALS versus control post-mortem spinal cord using the permutation test. NES, normalized enrichment score. **d** PROGENy signalling pathway activities in ALS versus control post-mortem tissue using the weighted mean method. Pathways increased in ALS are red and pathways decreased are blue. **e** Expression changes of p53 signalling pathway genes in ALS versus control according to their PROGENy weights. Genes in ALS increasing p53 activity are red

and genes decreasing p53 activity are blue. **f** Activities of 429 transcription factors in DoRothEA inferred from their regulon expression changes in ALS versus control post-mortem tissue using the enrichment test. **g** Scatterplot of ALS vs control gene expression changes in iPSMNs (x-axis) against post-mortem tissue (y-axis) using the Wald test statistic. **h** Heatmap showing the Pearson's correlation coefficient for transcriptome-wide changes between each genetic background in post-mortem tissue. **i** PROGENy p53 signalling pathway (left) and DoRothEA TP53 transcription factor regulon activity (right) amongst each of the genetic backgrounds in post-mortem tissue using the weighted mean method. *** represents $P < 0.0001$ and *$P < 0.05$.

upregulated PROGENy signalling pathway was p53 (NES + 5.0, $p < 0.001$; Fig. 4d). The top individual genes driving p53 pathway activity in ALS post-mortem included *FAS, RRM2B, CSTA, ZMAT3, TP53I3,* and *CDKN1A* (p21; Fig. 4e). Other signalling pathway activities that were also significantly increased in ALS post-mortem spinal cord were TNFα, androgen, hypoxia, and NFκB whereas EGFR and VEGF pathways were significantly decreased. Assessing the 429 TF activities revealed that TP53 was amongst the top TFs increased in ALS post-mortem tissue (NES + 3.2, $p = 0.03$; Fig. 4f).

Correlating transcriptome-wide ALS gene expression changes between iPSMNs and post-mortem spinal cord revealed a weak positive correlation (Pearson R = + 0.12, $p < 2.2 \times 10^{-16}$; Fig. 4g). Of the 43 differentially expressed genes changed in ALS iPSMNs, 17 (39.5%) were also

changed in ALS post-mortem spinal cord, with 7 co-upregulated and 10 co-downregulated. These included the DNA damage response and p53 pathway genes *CDKN1A, SESN1, FBXO22,* and *RRM2B* as well as the endoribonuclease *RNase L (RNASEL)*, lipid droplet coating *Perlipin5*, and oxidative stress responder *ISCU*. Meanwhile, overlapping down-regulated genes included the mitochondrial genes MTCO1P12 and MTCO2P12 and motor neuron progenitor markers, OLIG1 and OLIG2 (Supplementary Data 7). These results not only confirm p53-dependent DNA damage response upregulation in ALS but the overlap between iPSMNs and post-mortem provides insight into motor neuron-specific changes that begin early and persist into the later stages of the disease.

In contrast to iPSMNs, comparing genetic subgroups from post-mortem spinal cord tissue revealed strongly correlated gene

expression changes (R range +0.68 to +0.9) with 1750 overlapping differentially expressed genes between sporadic, C9orf72, SOD1, and FUS subgroups (Fig. 4h, Supplementary Fig. 14a–f). In each subgroup, upregulated differentially expressed genes were consistently over-represented by the stress response, cell death, and protein metabolism, while downregulated genes were enriched in protein binding and neuronal terms (Supplementary Fig. 14g–j). Examining signalling pathways and transcription factors in each genetic subgroup revealed that p53 signalling and TP53 TF activity were significantly increased in both in sporadic ($n = 161$; p53: NES + 5.0, $p < 0.001$; TP53 NES + 2.9, $p = 0.05$) and *C9orf72* ($n = 36$; NES + 5.3, $p < 0.001$; TP53 NES + 4.7, $p < 0.001$) and non-significantly increased in *SOD1* ($n = 5$; NES + 3.3, $p = 0.05$; TP53 NES + 1.2, $p = 0.1$) and *FUS* ($n = 2$; NES + 2.9, $p = 0.28$; TP53 NES + 2.9, $p = 0.29$; Fig. 4i, Supplementary Fig. 14k, l).

### TDP-43 pathology contributes to the DNA damage response

Although TDP-43 pathology (characterised by neuronal TDP-43 nuclear depletion and cytoplasmic accumulation) is observed in 97% of ALS, it is absent in SOD1 and FUS mutant cases (termed "non-TDP-43 ALS" hereafter)[38,39]. To identify the degree to which genotypes linked to TDP-43 pathology contribute to p53 upregulation in the pan-ALS analyses, we classified ALS samples based on whether their genetic background is associated with TDP-43 pathology. In iPSMNs, whilst non-TDP-43 ALS (SOD1 and FUS mutant) iPSMNs showed only a modest, non-significant increase in p53 (NES = + 2.0, $p = 0.25$), TDP-43 ALS iPSMNs exhibited strong and significant p53 upregulation (p53 NES = + 14.2, $p < 0.001$). Likewise, in non-TDP-43 ALS iPSMNs, the TP53 TF was mildly decreased in activity (NES = −1.6, $p = 0.12$), whereas in TDP-43 ALS iPSMNs TP53 was the most strongly upregulated TF (NES + 7.4, $p < 0.001$; Supplementary Fig. 15a, b). We found a similar pattern in post-mortem samples, with non-TDP-43 ALS showing smaller less-significant increases in p53 signaling and TP53 TF activity (p53: NES + 3.6, $p = 0.03$; TP53: NES + 3.3, $p = 0.01$) as compared to TDP-43 ALS (p53: NES + 5.2, $p < 0.001$; TP53: NES + 2.0, $p = 0.5$; Supplementary Fig. 15c, d). These findings suggest that the p53 signature from the pan-ALS analyses is largely driven by genetic backgrounds associated with TDP-43 proteinopathy.

To discover whether p53 signalling changes are regulated by TDP-43, we next examined RNA-seq from FACS-sorted neuronal nuclei into those with and without TDP-43 pathology from FTD-ALS post-mortem brain tissue[40]. We found that neuronal nuclei depleted of TDP-43 showed upregulation of p53 signalling and TP53 TF activity as compared to neuronal nuclei retaining TDP-43 (p53: NES + 0.4, $p = 0.02$; TP53: NES + 0.7, $p = 0.26$; Supplementary Fig. 15e, f).

To determine whether TDP-43 nuclear depletion directly promotes p53 activation, we integrated seven RNA-seq datasets from human cells that have undergone TDP-43 knockdown with shRNA, siRNA or CRISPR/Cas9 (Supplementary Data 8)[31,41–46]. We discovered significant upregulation of both p53 signalling (NES + 5.5, $p = 0.02$) and TP53 TF activity (NES + 3.5, $p = 0.02$) upon TDP-43 knockdown, supporting a direct role of TDP-43 in regulating p53 signalling (Supplementary Fig. 15e, f). To identify whether TDP-43 cytoplasmic aggregation also contributes to p53 activation, we next examined RNA-seq from mouse primary neurons overexpressing TDP-43[47]. Compared to controls, neurons over-expressing TDP-43 showed significant upregulation of the p53 pathway (NES + 4.1, $p = 0.002$) and TP53 transcription factor (NES + 4.1, $p = 0.002$ Supplementary Fig. 15g, h). Together, these results indicate that both depletion and accumulation of TDP-43 augment p53 activity, suggesting that tight regulation of TDP-43 levels is required to ensure an appropriate DNA damage response.

### ALS iPSMNs and post-mortem tissue exhibit extensive alterations in splicing

An increasingly proposed mechanism of disease in ALS is dysregulated alternative splicing, which may also contribute to genomic instability[48].

To identify alternative splicing changes in ALS iPSMNs, we utilised the splice graph tool MAJIQ[49,50], which quantifies local splicing variations from large heterogeneous RNA-seq datasets and corrects for dataset batch effects (Fig. 5a)[51]. Since total Ribo-Zero RNA libraries predominantly capture unprocessed nascent pre-mRNAs, we restricted iPSMN splicing analyses to poly(A) selected libraries, which captures mature mRNAs (10 datasets composed of 48 ALS and 43 control iPSMNs). Comparing ALS versus control iPSMNs identified 264 local splice variation events in 161 unique genes that were significantly different between ALS and control (TNOM $p < 0.05$, Δ PSI > 0.1; Fig. 5b, Supplementary Data 9). Amongst the genes exhibiting differential splicing in ALS were a significant number of genes involved with p53 and DNA repair (including *POLM, METTL22, HUWE1, HDAC1, MTA1, PMS1, ZSWIM7;* Fisher exact test $p = 2.3 \times 10^{-8}$). Likewise, there was a greater number of RBPs amongst differentially spliced genes than expected by chance (e.g. *YTHDC2, THOC1, PRR3, STAU2, PTBP3, SREK1, POLDIP3;* $p = 6.8 \times 10^{-18}$; Fig. 5e). Functional over-representation analysis of the 161 genes containing differential splicing showed enrichment in protein binding (FDR = 0.02), synaptic (FDR = 0.03) and neuronal functions (FDR = 0.04, Fig. 5c), which are central to ALS motor neuron pathophysiology.

By examining splicing changes in neuronal nuclei depleted of TDP-43[40], we found 12 overlapping differentially spliced genes (encompassing 17 splicing events) with ALS iPSMNs (including *POLDIP3, PPP6R3, CAMK2B, CEP290*; Supplementary Data 9)[40]. Similarly, comparing splicing changes upon TDP-43 knockdown with ALS iPSMNs revealed 4 overlapping genes containing 6 splicing events (*POLDIP3, CAMK2B, HERC2P3, CEP290*; Supplementary Data 9). Interestingly, the multi-exon skipping splicing event in *POLDIP3* was precisely the same event that occurs in both TDP-43 neuronal nuclei depletion and TDP-43 knockdown (Supplementary Fig. 16a–c)[43,52]. This indicates that TDP-43 nuclear loss of function may contribute to splicing changes in ALS iPSMNs.

The local splice variations identified by MAJIQ were predominantly complex, composed of combinations of various 3′ and 5′ splice sites rather than simple binary events (e.g. exon skipping or intron retention [IR]). Furthermore, splice events are not restricted to annotated reference transcriptome splice sites and of the 264 differential splicing events, 28 (10.6%) involved de novo splice junctions. Of these, 7 were found to be cryptic exons (*RELCH, HOXC4, RBM26, SLC35B3, TENM3, TPTEP2-CSNK1E, ZSCAN29*) although none of these overlapped with TDP-43 depletion[43]. 50 out of 264 (18.9%) differential splice events harboured IR within the local splice variation. Breaking down each local splice variation into its component splice types and categorising these into basic splicing modules revealed that exon skipping was the most common splicing type (182, 45.4%) followed by IR (54, 13.5%; Fig. 5d).

We next investigated alternative splicing in each ALS genetic subgroup separately. Whilst there were no sporadic ALS iPSMNs that had undergone poly(A) selection, there were 23 C9orf72, 9 FUS, 9 TARDBP and 4 SOD1 mutant poly(A) samples. Compared to controls, *TARDBP* mutants showed the greatest number of differential splicing events (1435), followed by *FUS* (1099), *C9orf72* (429) and *SOD1* (256; Supplementary Fig. 17a–d). Functional over-representation analysis in each mutant group revealed that genes exhibiting differential splicing were involved with protein binding, neuronal structures, and RNA processing (Supplementary Fig. 17e–h). Exon skipping was the most common splicing type in *TARDBP, FUS* and *C9orf72* subgroups, however, in *SOD1* mutants, intron retention was the most frequent (Supplementary Fig. 17i). Correlation of alternative splicing changes between genetic subgroups revealed weak associations with the strongest correlation between *FUS* and *C9orf72* mutations (R = + 0.09) and the weakest correlation between *SOD1* and *TARDBP* (R = −0.1; Supplementary Fig. 17j). Intersecting significant splicing changes between genetic backgrounds revealed that none were common to

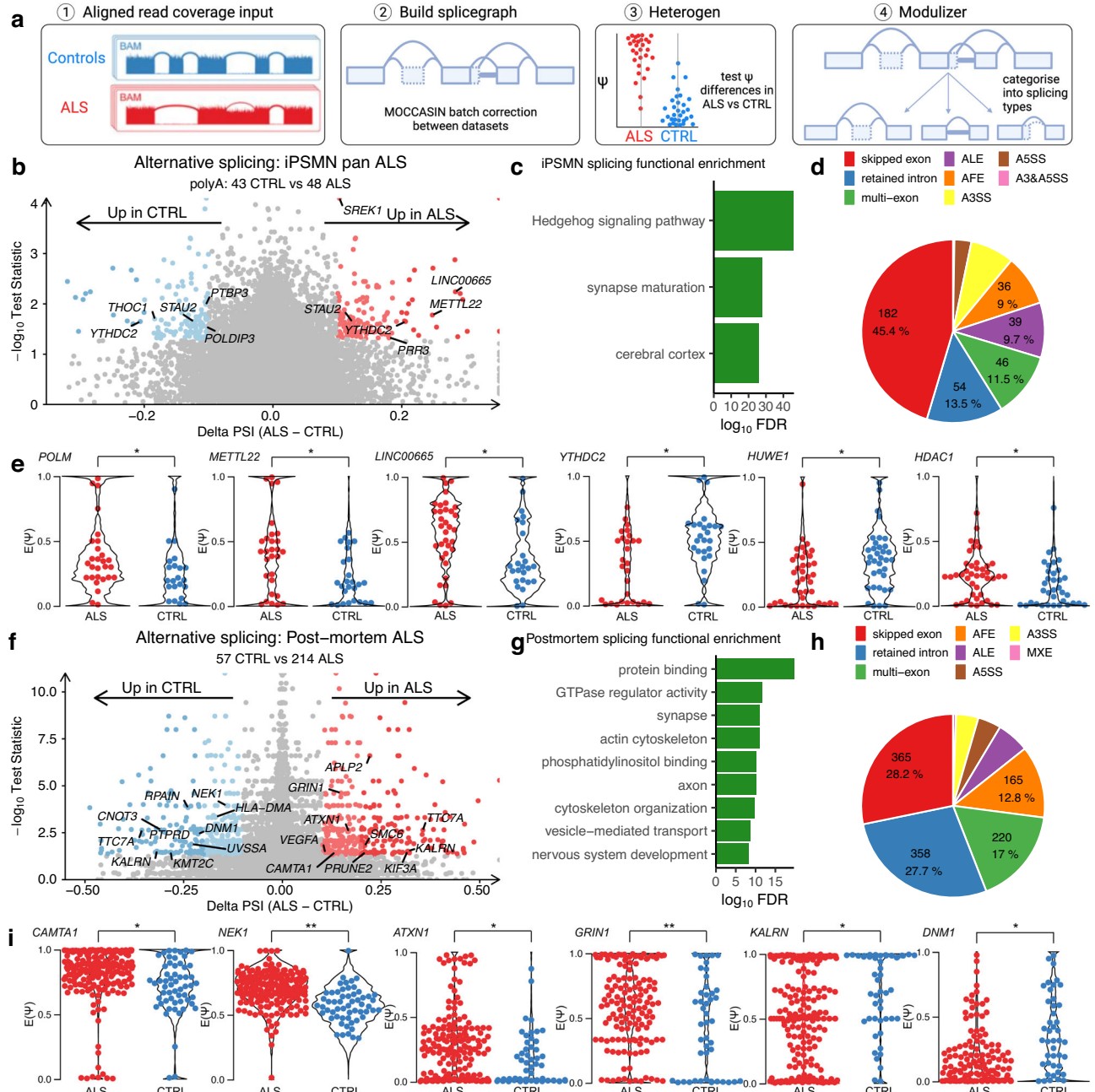

**Fig. 5 | Alternative splicing alterations in ALS iPSMNs. a** Splicing analysis of ALS and control iPSMNs with MAJIQ[50]. **b**, **f** Differential splicing in ALS versus control iPSMNs and post-mortem using the TNOM test. Events with $P < 0.05$ and Δ PSI (ALS - CTRL) > 0.1 are coloured red and < −0.1 blue. **c**, **g** Functionally enriched terms amongst genes with differential alternative splicing in iPSMNs and post-mortem using the hypergeometric test. **d**, **h** Categorisation of differential local splice variants into basic splicing types using MAJIQ modulizer in iPSMNs and post-mortem. **e**, **i** Violin plots showing PSI values (y-axis) for ALS (red) and control samples (blue) for splice events in iPSMNs and post-mortem with $p$-values from the TNOM test. ** represents $P < 0.01$ and *$P < 0.05$.

each genetic group, but that 33 were shared amongst two mutant groups (Supplementary Fig. 17k-l; Supplementary Data 10). Amongst these were the p53 signalling and RBP gene *CNOT3* as well as other RBPs including *SRSF10, DNAJC17, SRRM1, SIDT2,* and *SREK1*.

We next sought to identify the splicing changes in the ALS post-mortem spinal cord from the NYGC cohort (214 ALS patients and 57 controls). Because of RNA degradation in post-mortem tissue, the NYGC samples were generated using ribosomal depletion instead of poly(A) library selection[53]. Comparing splicing in post-mortem ALS versus control samples revealed 842 significant local splice events in 445 unique genes (Δ PSI > 0.1, tnom $p < 0.05$; Fig. 5f; Supplementary Data 11). Amongst the differential splicing events in ALS post-mortem were 4

established ALS genes (*CAMTA1, NEK1, ATXN1,* and *GRIN1*), 19 genes with altered splicing in neuronal nuclei depleted of TDP-43 (including *KALRN, PRUNE2, DNM1*) and 11 genes with altered splicing in TDP-43 knockdown (*e.g. KALRN, DNM1, NEK1*; Fig. 5i, Supplementary Data 11). We also found a significant number of genes that encode DNA damage repair factors (e.g. *APTX, CENPX, RIF1, CNOT3*; Fisher $p = 6.7 \times 10^{-16}$) and RBPs (e.g. *EIF4E3, HNRNPUL1, ATXN1, SRSF5*; Fisher $p = 1.3 \times 10^{-26}$).

Functional enrichment analysis confirmed that the 445 genes with altered splicing were involved in protein binding (FDR = $2.5 \times 10^{-10}$) and neuron compartments (FDR = $1.6 \times 10^{-6}$, Fig. 5g). Of the 842 differential splicing events, 178 (21.1%) involved de novo splice junctions and of these, 21 were cryptic exons of which EP400, PLEKHA1, BMP2K, and

KMT2C overlapped with TDP-43 depletion[43]. The majority of differential splice events harboured IR (462 / 842, 55%) and IR was the second most common splicing type, accounting for 27.7% of all post-mortem splicing events, behind skipped exons (28.2%; Fig. 5h). Intersecting genes exhibiting altered splicing in post-mortem tissue with iPSMNs revealed 12 overlapping genes including synaptojanin 1 (SYNJ1), kinesin 1B (KIF1B), dynamin 2 (DNM2), and polyA ribonuclease 3 (PAN3) as well as others involved with cytoskeletal functions (e.g. AGAP1, Cytohesin 1; Fisher exact test $p = 3.3 \times 10^{-9}$; Supplementary Data 11).

### Somatic mutation burden in ALS iPSMNs and post-mortem tissue

Genome instability triggers the DNA damage response and p53 signalling[54]. To explore somatic mutation burden in ALS iPSMNs we ran the GATK variant discovery pipeline, which detects single-nucleotide variants (SNVs), insertions and deletions (indels). Variant detection is sensitive to coverage and sequencing chemistries and so we restricted variant detection to the Answer ALS dataset (ALS $n = 238$, CTRL $n = 42$ iPSMNs). To increase the likelihood that identified variants were somatic mutations, we excluded common variants and known RNA editing sites. After adjusting for sequencing depth and donor age, across all filtered variant types we found significantly greater numbers of somatic mutations per iPSMN in ALS compared to control (Wald test $p < 2 \times 10^{-16}$; Fig. 6a). Examining each variant type revealed significantly greater numbers per iPSMN of SNVs ($p < 2 \times 10^{-16}$), insertions ($p = 1.5 \times 10^{-12}$) and deletions ($p = 1.1 \times 10^{-14}$) in ALS compared to control (Supplementary Data 12). Assessing the relative contributions of each base substitution type revealed largely similar SNV spectrum profiles in ALS and CTRL iPSMNs, predominantly composed of T > C followed by C > T substitutions (Supplementary Fig. 18a). Examining each genetic background separately revealed that sporadic iPSMNs showed significant increases in SNVs ($p < 2 \times 10^{-16}$), insertions ($p = 5.7 \times 10^{-15}$), and deletions ($p < 2 \times 10^{-16}$), C9orf72 mutants showed significant increases in SNVs ($p < 2 \times 10^{-16}$) and insertions ($p = 4.3 \times 10^{-10}$), whilst SOD1 mutants showed significant increases in insertions only ($p = 7.3 \times 10^{-10}$; Fig. 6b).

As iPSMNs are derived from peripheral cell types and the reprogramming process itself can induce somatic mutations[55], genome instability in iPSMNs may not be representative of cell types in the central nervous system. To address this, we next examined ALS post-mortem spinal cord tissue for genome instability using the GATK variant discovery pipeline. After filtering common variants and RNA editing sites as well as adjusting for sequencing depth, sequencing instrument, and age at death, we found a significantly greater number of somatic mutations in ALS compared to control post-mortem spinal cord ($p < 2 \times 10^{-16}$; Fig. 6c). Examining the variant types revealed that there were significantly greater numbers of SNVs ($p < 2 \times 10^{-16}$), insertions ($p < 2 \times 10^{-16}$) and deletions ($p = 4.6 \times 10^{-6}$) in ALS compared to control post-mortem (Supplementary Data 13). As with iPSMNs, base substitutions in post-mortem tissue were largely composed of T > C substitutions with similar SNV spectrum profiles between ALS and CTRL post-mortem tissue (Supplementary Fig. 18b). By assessing each genetic background independently, we found significantly greater numbers of somatic mutations in sporadic ($p < 2 \times 10^{-16}$), C9orf72 ($p < 2 \times 10^{-16}$) and SOD1 ($p = 1.8 \times 10^{-4}$) subgroups compared to control post-mortem tissue. With exception of SNVs in SOD1 cases ($p = 0.6$), we found significant increases for SNVs, insertions, and deletions in sporadic, C9orf72, and SOD1 subgroups (Fig. 6d). This greater burden of somatic mutations in ALS may contribute to the heightened DNA damage response and implicates defective DNA damage repair in ALS.

### Landscape of gene fusions across ALS

Gene fusions are another important class of genome alteration that can arise from the repair of damaged DNA. Fusions involve two genes becoming juxtaposed due to genomic structural rearrangements, including inversions and translocations. To explore whether ALS iPSMNs exhibit increased numbers of gene fusions, we ran the STAR Fusion pipeline on 11 paired-end RNA-seq iPSMN datasets (306 ALS, 90 CTRL iPSMNs, Table 1)[56]. We identified a total of 292 unique gene fusions in ALS iPSMNs and 152 unique gene fusions in control iPSMNs, with 91 shared in both conditions (Supplementary Fig. 18c). Among the 201 gene fusions identified in ALS iPSMNs but not in controls, were fusions affecting genes implicated in ALS including VAPB–APCDD1L-DT, ATXN1–ZFYVE27, TUBA1A–NEFM and OSTF1–APP. Furthermore, of the 292 gene fusions identified in ALS iPSMNs, 14 affected genes also exhibited altered splicing, supporting the possibility of trans-splicing, that post-transcriptionally joins exons from separate pre-mRNAs[57]. By comparing the proportion of each unique gene fusion in ALS with CTRL iPSMNs, we identified 9 gene fusions with a significantly greater burden in ALS iPSMNs (Supplementary Data 14). Interestingly, these mostly involved long noncoding RNAs (lncRNAs), for example, the gene fusion with the greatest burden in ALS was a neighbour fusion between the lncRNA LINC01572 and PMFBP1 (OR 3.3, 95% CI 1.6-Inf, Fisher's exact test p 0.001).

Examining the read-depth and age-adjusted frequency of gene fusions per iPSMN, revealed significantly greater numbers of gene fusions in ALS compared to control samples (Wald test $p = 0.006$; Fig. 6e). Comparing the frequencies of each type of gene fusion between ALS with control iPSMNs, revealed trends towards increased numbers in ALS iPSMNs for each of gene neighbours ($p = 0.2$), overlapping neighbours ($p = 0.86$), distant intra-chromosomal ($p = 0.90$), although this was significant only for inter-chromosomal fusions ($p = 0.0002$; Supplementary Fig. 18d). Comparing the number of gene fusions per iPSMN in each ALS genetic group with their respective dataset controls, revealed significantly greater numbers of gene fusions in C9orf72 ($p = 0.002$) and SOD1 mutant groups ($p = 0.0003$) as well as nonsignificant increases in sporadic ($p = 0.26$), TARDBP ($p = 0.17$) and FUS mutants ($p = 0.54$; Fig. 6f).

To identify whether ALS post-mortem tissue also displays increased gene fusions, we performed fusion discovery on post-mortem spinal cord. There were a total of 177 unique gene fusions in ALS and 96 in control post-mortem, with 71 shared. Comparing unique gene fusions between iPSMNs and post-mortem revealed 55 in ALS iPSMNs and post-mortem, of which 27 were also present in CTRLs (Supplementary Fig. 18c). Burden analysis identified 13 gene fusions with a significantly greater burden in ALS post-mortem tissue (Supplementary Data 14). As with iPSMNs, these mostly involved lncRNAs and the fusion with the greatest burden in ALS post-mortem was a gene neighbour fusion between the lncRNA AL353138.1 and PTCHD4 (OR 4.2, 95% CI 2.3-Inf, Fisher's p $1.1 \times 10^{-5}$).

After adjusting for read coverage, sequencing instrument, and age at death in a generalised linear model, we identified significantly greater numbers of gene fusions in ALS compared to controls ($p = 7.2 \times 10^{-6}$; Fig. 6g). Comparing the frequencies of each type of gene fusion between ALS with control post-mortem, revealed significantly increased numbers in ALS of gene neighbours ($p = 1.4 \times 10^{-4}$) and local rearrangements ($p = 0.02$), as well as nonsignificant increases in overlapping neighbours ($p = 0.6$), distant intra-chromosomal ($p = 0.12$), and inter-chromosomal fusions ($p = 0.8$; Supplementary Fig. 18e). Comparing the number of gene fusions per post-mortem sample in each ALS genetic group with controls, revealed significantly greater numbers of gene fusions in sporadic ALS ($p = 8.2 \times 10^{-6}$), C9orf72 ($p = 0.03$) and SOD1 mutant subgroups ($p = 0.0009$; Fig. 6h). Taken together, these findings reveal enrichment of SNVs, indels and gene fusions in ALS iPSMNs and post-mortem tissue, which we propose is a genomic signature arising from elevated DNA damage and/or impaired DNA repair.

## Discussion

Here, we present a comprehensive catalogue of transcriptome changes in ALS, comprising 429 iPSMNs and 271 post-mortem spinal cord

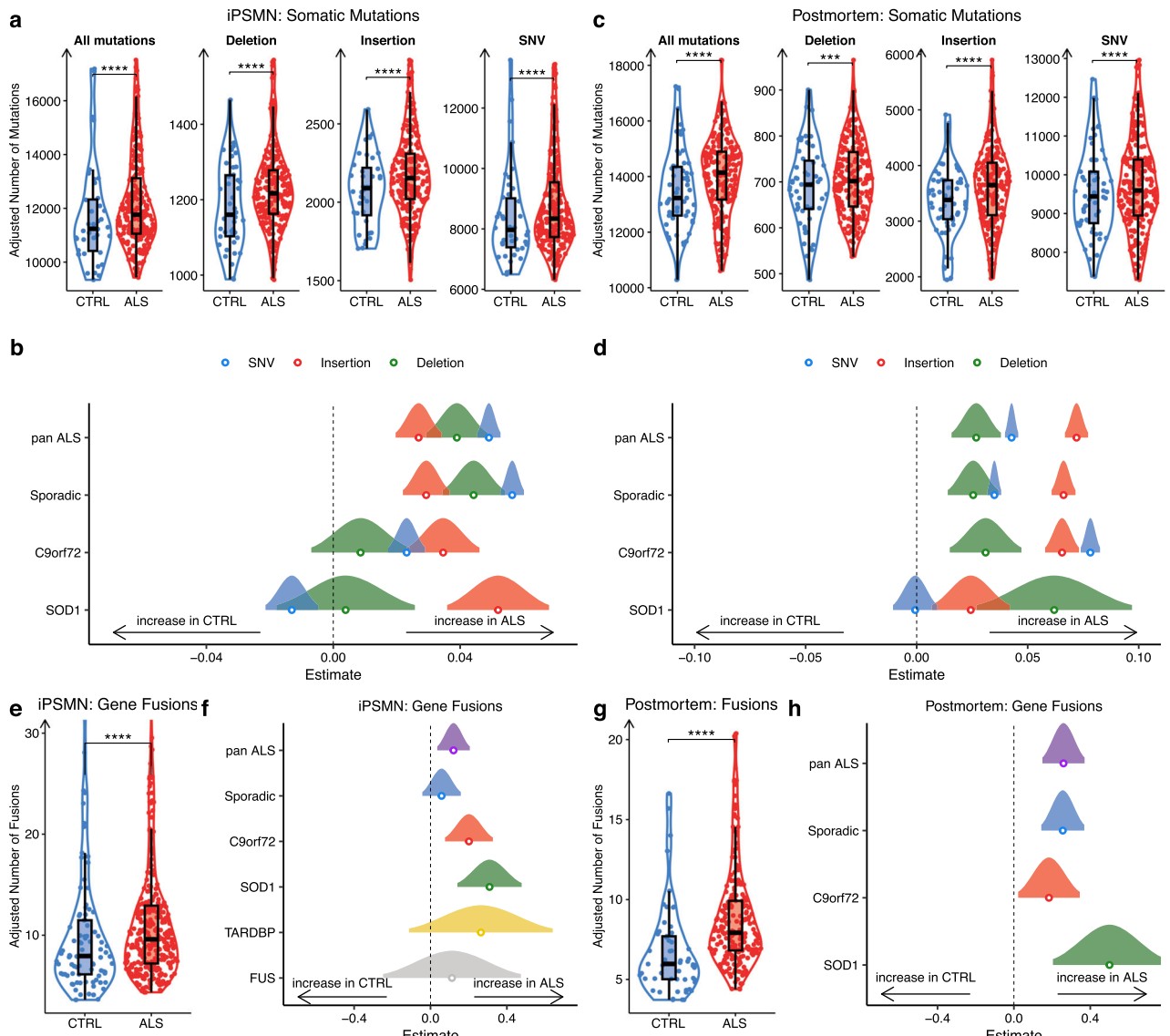

**Fig. 6 | ALS iPSMNs and post-mortem tissue accumulate somatic mutations and gene fusions. a** Violin plots showing the partial residuals of somatic mutations, controlling for age and read depth, identified in Answer ALS iPSMNs in ALS (red, $n = 238$) and CTRL (blue, $n = 42$) samples, for all mutation types, insertions, deletions, and single-nucleotide variants (SNV). Statistics are from the generalised linear model Wald test using a Poisson distribution. **b** Forest plot showing the generalised linear model point estimate and 95% confidence interval of changes in mutation types (SNV, blue; insertion, red; deletion, green) in ALS genetic subgroups versus controls. The vertical dashed line indicates no difference, to the right of the dashed line indicates an increase in ALS. **c, d** As for (**a, b**) except in NYGC post-mortem spinal cord samples ($n = 214$ ALS, $n = 57$ controls). In addition to age and read depth, the sequencing instrument is also controlled for. **e** Violin plots showing the partial residuals of gene fusions in CTRL (blue, $n = 90$) and ALS (red, $n = 306$) in paired-end sequenced iPSMNs, controlling for age, read depth and dataset. Statistics are from the generalised linear model Wald test using a Poisson distribution. **f** Forest plot showing the generalised linear model point estimate and 95% confidence interval changes in each genetic subtype versus controls. **g, h** As for (**e, f**) except in post-mortem ($n = 214$ ALS, $n = 57$ controls), controlling for age, read depth, dataset and sequencing instrument. In the boxplots, whiskers (error bars) represent 1.5 times the interquartile range, the hinges correspond to the first and third quartiles, and the centre represents the median. **** represents $P < 0.0001$, *** $P < 0.001$, ** $P < 0.01$, * $P < 0.05$.

---

samples, spanning 10 different ALS mutations and sporadic ALS. Systematically integrating these data provides a harmonized resource with substantially improved statistical power to detect perturbations across ALS. We identified that ALS iPSMNs and post-mortem tissue display an augmented DNA damage response, most notably in cases with TDP-43 proteinopathy. This was combined with the accumulation of somatic mutations and gene fusions, which may contribute to the elevated DNA damage response. Our findings add to the growing body of evidence implicating defective DNA damage repair and induction of the DNA damage response in ALS[58].

Our findings support previous smaller-scale studies showing p53 activation in ALS, particularly with *C9orf72* repeat expansions[47,59–66].

However, we also found that p53 activation was accompanied by increased somatic mutations and gene fusions across diverse ALS subgroups. This finding in both iPSMNs and post-mortem spinal cord tissue indicates that the DNA damage response in ALS begins early and persists in the later stages of the disease. We identified the greatest increase in p53 activity in *C9orf72* mutants, consistent with reports that *C9orf72* repeat expansion induces DNA damage, likely mediated by dipeptide repeat proteins and the formation of R loops and G quadruplexes[47,59–62]. However, we also found that p53 was strongly and significantly activated in *TARDBP* and sporadic subgroups. This finding in sporadic cases is particularly important since they represent ~80% of ALS cases, but have the least prior evidence for p53 activation[67,68].

Although we found p53 activation in ALS cases that lack TDP-43 pathology (FUS and SOD1 mutants), the magnitude of activation was substantially weaker than that in TDP-43 ALS cases, raising the possibility that TDP-43 contributes to the DNA damage response. In support of this, we found significant p53 upregulation in neuronal nuclei depleted of TDP-43, TDP-43 knockdown, and overexpression models, which is consistent with the established role of TDP-43 in DNA repair and p53 activation[40,47,65,69–71]. TDP-43 depletion results in the accumulation of DNA damage, whereas TDP-43 overexpression leads to a pro-apoptotic phenotype, which can be partially rescued by p53 inhibition[47,65,69–71]. Together, these results suggest that TDP-43 pathology exacerbates the DNA damage response, and subsequent p53 activation may promote motor neuron death in ALS.

Whether somatic mutations and p53 activation cause motor neuron degeneration in ALS remains to be established. It has been shown that the acquisition of DNA damage in post-mitotic neurons promotes cell cycle re-entry, attempting to activate cell cycle-associated DNA repair pathway; however, this triggers an apoptotic outcome[72]. Furthermore, studies of p53 ablation and inhibition in C9orf72 and TDP-43 mutant iPSMNs, mouse and fly models have demonstrated phenotypic rescue, supporting a pathogenic role of the DNA damage response[47,71,73]. Another possibility is that the DNA damage response indirectly causes neuronal dysfunction through overproduction of reactive oxygen species, mitochondrial dysfunction, accumulation of toxic proteins, or invoking the neuroinflammatory cGAS-STING pathway[74,75]. Thus, DNA damage may directly contribute to neuronal death, or perturb other mechanisms that maintain healthy neuronal function.

The observed relative hypermutation rate in ALS iPSMNs and post-mortem tissue supports the possibility that p53 activation is a reactive change in genomic instability. This begs the question of which upstream mechanisms drive DNA damage in ALS. Although many genetic causes of ALS are linked to defective DNA damage repair (FUS, TARDBP, SOD1, C9orf72, NEK1, SETX, VCP), the mechanism in sporadic patients remains enigmatic[75]. A post-mortem study of 16 sporadic ALS patients reported elevated levels and function of the base-excision repair enzyme APEX1, which may represent an appropriate reactive activation of DNA repair pathways to DNA damage[68]. Thus, p53 activation in sporadic ALS may be secondary to accelerated DNA damage rather than defective DNA repair and is possibly a consequence of other ALS pathogenic mechanisms such as mitochondrial dysfunction perturbed, autophagy, accelerated ageing and TDP-43 mislocalisation[76].

Despite the large sample size, we found relatively few differentially expressed genes across the ALS spectrum in iPSMNs, consistent with reports that the iPSMN model shows only mild differential gene expression signatures[77]. However, this likely reflects the heterogeneity between ALS genetic backgrounds and the conservative strategy utilised. Even within genetic subgroups, there is substantial heterogeneity, no more so than in the sporadic subgroup. Indeed, despite being the most well-powered, representing two-thirds of all ALS iPSMNs, sporadic iPSMNs displayed only four differentially expressed genes, reminiscent of what we previously reported in sporadic ALS iPSC-derived astrocytes[78]. Sporadic ALS encompasses patients with diverse genetic susceptibilities and likely includes patients carrying unknown pathogenic gene mutations[79]. Additionally, environmental risk factors play an important role in ALS aetiology, and while the iPSC model is an elegant approach to model sporadic ALS, a notable limitation is that it does not reproduce patients' environmental exposures or aging signatures[80]. In contrast to iPSMNs, the post-mortem spinal cord exhibited thousands of ALS differentially expressed genes, and ALS subgroups showed strongly correlated gene expression changes. This presumably reflects the extensive changes at the end-stage of ALS and supports previous ALS post-mortem studies reporting widespread reactive gliosis and neuroinflammation[9,12,41,81]. Thus, it is unlikely that these dramatic gene expression changes in post-mortem samples are

motor neuron-specific, which is supported by single-cell transcriptomics and cell-type deconvolution analyses that revealed that the motor neuron signal is largely masked by glial cell types[11,12,82–85].

In our compendium of iPSMNs, we highlight how the expression of rostro-caudal and dorso-ventral spinal cord markers vary between differentiation protocols (Supplementary Data 1, Supplementary Fig. 4–7). While there are no clear links between differentiation protocols and the expression of MN domain markers, these findings should guide the field in optimising in vitro culturing strategies to promote the maturation of iPSMNs[15]. Although we safeguard against dataset batch effects by using a generalised linear model that comprehensively adjusts for known confounding variables, an inherent limitation of integrative studies is that confounders may remain unknown or masked. We minimised these risks by utilising extensive quality control and validation subgroup analysis.

In summary, our findings show diverse mechanisms of motor neuron dysfunction in ALS at the RNA level, demonstrating that large-scale analyses can uncover ALS-associated pathway abnormalities. These findings illustrate the utility of integrated transcriptome analysis for ALS research.

## Methods

### Search strategy

We systematically reviewed RNA-seq databases, including Gene Expression Omnibus (GEO), NCBI sequence read archive (SRA), EBI arrayExpress, European Nucleotide Archive (ENA), synapse.org and manually searched reference lists as well as ALS data portals of relevant studies. The search strategy included keywords relating to ALS and motor neurons: "amyotrophic lateral sclerosis", "ALS", "motor neuron*", and "MND". Sequencing datasets matching human species and date range [inception – 2022] were selected, yielding a total of 503 unique datasets. The final search was conducted on 30th June 2022 (Supplementary Fig. 1).

### Eligibility criteria

We evaluated all datasets that had undergone short-read bulk RNA-sequencing (RNA-seq) from human iPSMN samples derived from individuals with ALS and non-ALS controls (healthy individuals or isogenic correction), regardless of the RNA extraction kit (e.g. Qiagen RNeasy mini kit, Invitrogen TRIZol), library preparation (poly-adeynlated or total ribosomal-depleted), short-read lengths (range 50-300 base pairs), read sequencing (single or paired-end), sequencing instrument (e.g. Illumina NovoSeq 6000, HiSeq 2500), or sequencing depth (Supplementary Data 1). All ALS subtypes were included, and the definition of ALS used by each dataset was accepted. In datasets with multiple time points through iPSMN differentiation only the final most terminally differentiated time point was utilised[17].

We excluded datasets that (i) had not undergone an accepted spinal motor neuron differentiation protocol using the steps detailed in Sances et al. [15], (ii) failed RNA-seq quality control measures (Supplementary Data 2), (iii) failed spinal motor neuron identity based on the expression of established spinal cord dorso-ventral and rostro-caudal markers (Supplementary Fig. 4–7), or (iv) exhibited unadjustable batch effects between ALS and control samples (e.g. different RNA library strategies or sequencing platforms). Long-read sequencing as well as single-cell and single-nuclear RNA-seq datasets were excluded.

### RNA-seq processing, integration and quality control

The iPSMN differentiation protocol method (including induction, specification and terminal differentiation) as well as RNA-seq library strategy (RNA extraction, library preparation, sequencing instrument and read metrics) for each dataset are noted in Supplementary Data 1. Raw RNA-seq reads (fastq files) and accompanying metadata were downloaded using nfcore/fetchngs v1.9 pipeline[86] and pysradb v1.3

using the sample SRA accession number. Reads were processed using the nfcore/rnaseq v3.9 pipeline[86]. Raw reads underwent adaptor trimming with Trim Galore, removal of ribosomal RNA with Sort-MeRNA, alignment to Ensembl GRCh38.99 human reference genome using splice-aware aligner, STAR v2.7.1 and BAM-level quantification with Salmon. Samples were subjected to extensive RNA-seq quality control utilising FastQC, RSeQC, Qualimap, dupRadar, Preseq, and SAMtools and results were collated with MultiQC. Samples that passed the nfcore/rnaseq quality control status checks were included in the integrated analysis (Supplementary Data 2). The median read depth was 115 (range 6 – 164) million reads per sample.

We used principal component analysis (PCA) and unsupervised clustering to interrogate the batch effects of clinical variables, iPSMN protocols and RNA-seq strategies between samples and datasets. Gene counts were normalised for library size and transformed on a $\log_2$ scale using the variance stabilizing transformation function in DESeq2. Principal components were calculated based on the 500 highest variance genes using the plotPCA function and individual PC gene loadings were extracted with the prcomp function. Samples clustered into two groups based on library preparation (poly-adenylated or total ribosomal depletion). We examined the motor neuron transcriptomic identities of iPSMNs by clustering using the ComplexHeatmap package based on the expression of canonical neuronal and glial cell type markers as well as dorsoventral[87] and rostrocaudal (HOX) gene markers. The Lee et al. dataset was excluded due to unadjustable RNA library batch effects between ALS (total ribosomal-depletion) and control (poly-adenylated) samples as well as inadequate neuronal marker expression when assessing iPSMN identity[24]. We excluded three control samples in AnswerALS that whole-genome sequencing revealed to have pathogenic ALS mutations. Additionally, 4 Answer ALS iPSMNs from patients with non-ALS motor neuron diseases were excluded. NeuroLINCS consists of 3 distinct iPSC protocols (iMNs, diMNs and undifferentiated iPSCs; Supplementary Data 1), of which only the iMN and diMN batches were included. Sex was confirmed by examining the expression of the X chromosome gene *XIST* (female) and Y chromosome genes *KDM5D, DDX37, RP54Y1* and *EIF4Y* (male).

## Modelling differential expression

STAR aligned and Salmon quantified transcript abundance were summarised at the gene-level using tximport in R v4.1.3. Differential gene expression analysis was then fitted using DESeq2[88]. The integrated analysis results of ALS iPSMNs were generated by comparing the ALS versus control groups using the Wald test, controlling for sex differences and dataset variation with the design formula $\sim sex + dataset + condition$. This design controls for technical variation due to library preparation (nested within the dataset variable), which was the main factor driving PCA structure (Supplementary Fig. 2-3), thereby increasing the sensitivity for identifying differences due to ALS. We orthogonally estimated technical variation using the RUVg method that takes empirically defined negative control genes to estimate low-rank technical variation in the data, specifying 5 RUV factors[89]. To examine the effect of each ALS genetic background on gene expression a similar approach was used, comparing the ALS versus control samples using the design formula $\sim sex + dataset + genetic\ subgroup$. For these subgroup analyses, control samples from each dataset were only utilised if the dataset also exhibited the relevant ALS genetic background. Non-TDP-43 ALS (TDP-43 pathology negative) samples were defined as SOD1 or FUS mutants.

Results for each genetic background were correlated by matching the Wald test statistic for each gene followed by Pearson correlation. In all analyses, genes were considered differentially expressed at FDR < 0.05. Significantly up- and down-regulated differentially expressed genes were used as input to functional over-representation analyses to identify enriched pathways using g:Profiler2. g:Profiler2 searches the following data sources: Gene Ontology (GO; molecular functions,

biological processes and cellular components), KEGG, REAC, Wiki-Pathways, CORUM and Human Phenotype Ontology. g:Profiler2 reports the hypergeometric test p-value with an adjustment for multiple testing using the Bonferroni correction. Over-represented function categories are plotted in bar charts, where the top significant terms were manually curated by removing redundant terms. Gene Set Enrichment Analysis (GSEA) was performed using FGSEA on GO:0072331 (signal transduction by p53 class mediator) gene set. The decoupleR package was used to estimate PROGENy signalling pathway activities and DoRothEA TF regulon activities inferred from gene expression changes[34]. PROGENy and DoRothEA weights are based on perturbation experiments that are not specific to motor neurons. Their signalling pathways may activate diverse downstream gene expression programmes depending on the cell type and perturbing agent utilised[90].

For analysis of protein changes in iPSMNs, processed proteomic data (.csv matrix of peptide intensities) were downloaded from the AnswerALS data portal. Mass spectrometry was performed and processed as reported in AnswerALS[23]. Peptide intensities were processed following the LIMMA Peptide tutorial instructions using log2 transformation and specifying a contrast of ALS versus control iPSMNs[91]. Quality control of peptide intensities across samples before and after log2 transformation was performed. Outlier samples were identified using sample-sample distances with the MDS plot and outlier peptides were detected using the mean-variance relationship. As no differentially expressed proteins in ALS versus control were identified using FDR < 0.05, a more lenient threshold of $p$-value <0.05 was utilised.

## Post-mortem tissue

Post-mortem spinal cord ALS RNA-seq samples were derived from samples from the New York Genome Centre (NYGC) ALS consortium. Samples from non-spinal cord sites were excluded. Raw reads were acquired from accession GSE137810 and processed using the same pipeline described above. In cases where multiple spinal cord samples were available from donors, only the cervical cord sample was included. Differential expression results for post-mortem spinal cord ALS were calculated by comparing ALS versus control samples, accounting in the design for the RNA library preparation method, sex and the site of the spinal cord tissue (cervical, thoracic or lumbar) with the formula $\sim library\ pep + sex + sample\ source + condition$. Post-mortem genetic subgroup analyses were performed using the design $\sim library\ prep + sex + sample\ source + genetic\ subgroup$. Post-mortem subgroup analyses were limited by the sample size for FUS ($n = 2$) and there were no TARDBP mutants available.

## TDP-43 depletion and overexpression

The post-mortem brain TDP-43 FACS sorted neuronal nuclei RNA-seq dataset was acquired from accession GSE126543 and processed using the same pipeline described above. Only NeuN-positive samples were utilised. Differential expression results were calculated by comparing TDP-43 negative versus TDP-43 positive samples.

For analysis of artificially TDP-43 depleted human cells, we searched RNA-seq databases for TDP-43 knockdown datasets. We found 8 datasets[20,31,41–46], of which one had low TARDBP expression in controls and did not achieve >60% TARDBP reduction with depletion and was excluded (Supplementary Data 8)[20]. Differential expression results were calculated by comparing TDP-43 depleted versus control samples, accounting for dataset batch effects in the design with the formula $\sim dataset + sex + condition$. Primary mouse neurons overexpressing TDP-43 were utilised from GSE162048. Neurons transduced with lentivirus overexpressing TDP-43 for 20 h were utilised and compared to control neurons.

## Alternative splicing analyses

All modes of alternative splicing were analysed using MAJIQ v2.4[49,50] on poly(A) selected RNA library iPSMNs samples. For

post-mortem NYGC samples only total ribosomal-depleted libraries were available. STAR aligned BAMs were used as input to the MAJIQ splice graph builder using Ensembl GRCh38.99 transcript annotation. The same batch effects from the differential gene expression analyses were controlled for using *MOCCASIN*[51]. Differential splicing was calculated using the *MAJIQ heterogen* function, which is designed for examining splicing across large and heterogeneous datasets. A threshold of 10% ΔΨ and TNOM *p*-value <0.05 was used to call significant splicing changes between groups. Changes in each specific class of splicing were examined using the *Voila modulize* function that breaks down the complex local splice variants into the classic binary splicing events (e.g. exon skipping or intron retention [IR]). For comparison of ALS iPSMN splicing events with TDP-43 deletion, RNAseq fastq files from iNeurons were downloaded from ENA PRJEB42763. These were processed using nfcore/rnaseq followed by MAJIQ v2.4 using the *MAJIQ deltapsi* function, which is more appropriate than *heterogen* for a small single batch homogenous experimental replicate dataset.

### Somatic mutation and gene fusion detection

Somatic mutations were detected in iPSMNs and post-mortem tissue using the nfcore/rnavar pipeline v1.0.0[86], which is based on GATK v4.2.6 short variant discovery workflow. Variant discovery is highly sensitive to coverage and sequencing chemistries and so for iPSMNs only the Answer ALS dataset was used for variant detection thus avoiding confounding batch effects between multiple datasets. For post-mortem tissue, the spinal cord samples from the NYGC dataset were included, although within NYGC there exist two sequencing batches (NovoSeq 6000 with 200 bp read lengths [$n = 186$]; HiSeq 2500 with 250 bp reads [$n = 85$]). Raw RNA-seq reads were mapped using STAR in two-pass mode. SplitNCigarReads tool was used to reformat alignments that span introns for the HaplotypeCaller. Base-Recalibrator and ApplyBQSR were used for base quality recalibration. Single-nucleotide variants (SNVs) and indels were called using the HaplotypeCaller and variants were filtered using VariantFiltration specifying a minimum phred-scaled confidence threshold of 20 and minimum quality depth of 2.0. Variants overlapping with the dbSNP database and RNA editing variants from the REDIportal v2.0 were filtered out using VCFtools. Variants were annotated using snpEff and Ensembl VEP whilst VCFtools vcf-annotate –fill-type module was used on the filtered output to classify variants into SNVs, insertions or deletions. The characteristics of mutations were assessed using the MutationalPatterns package v3.6.0, which summarises the number and proportions of each type of base substitution[92]. To compare the number of somatic mutations per sample in ALS versus CTRL groups, a generalised linear model was fit specifying a Poisson distribution adjusting for differences in read coverage per sample and donor age using a spline: $variant\ count \sim condition + rcs(read\ depth, 3) + rcs(age, 3)$. For post-mortem tissue we included an additional term to adjust for the sequencing batch: $variant\ count \sim condition + batch + rcs(read\ depth, 3) + rcs(age, 3)$.

Gene fusions were identified in iPSMNs and post-mortem tissue using the nfcore/rnafusion pipeline v2.0.0[86], utilising the STAR-Fusion v1.10.1[56] workflow on paired-end RNA-seq datasets. Raw RNA-seq reads were aligned using STAR to identify chimeric transcripts, which are defined as a part of a read aligning to one gene and another part of the same read to a different gene (split) or when each end of a paired read set aligns to different genes (spanning). STAR Fusion applies numerous filters to avoid spurious fusion detection including removing chimeric reads overlapping with sequence-similar regions and removing duplicate paired-end alignments. Fusion events were filtered using the FusionFilter module default settings for spanning and split reads. Fusion events with fusion fragments per million (FFPM) < 0.1 were removed. Gene fusion events are classified using FusionAnnotator

module into inter-chromosomal and intra-chromosomal. Intra-chromosomal is subclassified into local gene orientation rearrangements, neighbours (<100 kb apart), overlapping neighbours (genes span overlap by at least 1 base pair) and distant (>100 kb apart). To compare the number of fusions per sample in ALS versus CTRL groups, a generalised linear model was fit specifying a Poisson distribution adjusting for differences in read coverage per sample and donor age and batch effects between datasets: $fusion\ count \sim condition + dataset + rcs(read\ depth, 3) + rcs(age, 3)$. For post-mortem tissue we included an additional term to adjust for the sequencing batch: $fusion\ count \sim condition + batch + rcs(read\ depth, 3) + rcs(age, 3)$. To detect burden differences of individual gene fusions in ALS compared to CTRL samples, the Fisher test was used to calculate the Odds Ratio, 95% confidence interval and p-value. To examine the effect of each ALS genetic group, controls were only utilised if the dataset also included ALS samples from the relevant genetic group. In the violin plots, the observed number of variants and fusions were controlled for read depth and donor age by using the partial residuals from the linear regression model, which was performed using the partialize function from the tools package. For genetic subgroups, forest plots were generated using the plot_summs function, which depicts the regression coefficient from the linear regression model showing the point estimate and 95% confidence interval.

No statistical method was used to predetermine the sample size. The experiments were not randomised. The Investigators were not blinded to allocation during experiments and outcome assessment. Schematics were created with BioRender.com. The tidyverse suite of packages was used for tidying data in R (e.g. tidyr 1.2.0, tibble 3.1.7, ggplot2 3.3.6). In the boxplots, whiskers (error bars) represent 1.5 times the interquartile range, the hinges correspond to the first and third quartiles, and the centre represents the median. An interactive web resource for browsing the processed sequencing data is available for exploration at https://oliverziff.shinyapps.io/als_genome_instability/.

### Reporting summary

Further information on research design is available in the Nature Portfolio Reporting Summary linked to this article.

## Data availability

iPSMN raw sequencing data used in this study are available in public repositories under accession numbers shown in Table 1. Post-mortem raw sequencing data is accessible at GSE137810 and https://collaborators.nygenome.org/. The accession numbers for the TDP-43 depletion raw sequencing data used are in Supplementary Data 8. Some raw data have restricted access (NeuroLINCS dbGaP Accession number: phs0001231.v2.p1; AnswerALS database). Granting access to these is beyond the control of the authors. Access can be obtained by applying to the relevant Data Access Committees. AnswerALS requires a signed DUA to have full access. An interactive web resource for browsing the integrated analysis results can be viewed at https://oliverziff.shinyapps.io/als_genome_instability/. Source data are provided with this paper.

## Code availability

Full code to reproduce analyses and figures is available through GitHub at https://github.com/ojziff/als_genome_instability.

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

## Acknowledgements

We thank the Answer ALS, NeuroLINCS and New York Genome Center consortia for making their data publicly available, and to Terry Thompson and Barry Landin, for their help with access. We are grateful for discussions and advice from Jesper Svejstrup, James Lee, James Briscoe, Rory Maizels, the MAJIQ team as well as the nfcore community, particularly Hashil Patel, Phil Ewels, Praveen Raj, Martin Proks and Annik Renevey. We thank the Francis Crick Institute scientific platforms, especially members of the high performance compute team Danny Lang and John Roche. This work was funded by the Francis Crick Institute, which receives its core funding from Cancer Research UK (FC010110), the UK Medical Research Council (FC010110), and the Wellcome Trust (FC010110). O.J.Z. holds a Crick Clinical PhD Fellowship supported by the University College London Hospitals Biomedical Research Centre (BRC689/ED/CB/100130). R.P. holds an MRC Senior Clinical Fellowship (MR/S006591/1) and a Lister Research Prize Fellowship.

## Author contributions

Study was conceived and designed by O.J.Z. Gene expression, splicing, variant and fusion analyses were performed by O.J.Z. with supervisorial input from G.K., J.H., A.M.C., R.L., C.M.R., N.M., K.L., A.A.C. and R.P. Interpretation of differentiation protocols was performed by J.N. and interpretation of motor neuron markers was performed by J.M. The manuscript draft was written by O.J.Z. with input from all authors. Interpretation of data and contributions to the write-up was provided by J.N., G.T., J.M., C.M.R., R.L., A.M.C., N.M., K.L., A.A.C., S.J.B., G.K., and J.H.

## Funding

## Competing interests

N.M. has stock options in and has consulted for Achilles Therapeutics and holds European patents relating to targeting neoantigens (PCT/EP2016/ 059401), identifying patient response to immune checkpoint blockade (PCT/ EP2016/071471), determining HLA LOH (PCT/GB2018/052004), predicting survival rates of patients with cancer (PCT/GB2020/050221). The remaining authors have no conflicts of interest.
