## [Peer Review File · Nature Communications]

Reviewers' comments:

Reviewer #1 (Remarks to the Author):

Here Ziff et al perform transcriptome analysis of 429 iPSC-derived MN lines (323 ALS and 106 controls) including lines carrying 10 monogenic causes of ALS; follow up analysis is carried out in RNAseq data from 153 ALS and 80 control postmortem spinal cord samples. Transcriptome datasets were not generated by the authors but are publicly available. The major conclusion was that 'DNA damage response', and 'p53-signalling' genes are differentially expressed in ALS patients compared to controls.

The positioning of p53 as a driver of ALS-associated neurodegeneration is not new. A key paper from the Gitler group (PMID 33482083) was published in 2021 although this was not cited in this study.

The major limitation of this study is that there is no attempt to assign causation, even between the changes in p53-pathway genes and DNA damage genes. Whether transcriptome changes are up or downstream in the cascade of pathogenesis is not explored. There are also certain inconsistencies in the application of the analysis methods and I believe the experiment to measure consequences of DNA damage is flawed in design (see below).

Points:

- The functional enrichment analysis of gene expression changes is key to the implication of the p53 signalling and the DNA damage response which is the main message of the manuscript. It is not clear that a multiple testing correction applied to the enrichment analysis which is crucial. The use of statistical tests should be consistent; the NES test is used most often, could this be reported throughout? Also, please include the test statistics for each result referring to DNA damage/p53-signalling in the main text, rather than only referring to extended data.
- The perturbation PROGENy analysis is in effect a further enrichment analysis using a database of perturbation experiments which are not MN specific. These limitations should be described.
- Post mortem transcriptome analysis identifies >10,000 differentially expressed genes which seems excessive (compared to ~20) in the iPSN analyses and makes the likelihood of false positive results more likely. Can the authors comment on this? Here p53-signalling gene expression changes are reported for genetic subgroups but it should be made clear that the FUS result is not statistically significant.
- The splicing analysis is misleading. It appears that p53-signalling/DNA damage genes were not statistically enriched within differentially spliced genes. This should be made clear. The authors postulate that intron retention events may be a cause of DNA damage but not provide evidence for this.
- The authors measure numbers of SNVs and gene fusion events within expressed RNAs in AnswerALS iPSN and use an excessive number of variants to argue for excessive DNA damage in ALS. In my opinion the logic here is flawed; DNA damage causing sequence changes would be expected to produce somatic heterogeneity within the CNS. It is difficult to interpret these changes in neurons derived from peripheral cells.
- Was the meta-analysis applied to the analysis of specific genetic-subtypes or was the RUVg used to correct for batch effects in these analyses? This is not clear in the methods.

Reviewer #2 (Remarks to the Author):

The manuscript by Ziff et al is a large scale meta-analysis of multiple publically available RNAseq datasets for ALS. This includes data from Answer ALS and NeuroLINCS as well as data from an ALS postmortem spinal cord cohort. The authors find that p53 activation is observed in multiple

subgroups of ALS and that DNA damage responses are also enriched as key mechanisms in both familial and sporadic ALS, beginning early and persisting into later stages of disease. The authors also highlight genomic instability via splicing alterations, single nucleotide variants, insertions, deletions and gene fusions. There is a good rationale and a lot of data used, which is a major positive.

While the analysis is reasonable and informative, this manuscript has significant weaknesses and novelty to warrant publication in Nat Comm. First, the work is not entirely novel given the previous publication in Cell (2021) that p53 is central to c9 ALS. Second, there would need to have some additional validation of the DNA damage deficits in iPSCs and/or ALS tissues, validate protein levels for DEGs (especially for p53 and DNA damage genes), and validate R loop formation. There is an overall lack of mechanism discussed in the paper in that p53 is more of a consequence versus mechanism. Should discuss how p53 activation and DDR would impact the neurons or disease.

Other points:

3. The authors cite that OCT4 was significantly downregulated. Could this be an artifact of the iPSC modeling? Undifferentiated iPSCs? Or persistent OCT4 expression. Would need to discuss. Upregulation/activation of MAPK and p53, along with OCT4 expression may indicate a proliferating progenitor population. OCT4 also comes up quite significantly in the sALS vs control comparison. Could this set of samples be confounding? The authors should consider accounting for co-variates or filtering out specific markers.
2. Correlation values are very low and it is not clear if these are significant. Perhaps this is indicated in the extended data but should show p-value in text.
3. The base sub data is confusing. The authors indicate an increase number of C>T in ALS then repeats this info with a lower pvalue. And then suggest C>T is lower in ALS relative to control. Potentially typos?
4. Was the analysis for gene fusions events somehow normalized by sample size, considering there were many more ALS samples than control?
5. There is not discussion of sex as a variable

Reviewer #3 (Remarks to the Author):

Ziff et al. combined the huge datasets of ALS, unifying them to investigate the pathological mechanisms of ALS. The research direction of integrating patient samples and data sets to explore the ALS pathology is an interesting approach that should coincide with current trends. The problem with such an approach is the difficulty in integrating various data-sets, obtained by various methods, and with appropriate correction for bias. In addition, this reviewer is also concerned that there is no axis to evaluate the appropriateness for correcting bias, and there are limited clues to justify the results of integrated analysis. In this study, the authors pointed out the activation of p53 in ALS samples from different datasets, and confirmed a similar trend of p53 activation by using autopsy specimens of spinal cords. However, the reviewers and the readership will face a considerable difficulty in judging the scientific validity of results from the integrated data-sets of this paper. In particular, the activation of p53, which the authors have pointed out, is a phenomenon commonly observed in a wide variety of pathological conditions, and is unfortunately not new. This pathway also includes many genes that tend to be identified as a "statistically significant" pathway. This reviewer believes that more specific gene sets and specific signaling pathways may have hidden molecular clues that will lead to understanding of the pathology and the development of therapeutic methods. Therefore, it would be more important to use the presented methodology for finding a specific pathway such as a screening step, and to verify the identified results by another modality. This reviewer thinks that the presented methodology in the paper is extremely important for the scientific field, and believes that the following points, as described below, need to be revised in order to maximize the value of this paper and ensure that the method will be widely accepted in the future.

Major points

- 1) The authors combined the deposited datasets from several studies for metanalysis. However, the differentiation characteristics of iPSC-derived motor neurons, evaluated by the expression of motor neuron markers like CHAT, MNX1, ISL1, FOXP1, etc., vary among different studies in extended data

figures 4 and 5. The homogeneity of the differentiation quality is the most important factor in the discussion of disease phenotypes. Can the authors correct the gene expression status, or set a stricter threshold for selecting the datasets? These steps might help to identify a more specific pathway that is related to the motor neurons of ALS. Variation in the proportion of astrocytes in the culture dish can also be expected to cause noise in the analysis.

2) The datasets used were iPSC-derived samples obtained from different differentiation stages and different culture conditions, and were too crude to be analyzed on the same platform. Correction with appropriate intrinsic controls is also challenging. A meta-analysis of the results found in each dataset might be acceptable.

3) As mentioned in the above comments, this reviewer believes that the reproducibility of the results of such meta-analyses, based on datasets collected from different platforms, should be confirmed by other methods. Since the relationship between p53 and ALS is a known phenomenon that has already been reported elsewhere, this reviewer believes that this point has been clarified. On the flip side, the identified results can be considered as a result without novelty. Can the authors direct more focus toward new pathways, identified only by the present meta-analysis?

4) In figure 3, the authors point out that the p53 pathology is common in different ALS subtypes. This reviewer believes that it is also important to take advantage of the different ALS subtypes that are being addressed in this study. Can the authors classify the characteristics of the pathology and the signaling pathways for each ALS subtype? For example, searching for ALS with TDP43 pathology and others, and their characteristic pathways, would be a new focus for this meta-analysis?

5) In figure 4, the authors showed the altered splicing or SNVs in ALS iPSCs. To confirm the reproducibility of the findings, the authors should show the similar alteration in the other cohort, independent of the presented meta-analysis.

6) Although the meta-analysis with multiple data sets may be novel, the p53 pathway found as a result of this study has already been reported in ALS and is not novel.

Minor points

1) In the extended data of figures 4 and 5, it is unclear what "variance stabilized gene counts" mean, and what the range of variance is as described by the blue/red color?

2) The authors used "iPSNs" as the abbreviation for induced pluripotent stem cell-derived motor neurons. iPSC-derived motor neurons or iPSC-MNs etc. might be more appropriate and helpful for the readership in order to understand that this study focuses on the pathology of motor neurons with ALS.

3) As commented as major points, the correction among different batches of RNAseq and/or single-cell RNAseq analysis is important and must be quite challenging even in the unified experimental method or condition. The authors should show how the authors corrected the noise among different platforms and evaluated the corrected quality to justify the usage of these datasets.

4) The authors excluded the samples and datasets of Lee et al. and NeuroLINCS. This reviewer can intuitively understand that these samples are outliers when compared with other datasets, but this reviewer believes that the authors should clarify the threshold criteria to maintain reproducibility.

5) The title seems slightly exaggerated, and it should be revised by focusing on the methodology.

Reviewer #4 (Remarks to the Author):

I have read with interest the work by Ziff et al., titled "Genome instability underlies an augmented DNA damage response in familial and sporadic ALS human iPSC-derived motor neurons", dealing with the bioinformatics analysis of deep transcriptome data from ALS human induced pluripotent stem cell-derived motor neurons (iPSNs). The authors collected RNAseq data from 429 donors spanning 10 ALS mutations and sporadic ALS. They found an increase in the DNA damage response, which is characterized by the activation of p53 signalling.

All data were initially checked and low-quality samples were filtered out, leaving 323 ALS samples and 106 controls. Comparing these two groups revealed only 43 DE genes. In my opinion, this specific comparison is biologically questionable. Indeed, it has been performed by mixing heterogeneous RNAseq data from heterogeneous genetic backgrounds. Comparing transcriptomes from different genetic backgrounds should provide misleading results.

Additionally, the effect of the differentiation protocol has not been taken into account. It could really impact the detection of DE genes.

Regarding the identification of aberrant splicing, the authors compared again all ALS samples versus control iPSNs. I don't think different genetic backgrounds should be mixed.

About SNVs, statistics are based on the number of variants. It is not correct because the number of variants per sample is strongly connected with the coverage depth.

Although the idea is quite interesting and the results are promising, I see methodological issues that need to be addressed. Additionally, the p53 signalling pathway includes many genes. A punctual study of this pathway has to be performed and validated in independent cohorts.

Reviewer 1

Comment 1

Here Ziff et al perform transcriptome analysis of 429 iPSC-derived MN lines (323 ALS and 106 controls) including lines carrying 10 monogenic causes of ALS; follow up analysis is carried out in RNAseq data from 153 ALS and 50 control postmortem spinal cord samples. Transcriptome datasets were not generated by the authors but are publicly available. The major conclusion was that 'DNA damage response', and 'p53-signalling' genes are differentially expressed in ALS patients compared to controls. The positioning of p53 as a driver of ALS-associated neurodegeneration is not new. A key paper from the Gitler group (PMID 33482083) was published in 2021 although this was not cited in this study.

Author Response

- We thank the reviewer for these comments. Whilst we agree that p53 activation has been previously described in ALS, these have been limited to mutant forms of ALS predominantly with C9orf72¹⁻⁵. Previous reports have also been restricted in sample size usually < 10 patients in total. Conversely, this is the **first study with sufficient power** to demonstrate genome instability **across ALS subtypes in both iPSMNs and post-mortem**. This is especially novel in **sporadic ALS**, which accounts for **90%** of cases^{19,20}. The scale of this resource also enables us to **compare the magnitude of p53 activation between ALS subgroups** for the first time, exposing that TDP-43 pathology cases show substantially greater p53 activation than non-TDP-43 ALS cases - see response to Reviewer 3, Comment 6. Furthermore, this is the first study to show **increased somatic mutations and gene fusions** in both ALS iPSMNs and post-mortem.
- We apologise for accidentally omitting the excellent paper from the Gitler group¹. We note however that it predominantly used mouse neurons and was restricted to C9orf72 poly(PR) treatment. It did not examine other ALS subgroups. Other ALS iPSMN studies examining DNA damage have been restricted to very small sample sizes and mutant forms of ALS, e.g. Lopez-Gonzalez et al 2016 studied only 3 C9orf72 mutant iPSMNs³. Likewise, post-mortem studies reporting p53 upregulation have been limited to only a handful of patients, e.g. Farg et al 2016 in 10 C9orf72 patients².
- The novelty of this **resource** is that it provides the largest catalogue of human ALS associated gene alterations to date, obtained by characterising 429 human ALS iPSMN and 271 post-mortem transcriptomes. The unbiased genome-wide approach used reveals **multiple novel targets in ALS** beyond p53 e.g. RNase L expression, MAPK signalling, CNOT3 splicing. It also provides the **largest study of splicing changes in ALS iPSMNs**, not only revealing TDP-43 regulated events (e.g. POLDIP3, CAMK2B, CEP290) but also revealing new splicing events shared between ALS subgroups (e.g. CNOT3, SRSF10, DNAJC17). Thus, this paper offers substantial advances in the understanding of changes across ALS

subtypes, which is crucial for identifying therapeutic targets.

Author Changes

- To make clearer the novelty of this resource we have now shown the composition of the iPSMN and postmortem cohorts in Figure 1 Schematic sections 4 & 5:

Schematic summarising our analytic framework using iPSC-derived motor neurons (iPSMNs) and postmortem tissue to interrogate perturbations across the spectrum of ALS.

- We cite the mentioned paper from the Gitler group as well as other ALS studies of DNA damage, and clarify the novelty regarding DNA damage in this study:

Our findings support previous, smaller studies showing p53 activation in ALS, particularly with *C9orf72* repeat expansions^{1-5,20-23}. However, the novelty in this study is the finding of p53 activation, together with increased somatic mutations and gene fusions, across diverse ALS subgroups, not only in iPSMNs but also in post-mortem spinal cord tissue. We identified the greatest increase in p53 activity in

C9orf72 mutants, consistent with reports that the *C9orf72* repeat expansion induces DNA damage, likely mediated by dipeptide repeat proteins and the formation of R loops and G quadruplexes¹⁻⁵. However, we also found p53 to be strongly and significantly activated in *TARDBP* and sporadic subgroups. This finding in sporadic cases is particularly striking since they represent 90% of ALS but have the least prior evidence for p53 activation.

Comment 2

The major limitation of this study is that there is no attempt to assign causation, even between the changes in p53-pathway genes and DNA damage genes. Whether transcriptome changes are up or downstream in the cascade of pathogenesis is not explored.

Author Response

- We thank the reviewer for this comment. The finding of increased somatic mutations and gene fusions in both ALS iPSMNs and post-mortem, provides an explanation for the DNA damage response upregulation as these DNA sequence changes are upstream to the transcriptome DNA damage response.

Author Changes

- By adding somatic mutation and gene fusion analyses to the postmortem spinal cord tissue, we have further provided evidence for genome instability causing the DNA damage response in ALS - please see our response to Reviewer 1, Comment 10.
- We have added a new analysis **comparing TDP-43 ALS to non-TDP-43 ALS**, revealing that TDP-43 ALS shows significantly greater p53 activation than non-TDP-43 ALS in both iPSMNs and post-mortem.
- To examine for a **functional link between TDP-43 nuclear depletion and p53 activation**, we have now examined RNA-seq from neuronal nuclei depleted of TDP-43 from patients' brains⁶ as well as integrating 7 RNA-seq datasets from human cultured cells with TDP-43 knockdown via shRNA, siRNA, or CRISPR/Cas9-mediated inhibition⁷⁻¹³. Both neuronal nuclei depleted of TDP-43 and cells with TDP-43 knockdown show significant upregulation of p53 signaling, indicating that TDP-43 nuclear loss contributes to p53 activation.

Results text:

TDP-43 pathology contributes to the DNA damage response

Although TDP-43 pathology (characterised by neuronal nuclear depletion and cytoplasmic accumulation) is observed in 97% of ALS, it is absent in SOD1 and FUS mutant cases (termed non-TDP-43 ALS) ^{24,25}. Interestingly, SOD1 and FUS mutants displayed the weakest p53 upregulation of the genetic subtypes in both iPSMNs and post-mortem. To identify the degree to which genotypes linked to TDP-43 pathology contribute to p53 upregulation in the pan-ALS analyses, we classified ALS samples based on whether their genetic background is associated with TDP-43 pathology. In iPSMNs, whilst non-TDP-43 ALS (SOD1 and FUS mutant) iPSMNs showed only a modest, non-significant increase in p53 (NES = +2.0, $p = 0.25$), TDP-43 ALS iPSMNs exhibited strong and significant p53 upregulation (p53 NES = +14.2, $p < 0.001$). Likewise, in non-TDP-43 ALS iPSMNs the TP53 TF was mildly decreased in activity (NES = -1.6, $p = 0.12$), whereas TDP-43 ALS iPSMNs showed TP53 was the most strongly upregulated TF (NES + 7.4, $p < 0.001$; Extended Data Fig. 15a,b). We found a similar pattern in post-mortem samples, with non-TDP-43 ALS showing smaller increases in p53 signaling and TP53 TF activity (p53: NES +3.6, $p = 0.03$; TP53: NES +3.3, $p = 0.01$) as compared to TDP-43 ALS (p53: NES +5.2, $p < 0.001$; TP53: NES +2.0, $p = 0.5$; Extended Data Fig. 15c,d). These findings suggest that the p53 signature from the pan-ALS analysis is largely driven by genetic backgrounds associated with TDP-43 proteinopathy.

To discover whether p53 signalling changes are regulated by TDP-43, we next examined RNA-seq from FACS sorted neuronal nuclei into those with and without TDP-43 pathology from FTD-ALS postmortem brain tissue ⁶. We found that neuronal nuclei depleted of TDP-43 showed significant upregulation of p53 signalling (NES +0.4, $p = 0.02$) and non-significant upregulation of TP53 TF as compared to neuronal nuclei retaining TDP-43 (NES +0.7, $p = 0.26$; Extended Data Fig. 15e,f).

To determine whether TDP-43 depletion directly promotes p53 activation, we integrated seven RNA-seq datasets from human cells that have undergone TDP-43 knockdown with shRNA, siRNA, or CRISPR/Cas9 (Table S8) ⁷⁻¹³. In support of a direct role for TDP-43 regulation of p53 signalling, we discovered significant upregulation of both p53 signalling (NES +5.5, $p = 0.02$) and TP53 TF activity (NES +3.5, $p = 0.02$) upon TDP-43 knockdown (Extended Data Fig. 15e,f).

Extended Data Figure 15 TDP-43 loss of function contributes to p53 signaling activation

PROGENy signalling pathway barcharts (left) and DoRothEA transcription factor activities volcano plot (right) in (a-b) non-TDP-43 ALS (i.e. SOD1 and FUS mutant) and TDP-43 ALS iPSMNs; (c-d) non-TDP-43 ALS (i.e. SOD1 and FUS mutant) and TDP-43 ALS post-mortem; (e-f) FACS sorted neuronal nuclei depleted of TDP-43 and TDP-43 knockdown cell models.

Discussion text:

Although we found p53 activation in ALS cases that lack TDP-43 pathology (FUS and SOD1 mutants), the magnitude of activation was substantially weaker than in TDP-43 ALS cases, raising the possibility that TDP-43 contributes to the DNA damage response. In support of this, we found significant p53 upregulation in neuronal nuclei depleted of TDP-43, as well as TDP-43 knockdown models. TDP-43 has an established role in DNA repair and TDP-43 depletion results in the

accumulation of DNA damage^{23,26}. Conversely, TDP-43 overexpression in iPSC-derived neurons led to a pro-apoptotic phenotype, which was partially rescued by p53 inhibition^{27,28}. Other studies of p53 inhibition in TDP-43^{A315T} iPSC-derived neurons has also demonstrated a partial phenotypic rescue^{27,29}. Together, this suggests that TDP-43 pathology exacerbates the DNA damage response, and the subsequent p53 activation may promote motor neuron death in ALS.

- New discussion text clarifying how p53 activation lies downstream of somatic mutations and gene fusions and how p53 activation may lead to motor neuron death:

The increased somatic mutational burden amongst ALS iPSMNs and post-mortem tissue supports the possibility that p53 activation is a reactive change to genomic instability. This begs the question of what drives genomic damage in ALS. Although many of the genetic causes of ALS are linked to defective DNA repair (*FUS*, *TARDBP*, *SOD1*, *C9orf72*, *NEK1*, *SETX*, *VCP*), the mechanism of genome instability in sporadic patients remains enigmatic³⁰. In sporadic cases, DNA damage is possibly a consequence of other ALS pathogenic mechanisms such as mitochondrial dysfunction, autophagy, and TDP-43 mislocalisation³¹.

- We have further examined the p53 and DNA damage pathway genes by performing GSEA on all their daughter Gene Ontology processes - please see our response to Reviewer 3, Comment 3.

Comment 3

The functional enrichment analysis of gene expression changes is key to the implication of the p53 signalling and the DNA damage response which is the main message of the manuscript. It is not clear that a multiple testing correction applied to the enrichment analysis which is crucial.

Author Response

- The reported p-value is from the hypergeometric test after correction for multiple testing, which is performed by the gprofiler2 gost function using Bonferroni correction.

Author Changes

- We have made clearer in the methods text the multiple testing correction:

g:Profiler2 reports the hypergeometric test p-value with an adjustment for multiple testing using the Bonferoni correction.

- In all the functional over-representation bar charts, on the x-axis label, we change the p-value to false discovery rate (FDR) e.g.

Comment 4

The use of statistical tests should be consistent; the NES test is used most often, could this be reported throughout? Also, please include the test statistics for each result referring to DNA damage/p53-signalling in the main text, rather than only referring to extended data.

Author Response

- We thank the reviewer for highlighting this. Whilst the normalised enrichment score (NES) are reported for all gene set enrichment analyses (GSEA) results, for all over-representation analyses we report the adjusted p-value from the hypergeometric distribution. Whilst this also gives other metrics of over-representation (e.g. precision, recall), it does not provide a NES.

Author Changes

- We use gprofiler2 to perform the over-representation analyses, which invokes the hypergeometric test. To make this clearer we change the use of “enrichment” to “over-representation”.
- We have ensured that the hypergeometric adjusted p-value is reported in the main text when referring to functional over-represented terms and have explicitly stated FDR rather than p-value, for example:

Using functional **over-representation** analysis, we found that upregulated genes in ALS were enriched in the DNA damage response (hypergeometric **FDR** = 2.2×10^{-5} ;

SESNI, *RRM2B*, *TNFRSF10B*) and p53 signalling (FDR = 2.7×10^{-5} ; *CDKN1A*, *TP53TG3E*, *FBXO22*) whereas downregulated genes were overrepresented by DNA-binding transcription factor activity (FDR = 0.003; *MYOG*, *TBX5*, *POU5F1* [*Oct4*]) and ventral spinal cord development (FDR = 0.004; *LMO4*, *OLIG2*, *FOXP4*; Fig. 2b).

Comment 5

The perturbation PROGENy analysis is in effect a further enrichment analysis using a database of perturbation experiments which are not MN specific. These limitations should be described.

Author Changes

- We have now noted this limitation in the text:

PROGENy and DoRothEA weights are based on perturbation experiments that are not specific to motor neurons. Their signalling pathways may activate diverse downstream gene expression programmes depending on the cell type and perturbing agent utilised³².

Comment 6

Post mortem transcriptome analysis identifies >10,000 differentially expressed genes which seems excessive (compared to ~20) in the iPSN analyses and makes the likelihood of false positive results more likely. Can the authors comment on this?

Author Changes

- We thank the reviewer for raising this interesting point. We have now commented on the difference in the number of differentially expressed genes between iPSMNs and postmortem:

Despite the large sample size, we found relatively few differentially expressed genes across the ALS spectrum in iPSMNs, consistent with reports that the iPSMN model shows only mild differential gene expression signatures³³. However, this likely also reflects the heterogeneity between ALS genetic backgrounds and the rather conservative strategy utilised. Even within genetic subgroups, there is substantial heterogeneity and none more so than in the sporadic subgroup, which despite being the most well-powered, representing two-thirds of all ALS iPSMNs, had only 4 differentially expressed genes, reminiscent of what we found in sporadic ALS iPSC-

derived astrocytes¹⁸. Sporadic ALS represents a heterogeneous group **with diverse genetic susceptibilities** and also likely includes patients carrying as yet unknown pathogenic gene mutations³⁴. Additionally, environmental risk factors play an important role in ALS aetiology particularly in patients without a highly penetrant mutation^{35–39}. Whilst the iPSC model is an elegant approach to model sporadic ALS, a notable limitation is that it does not reproduce the patients' environmental exposures. **In contrast to iPSMNs, post-mortem spinal cord exhibited thousands of ALS differentially expressed genes and ALS subgroups showed strongly correlated gene expression changes. This supports previous ALS post-mortem studies reporting widespread reactive gliosis and extensive neuroinflammation^{7,40–42}. However, spatial and single-cell transcriptomics as well as cell-type deconvolution analyses on post-mortem samples have revealed that the motor neuron signal is largely masked by glial cell types^{40,43–47}.**

- Results text comparing postmortem genetic subgroups:

In contrast to iPSMNs, comparing genetic subgroups from post-mortem spinal cord tissue revealed highly correlated gene expression changes (R range +0.68 to +0.9) with 1,750 overlapping differentially expressed genes between sporadic, C9orf72, SOD1, and FUS subgroups (Fig. 4h, Extended Data Fig. 14a-j).

Comment 7

Here [in postmortem] p53-signalling gene expression changes are reported for genetic subgroups but it should be made clear that the FUS result is not statistically significant.

Author Response

- We thank the reviewer for this point that FUS mutants did not show significant p53 upregulation in post-mortem. Indeed, in both iPSMNs and post-mortem tissue, SOD1 and FUS mutants showed weaker p53 activation than the TDP-43 ALS subgroups.

Author Changes

- We have made clear that SOD1 and FUS are not significant for postmortem p53 signalling activation:

Examining signalling pathways and transcription factors in each genetic subgroup revealed that p53 signalling and TP53 TF activity were **significantly** increased in both in sporadic (n = 161; p53: NES +5.0, p < 0.001; TP53 NES +2.9, p = 0.05) and C9orf72 (n = 36; NES +5.3, p < 0.001; TP53 NES +4.7, p < 0.001) and non

significantly increased in *SOD1* (n = 5; NES +3.3, p = 0.05; TP53 NES +1.2, p = 0.1) and *FUS* (n = 2; NES +2.9, p = 0.28; TP53 NES +2.9, p = 0.29; Fig. 4g-h, Extended Data Fig. 14l,m).

- We have now added a new section comparing TDP-43 ALS with non-TDP-43 ALS - please see our response to Reviewer 1, Comment 2.

Comment 8

The splicing analysis is misleading. It appears that p53-signalling/DNA damage genes were not statistically enriched within differentially spliced genes. This should be made clear.

Author Changes

- We have now added a one-sided Fisher exact test overlapping differentially spliced genes with p53 genes and confirmed a significant overlap:

Amongst the genes exhibiting differential splicing in ALS were a significant number of genes involved with p53 and DNA repair (including *POLM*, *METTL22*, *HUWE1*, *HDAC1*, *MTA1*, *PMS1*, *ZSWIM7*; Fisher exact test p = 2.3×10^{-8}). Likewise, there was a greater number of RBPs amongst differentially spliced genes than expected by chance (e.g. *YTHDC2*, *THOC1*, *PRR3*, *STAU2*, *PTBP3*, *SREK1*, *POLDIP3*; p = 6.8×10^{-18} ; Fig. 5c).

- We also add a new analysis of splicing in TDP-43 depleted cells (neuronal nuclei and TDP-43 knockdown) and find overlapping events with ALS iPSMNs.

By examining splicing changes in neuronal nuclei depleted of TDP-43⁶, we found 12 overlapping differentially spliced genes (encompassing 17 splicing events) with ALS iPSMN (including *POLDIP3*, *PPP6R3*, *CAMK2B*, *CEP290*; Table S8)⁶. Similarly, comparing splicing changes upon TDP-43 knockdown with ALS iPSMNs revealed 4 overlapping genes containing 6 splicing events (*POLDIP3*, *CAMK2B*, *HERC2P3*, *CEP290*; Table S8). Interestingly, the multi-exon skipping splicing event in *POLDIP3* was precisely the same event that occurs in both TDP-43 neuronal nuclei depletion and TDP-43 knockdown (Extended Data Fig. 15a-c)^{9,48}. This indicates that TDP-43 nuclear loss of function may contribute to splicing changes in ALS iPSMNs.

Comment 9

The authors postulate that intron retention events may be a cause of DNA damage but not provide evidence for this.

Author Response

- We thank the reviewer for this comment. In this integrative analysis of 429 iPSMNs, we find hundreds of splicing changes and discuss the connection between DNA damage and splicing. Whilst we appreciate we have not shown causality between splicing changes and DNA damage, there already exists extensive evidence in the literature that splicing defects, particularly intron retention, cause DNA damage through RNA:DNA cotranscriptional hybrids (R-loops), even within iPSMNs^{49–53}.
- To show convincing evidence that intron retention causes DNA damage in ALS, we would require well-powered numbers of samples of iPSMNs and postmortem spanning each of the ALS subgroups, using a novel approach to assess R loops, such as DRIP-seq^{49,54}. Therefore, it is beyond the scope of this integrative bioinformatics study to examine the causation between intron retention per se and DNA damage in ALS.

Author Changes

- We have now removed the suggestion that intron retention may cause DNA damage.

Comment 10

The authors measure numbers of SNVs and gene fusion events within expressed RNAs in AnswerALS iPSN and use an excessive number of variants to argue for excessive DNA damage in ALS. In my opinion the logic here is flawed; DNA damage causing sequence changes would be expected to produce somatic heterogeneity within the CNS. It is difficult to interpret these changes in neurons derived from peripheral cells.

Author Changes

- We thank the reviewer for raising this. To address this we have now examined somatic mutations and gene fusions in ALS postmortem spinal cord.

As iPSMNs are derived from peripheral cell types and the reprogramming process itself can induce somatic mutations⁵⁵, genome instability in iPSMNs may not be representative of cell types in the central nervous system. To address this, we next examined ALS post-mortem spinal cord tissue for genome instability using the GATK variant discovery pipeline. After removing RNA editing sites and adjusting for sequencing depth, sequencing instrument, and age at death, we found a

significantly greater number of somatic mutations in ALS compared to control post-mortem spinal cord ($p = 4.6 \times 10^{-11}$; Fig. 6c). Examining the variant types revealed that although there were less SNVs in ALS compared to control post-mortem ($p = 0.007$), there were significantly greater numbers of insertions ($p < 2 \times 10^{-16}$) and deletions ($p = 3.8 \times 10^{-7}$) in ALS (Table S11). Examining base substitutions in post-mortem tissue revealed significant increases in the number of T>A ($p < 3 \times 10^{-5}$) and T>C ($p < 2 \times 10^{-16}$) base substitutions in ALS relative to controls (Extended Data Fig. 18b). By assessing each genetic background independently, we found that for sporadic and C9orf72 subgroups there were significant increases in insertions (both $p < 2 \times 10^{-16}$) and deletions (sporadic $p = 3.2 \times 10^{-6}$, C9orf72 $p = 0.036$), however SOD1 mutants showed significant increases in SNVs ($p < 2 \times 10^{-16}$) as well as insertions ($p = 1.1 \times 10^{-12}$), and deletions ($p = 1.1 \times 10^{-9}$; Fig. 6d). These findings implicate a role for de novo mutations and defects in DNA damage repair in ALS.

To identify whether ALS post-mortem tissue also displays increased gene fusions, we performed fusion discovery on post-mortem spinal cord. There were a total of 177 unique gene fusions in ALS and 96 in control post-mortem, with 71 shared. Comparing unique gene fusions between iPSMNs and post-mortem revealed 55 in ALS iPSMNs and post-mortem, of which 27 were also present in CTRLs (Extended Data Fig. 178c). Burden analysis identified 13 gene fusions with a significantly greater burden in ALS post-mortem tissue (Table S12). As with iPSMNs, these mostly involved lncRNAs and the fusion with the greatest burden in ALS post-mortem was a gene neighbour fusion between the lncRNA AL353138.1 and PTCHD4 (OR 4.2, 95% CI 2.3-Inf, Fisher's $p = 1.1 \times 10^{-5}$). After adjusting for read coverage, sequencing instrument, and age at death in a generalised linear model, we identified significantly greater numbers of gene fusions in ALS compared to controls ($p = 7.2 \times 10^{-6}$; Fig. 6g). Comparing the frequencies of each type of gene fusion between ALS with control post-mortem, revealed significantly increased numbers in ALS for gene neighbours ($p = 1.4 \times 10^{-4}$) and local rearrangements ($p = 0.02$), as well as nonsignificant increases in overlapping neighbours ($p = 0.6$), distant intra-chromosomal ($p = 0.12$), and inter-chromosomal fusions ($p = 0.8$; Extended Data Fig. 17e). Comparing the number of gene fusions per post-mortem sample in each ALS genetic group with controls, revealed significantly greater numbers of gene fusions in sporadic ALS ($p = 8.2 \times 10^{-6}$), C9orf72 ($p = 0.03$) and SOD1 mutant subgroups ($p = 0.0009$; Fig. 6h).

Fig. 6: ALS iPSMNs and post-mortem tissue accumulate somatic mutations and gene fusions.

a, Violin plots showing the numbers of somatic mutations, adjusted for age and read depth, identified in Answer ALS iPSMNs in ALS (red) and CTRL (blue) samples for all mutation types, single-nucleotide Variant (SNV), Insertions, and Deletions. **b**, Forest plot showing the point estimate and 95% confidence interval of changes in mutation types (SNV, blue; Insertion, red; Deletion, green) in ALS genetic subgroups versus controls. Vertical dashed line indicates no difference, to the right of the dashed line indicates increase in ALS. **c-d**, As for a-b except in post-mortem spinal cord. **e**, Violin plots showing the adjusted number of gene fusions in CTRL (blue) and ALS (red) in iPSMNs. **f**, Forest plot showing the point estimate and 95% confidence interval changes in each genetic subtype versus controls. To the right of the dashed line indicates an increase in ALS subtypes. **g-h**, As for e-f except in post-mortem. **** represents Wald test $P < 0.0001$, *** $P < 0.001$, ** $P < 0.01$, * $P < 0.05$, after adjusting for read coverage, age, and dataset covariates.

Extended Data Figure 18 Somatic mutations and fusion types in genetic subgroups

a-b, Adjusted number of SNVs classified according to their base substitution type in ALS (red) and CTRL (blue) in (a) iPSMNs and (b) postmortem. c, UpSet plot depicting overlapping unique gene fusions in iPSMNs and postmortem ALS and CTRL samples. d-e, Adjusted numbers of each type of gene fusion events per sample in ALS (red) and CTRL (blue) samples in (d) iPSMNs and (e) postmortem. A generalised linear model with Poisson distribution was fit to compare ALS with control, adjusting for coverage, age, and dataset. Plotted values represent the partial residuals after adjusting for dataset batches, read depth and age and statistics are from the generalised linear model Wald test accounting for dataset batches, age, and read coverage. **** $p < 0.0001$, *** $p < 0.001$, ** $p < 0.01$, * $p < 0.05$.

Methods text:

For post-mortem tissue, the spinal cord samples from the NYGC dataset were included, although within NYGC there exists two sequencing batches (NovoSeq 6000 with 200 bp read lengths [n=186]; HiSeq 2500 with 250 bp reads [n=85]).

- The numbers of SNVs was recently reported in Nature in neurons from individuals with Alzheimer's Disease⁵⁶. In Fig 1d-h, age was shown to influence the number of variants and so we have now adjusted for age in both the variant and fusion generalised linear models and for both iPSMNs and postmortem:

After adjusting for sequencing depth and donor age, across all filtered variant types, we found significantly greater numbers of somatic mutations per iPSMN in ALS compared to control (Wald test $p < 2 \times 10^{-16}$; Fig. 6a).

Since gene fusion discovery is also sensitive to differences in read coverage and age, we adjusted for the read coverage, the donor age, and dataset batch effects in a generalised linear model.

To compare the number of somatic mutations per sample in ALS versus CTRL groups, a generalised linear model was fit specifying a Poisson distribution adjusting for differences in read coverage per sample and donor age using a spline: $variant\ count \sim condition + rcs(read\ depth, 3) + rcs(age, 3)$. For post-mortem tissue we included an additional term to adjust for the sequencing batch: $variant\ count \sim condition + batch + rcs(read\ depth, 3) + rcs(age, 3)$.

To compare the number of fusions per sample in ALS versus CTRL groups, a generalised linear model was fit specifying a Poisson distribution adjusting for differences in read coverage per sample and donor age and batch effects between

datasets: $\text{fusion count} \sim \text{condition} + \text{dataset} + \text{rcs}(\text{read depth}, 3) + \text{rcs}(\text{age}, 3)$. For post-mortem tissue we included an additional term to adjust for the sequencing batch: $\text{fusion count} \sim \text{condition} + \text{batch} + \text{rcs}(\text{read depth}, 3) + \text{rcs}(\text{age}, 3)$.

- For the violin plots, instead of displaying the raw observed number of variants and fusion (which can be misleading), we have now plotted the normalised counts for read depth and age using partial residuals as per the jtools vignette. The partial residuals allow us to remove the variation due to read depth and age, but leave in the residual variation due to ALS vs control, thus offering the best visual representation of the modelled data.

In the violin plots, the observed number of variants and fusions were adjusted for read depth and donor age by using the partial residuals from the generalised linear model, which was performed using the `partialize` function from the `jtools` package.

- For genetic subgroups, we utilise forest plots in Figure 6 to depict the results from the generalised linear models, showing the point estimate and 95% confidence interval.

For genetic subgroups, forest plots were generated using the `plot_summs` function, which depict the regression coefficient results from the generalised linear model showing the point estimate and 95% confidence interval.

Comment 11

Was the meta-analysis applied to the analysis of specific genetic-subtypes or was the RUVg used to correct for batch effects in these analyses? This is not clear in the methods.

Author Response

- We thank the reviewer for raising this. RUVg was not used to correct for batch effects, only to assess for batch effects. For the genetic subtype analyses, as with the pan-ALS integrated analysis, we combined all relevant samples for each genetic background and accounted for dataset batch effects as a covariate in the linear model. Batch effects were then corrected for using the DESeq2 design $\sim \text{gender} + \text{dataset} + \text{genetic_group}$.

Author Changes

- We have made clearer that we used the the DESeq2 design, rather than RUVg in the genetic subgroups

We orthogonally estimated technical variation using the RUVg method that takes empirically defined negative control genes to estimate low-rank technical variation

in the data, specifying 5 RUV factors⁵⁷. To examine the effect of each ALS genetic background on gene expression ~~a similar approach was used, we compared~~ ALS versus control samples using the DESeq2 design formula $\sim gender + dataset + genetic_group$. For these genetic subgroup analyses, control samples from each dataset were only utilised if the dataset also exhibited the relevant ALS genetic background.

- Describing this study as a meta-analysis is partly misleading and was a mistake on our part. Whilst a meta-analysis does combine multiple studies, it is a synthesis of their pre-analysed results. With respect to differential gene expression, meta-analysis typically refers to combining each individual study gene p-values, for example using the metaRNAseq package. However, the approach we have taken of integrating the raw sequencing data from the component datasets, and calculating gene expression changes based on each sample gene counts is substantially more robust at addressing batch effects between datasets. To make this clearer we now replace the use of “meta-analysis” with “integrative analysis”.

Reviewer 2

Comment 1

The manuscript by Ziff et al is a large scale meta-analysis of multiple publically available RNAseq datasets for ALS. This includes data from Answer ALS and NeuroLINCS as well as data from an ALS postmortem spinal cord cohort. The authors find that p53 activation is observed in multiple subgroups of ALS and that DNA damage responses are also enriched as key mechanisms in both familial and sporadic ALS, beginning early and persisting into later stages of disease. The authors also highlight genomic instability via splicing alterations, single nucleotide variants, insertions, deletions and gene fusions. There is a good rationale and a lot of data used, which is a major positive.

- We thank the reviewer for these comments.

Comment 2

While the analysis is reasonable and informative, this manuscript has significant weaknesses and novelty to warrant publication in Nat Comm. First, the work is not entirely novel given the previous publication in Cell (2021) that p53 is central to c9 ALS.

Author Response

- We thank the reviewer for mentioning this publication which was also mentioned by Reviewer 1, Comment 1. Whilst we agree that p53 activation has been previously described

in ALS, these have been limited to mutant forms of ALS predominantly in C9orf72¹⁻⁵. Previous reports have also been restricted in sample size usually < 10 patients in total. Conversely, this is the **first study with sufficient power** to demonstrate genome instability **across ALS subtypes in both iPSMNs and post-mortem**. This is especially noteworthy in **sporadic ALS**, which accounts for **90%** of cases but has the least prior evidence for increased p53^{19,20}. The scale of this resource also enables us to **compare the magnitude of p53 activation between ALS subgroups** for the first time, exposing that TDP-43 pathology cases show substantially greater p53 activation than non-TDP-43 ALS cases - see response to Reviewer 3, Comment 6. Furthermore, this is the first study to show **increased somatic mutations and gene fusions** in both ALS iPSMNs and post-mortem.

- Whilst the paper from the Gitler group¹ is an excellent paper, there are important differences with ours. Most notably, they predominantly used mouse neurons and as noted by the reviewer it was restricted to C9orf72 poly(PR) treatment. It did not examine other ALS subgroups. Other ALS iPSMN studies examining DNA damage have been restricted to very small sample sizes and mutant forms of ALS, e.g. Lopez-Gonzalez et al 2016 studied only 3 C9orf72 mutant iPSMNs³. Likewise, post-mortem studies reporting p53 upregulation have been limited to only a handful of patients, e.g. Farg et al 2016 in 10 C9orf72 patients².
- The novelty of our **resource** is that it provides the largest catalogue of human ALS associated gene alterations to date, obtained by characterising 429 human ALS iPSMN and 271 post-mortem transcriptomes. The unbiased genome-wide approach used reveals **multiple novel targets in ALS** beyond p53 e.g. RNase L expression, MAPK signalling, CNOT3 splicing. It also provides the **largest study of splicing changes in ALS iPSMNs**, not only revealing TDP-43 regulated events (e.g. POLDIP3, CAMK2B, CEP290) but also revealing new splicing events shared between ALS subgroups (e.g. CNOT3, SRSF10, DNAJC17). Thus, this paper offers substantial advances in the understanding of changes across ALS subtypes, which is crucial for identifying therapeutic targets.

Author Changes

- We add discussion of other previous DNA damage papers in ALS, and clarify the novelty in this study:

Our findings support previous, smaller studies showing p53 activation in ALS, particularly with *C9orf72* repeat expansions^{1-5,20-23}. However, the novelty in this study is the finding of p53 activation, together with increased somatic mutations and gene fusions, across diverse ALS subgroups, not only in iPSMNs but also in post-mortem spinal cord tissue. We identified the greatest increase in p53 activity in *C9orf72* mutants, consistent with reports that the *C9orf72* repeat expansion induces DNA damage, likely mediated by dipeptide repeat proteins and the formation of R loops and G quadruplexes¹⁻⁵. However, we also found p53 to be strongly and significantly activated in *TARDBP* and sporadic subgroups. This finding in sporadic

cases is particularly striking since they represent 90% of ALS but have the least prior evidence for p53 activation.

Comment 3

Second, there would need to have some additional validation of the DNA damage deficits in iPSCs and/or ALS tissues, validate protein levels for DEGs (especially for p53 and DNA damage genes), and validate R loop formation.

Author Response

- We thank the reviewer for raising these helpful suggestions. In this integrative analysis of 400+ iPSMNs, we find hundreds of splicing changes and discuss the connection between DNA damage and splicing. Whilst we appreciate we have not shown direct causality between splicing changes and DNA damage, there already exists extensive evidence in the literature that splicing defects, particularly intron retention, cause DNA damage through RNA:DNA cotranscriptional hybrids (R-loops), even within iPSMNs^{49–53}.
- To show convincing evidence that intron retention causes DNA damage in ALS, we would require well-powered numbers of samples of iPSMNs and postmortem spanning each of the ALS subgroups, using a novel approach to assess R loops, such as DRIP-seq^{49,54}. Therefore, it is beyond the scope of this integrative bioinformatics study to examine the causation between intron retention and DNA damage in ALS.

Author Changes

- The **postmortem spinal cord dataset** has been updated to include the most recent releases from NYGC and Target ALS. This brings the total number of postmortem spinal cord samples to 271 (ALS n = 214, CTRL n = 57), up from 203 previously (ALS 153, CTRL 50). This additional data has boosted p53 activation, notably in sporadic ALS, and the postmortem results are now included in a new main figure. We have also added a new somatic mutation and gene fusion analysis in postmortem spinal cord, demonstrating that ALS samples show greater numbers of insertion and deletion mutation and gene fusions.

Fig. 4: Postmortem spinal cord shows p53 activation

a, Volcano plot of differential gene expression in ALS versus control postmortem spinal cord. **b**, Functionally enriched terms in up-regulated (red) and down-regulated (blue) differentially expressed genes. **c**, GSEA for signal transduction by p53 in ALS versus control postmortem spinal cord. NES, normalized enrichment score. **d**, PROGENy signalling pathway activities in ALS versus control postmortem tissue. Pathways increased in ALS are red and pathways decreased are blue. *** represents $P < 0.0001$ and * $P < 0.05$. **e**, Expression changes of p53 signalling pathway genes in ALS versus control according to their PROGENy weights. Genes in ALS increasing p53 activity are red and genes decreasing p53 activity are blue. **f**, Activities of 429 transcription factors in DoRothEA inferred from their regulon expression changes in ALS versus control postmortem tissue. **g**, Scatterplot of ALS vs control gene expression changes in iPSMNs (x-axis) against postmortem tissue (y-axis). **h**, PROGENy p53 signalling pathway (left) and DoRothEA TP53 transcription factor regulon activity (right) amongst each of the genetic backgrounds in postmortem tissue. **i**, Heatmap of Pearson correlation coefficients for transcriptome-wide changes between iPSMNs (columns) and postmortem (rows) for each genetic background.

- To validate the existence of genomic instability in ALS CNS tissue, we have now examined somatic mutations and gene fusions in postmortem samples - please see our response to Reviewer 1, Comment 10.
- We have now added a proteome-wide analysis using the mass spectrometry of AnswerALS iPSMNs to the results:

Results:

To identify how gene expression changes relate to protein expression changes in ALS iPSMNs we interrogated mass spectrometry data in AnswerALS, which includes 204 iPSMNs (ALS n = 171, Controls n = 33). Across the whole proteome, no proteins were significant at $FDR < 0.05$ in ALS compared to control, however, 276 were significantly different at unadjusted $P < 0.05$ (Extended Data Fig. 11a; Table S5). Amongst these were 46 RBPs (e.g. RRBP1, RPS29, EIF1, UBA1), amyloid precursor protein (APP) as well as the p53 pathway components RBBP7, CSNK2B, PRMT1, CNOT9 - the latter of which were all increased in ALS iPSMNs. Functional over-representation analysis revealed that proteins increased in ALS were enriched in protein metabolism (e.g. PSDMD9, NAE1, PSMB5), RNA metabolism (e.g. APP, CSTF1, CIRBP, SNRPD2), protein binding (e.g. RRBP1, RPS29, EIF1) as well as other processes established in ALS pathophysiology (stress response, cholesterol synthesis, nucleocytoplasmic transport) whilst proteins decreased in ALS were enriched in golgi transport (Extended Data Fig. 11b). Enrichment analysis of the p53 pathway protein set revealed this to be increased in ALS iPSMNs, albeit nonsignificantly (NES +0.98, $p = 0.5$; Extended Data Fig. 11c). Comparing changes in mRNA with protein expression revealed overlapping differentially expressed genes and a weak inverse correlation ($R = -0.13$; Extended Data Fig. 11d), consistent with previous reports that mRNA and protein correlations are poor^{58,59}.

Extended Data Figure 11 Mass spectrometry of iPSMNs in AnswerALS

a: Volcano plot showing differential protein expression changes in ALS versus control iPSMNs from Answer ALS mass spectrometry data. There were no significant proteins at $FDR < 0.05$ and a more lenient threshold of $P \text{ value} < 0.05$ is used to colour significantly changed genes.

b: Gene Ontology terms enriched in up-regulated (red) and down-regulated (blue) differentially expressed proteins in ALS versus control iPSMNs.

c: Protein set enrichment analysis of signal transduction by p53 (GO:0072331, $n = 103$) in ALS versus control. NES, normalized enrichment score.

d: Scatterplot of gene expression changes (test statistic; x-axis) against protein expression changes (y-axis) in ALS versus control iPSMNs. Overlapping differentially genes/proteins are coloured red. The solid blue line represents the linear correlation and Pearson correlation $R = -0.07$.

Methods text:

For analysis of protein changes in iPSMNs, processed proteomic data (.csv matrix of peptide intensities) was downloaded from the AnswerALS data portal. Mass spectrometry was performed and processed as reported in AnswerALS⁶⁰. Peptide intensities were processed following the LIMMA Peptide tutorial instructions using log2 transformation and specifying a contrast of ALS versus healthy control iPSMNs⁶¹. Quality control of peptide intensities across samples before and after log2 transformation was performed. Outlier samples were identified using sample-sample distances with the MDS plot and outlier peptides were detected using the mean-variance relationship. As no differentially expressed proteins were identified using $FDR < 0.05$, to identify more subtle changes in ALS versus control a more lenient threshold of p-value < 0.05 was utilised.

- As we do not validate that R loop formation causes DNA damage in ALS, we have now removed the suggestion that intron retention may cause DNA damage.

Comment 4

There is an overall lack of mechanism discussed in the paper in that p53 is more of a consequence versus mechanism. Should discuss how p53 activation and DDR would impact the neurons or disease.

Author Changes

- We thank the reviewer for this suggestion. To examine for a **functional link between TDP-43 nuclear depletion and p53 activation**, we have now examined RNA-seq from neuronal nuclei depleted of TDP-43 from patients' brains⁶ as well as integrating 7 RNA-seq datasets from human cultured cells that have undergone TDP-43 knockdown with shRNA, siRNA, or CRISPR inhibition⁷⁻¹³. Both neuronal nuclei depleted of TDP-43 and cells undergoing TDP-43 knockdown show significant upregulation of p53 signaling, indicating that TDP-43 nuclear loss contributes to p53 activation. Please see our response to Reviewer 1, Comment 2.
- We have added a discussion of how somatic mutations and gene fusions may contribute to p53 activation as well as p53 activation could play a role in ALS and in motor neuron death:

Although we found p53 activation in non-TDP-43 ALS (FUS and SOD1 mutants), the magnitude of activation was substantially weaker than in TDP-43 ALS cases, raising the possibility that TDP-43 contributes to the DNA damage response. In support of this, we found significant p53 upregulation in neuronal nuclei depleted of TDP-43, as well as TDP-43 knockdown models. TDP-43 has an established role in

DNA repair and TDP-43 depletion results in the accumulation of DNA damage^{23,26}. Conversely, TDP-43 overexpression in iPSC-derived neurons led to a pro-apoptotic phenotype, which was partially rescued by p53 inhibition^{27,28}. Other studies of p53 inhibition in TDP-43^{A315T} iPSC-derived neurons have also demonstrated a partial phenotypic rescue^{27,29}. Together, this suggests that TDP-43 pathology contributes to heightened DNA damage, and the subsequent p53 activation may promote motor neuron death in ALS.

The observed relative hypermutation rate in ALS iPSMNs and post-mortem tissue supports the possibility that p53 activation is a reactive change to genomic instability. This begs the question of what drives genomic damage in ALS. Although many of the genetic causes of ALS are linked to defective DNA repair (*FUS*, *TARDBP*, *SOD1*, *C9orf72*, *NEK1*, *SETX*, *VCP*), the mechanism of genome instability in sporadic patients remains enigmatic³⁰. In sporadic cases, DNA damage is possibly a consequence of other ALS pathogenic mechanisms such as mitochondrial dysfunction, autophagy, accelerated aging, and TDP-43 mislocalisation³¹.

Comment 5

The authors cite that OCT4 was significantly downregulated. Could this be an artifact of the iPSC modeling? Undifferentiated iPSCs? Or persistent OCT4 expression. Would need to discuss. Upregulation/activation of MAPK and p53, along with OCT4 expression may indicate a proliferating progenitor population. OCT4 also comes up quite significantly in the sALS vs control comparison. Could this set of samples be confounding? The authors should consider accounting for co-variates or filtering out specific markers.

- We thank the reviewer for raising this and for noting that the OCT4 (*POU5F1*) decrease in ALS iPSMNs is also observed independently in the sporadic ALS subgroup comparison. Examining OCT4 normalised gene expression (transcripts per million, TPM) in each sample shows that, whilst the majority of samples have no detectable OCT4 expression, there are a handful of control samples lowly expressing OCT4. Rather than a differentiation issue with the sporadic ALS samples, there are ~20 CTRL samples with weak OCT4 expression driving the differential gene expression. Using the NeuroLINCS undifferentiated iPSCs for comparison, we can see that OCT4 expression in undifferentiated iPSCs is ~50x higher with TPM values 1500 - 2000:

- Reassuringly, all the 28 iPSMNs with > 5 TPM OCT4 expression showed minimal expression of other undifferentiated pluripotency markers (NANOG, ZFP42, ESRG, CNMD, SFRP2). There was mild SOX2 expression, however this is also a neural marker and so its expression in iPSMNs is expected⁶². Conversely, NeuroLINCS undifferentiated iPSCs showed strong expression of all undifferentiated pluripotency markers:

- OCT4 (POU5F1) is a transcription factor and the level of the mRNA encoding a transcription factor (TF) is a relatively crude indicator of its overall activity. To more accurately infer the TF activity we examined the expression of the TFs entire regulon using DoRothEA. OCT4 and NANOG TF regulons exhibited consistently low activity across all iPSMNs (including those with OCT4 mRNA levels >5 TPM). SOX2 was active across iPSMN samples, but as already noted, this is expected as it is also a recognized neural marker. Additionally, the transcription factor FOXP1, which promotes lateral motor column motor neuron identity, was

active in all iPSMNs. Thus, despite the weak OCT4 mRNA expression in a small number of iPSMNs, it is highly unlikely that these represent undifferentiated iPSCs.

- To further confirm that the small number of lowly expressing OCT4 iPSMNs did not have a substantial effect on the results, we performed a sensitivity analysis, removing the 28 iPSMNs with OCT4 expression > 5 TPM. This confirmed that the overall results were largely unaffected:

Author Changes

- To more thoroughly display the differentiation state we have added to Extended Data Fig 5 Heatmap markers of undifferentiated pluripotent stem cells (including NANOG, POU5F1 [OCT4], SOX2, ALPL, ESRG, CNMD, SFRP2), neural precursors, neural progenitors, floor plate, roof plate, neural crest and dorsal progenitors. For comparison, we also add the undifferentiated iPSCs from NeuroLINCS. Reassuringly, this confirms that throughout all iPSMNs, OCT4 (POUF51) as well as the other pluripotent undifferentiated markers are lowly expressed.

Extended Data Figure 5 Motor neuron differentiation state

Heatmap depicting normalised gene counts of **undifferentiated stem cells, neural precursors, neural progenitors, floor plate, roof plate, neural crest and dorsal progenitors, and neuronal markers** (columns) across the iPSC-derived spinal motor neuron datasets (rows). Blue represents low gene expression and red represents high expression. To improve the visualisation of all datasets, for AnswerALS only 100 samples are plotted. Lee dataset is removed from the **integrated** analysis because of inadequate neuronal marker expression. **The**

NeuroLINCS iPSC batch (bottom block) are not included in the integrated analysis and are only included in this heatmap to enable comparison of iPSMNs with undifferentiated pluripotent iPSCs.

Comment 6

Correlation values are very low and it is not clear if these are significant. Perhaps this is indicated in the extended data but should show p-value in text.

- We thank the reviewer for this point and agree that many of the correlations between genetic backgrounds reported are weak (R ranges from 0.01 to 0.39). We believe this is an important finding since it confirms the suspected heterogeneity between genetic subgroups. We should note that when correlating 40,000+ genes correlation p-values are usually tiny and do not reflect the correlation strength, which is depicted in the R value. Indeed, despite weak correlations, many of our transcriptome-wide correlations have $P < 2 \times 10^{-16}$.

Author Changes

- We have now added the p-value to all scatterplots and text throughout the manuscript and extended data, e.g.:

Correlating transcriptome-wide gene expression changes between ALS genetic backgrounds revealed weak associations, with the strongest correlation between *SOD1* and sALS lines (Pearson R = +0.38, $p < 2.2 \times 10^{-16}$) and the weakest between *SOD1* and *TARDBP* (R = -0.14, $p < 2.2 \times 10^{-16}$; Fig. 3f, Extended Data Fig. 10a).

Comment 7

The base sub data is confusing. The authors indicate an increase number of C>T in ALS then repeats this info with a lower pvalue. And then suggest C>T is lower in ALS relative to control. Potentially typos?

- We thank the reviewer for raising this potentially confusing point. Although this is not a typo, we can see what this reviewer means. In Fig 5c we showed the *number* of each type of base substitution in ALS and CTRLs. Whereas in Ext Data Fig 13a we reported the *relative proportion* of each type of base substitution.

Author Changes

- Since the relative proportions of variants add little compared to the total number of variants, and at the cost of confusing the reader by showing both, we have now removed the old Ext Data Fig 13a showing the relative proportions.

Comment 8

Was the analysis for gene fusions events somehow normalized by sample size, considering there were many more ALS samples than control?

- We thank the reviewer for raising this point, which applies to all analyses throughout the study and not just gene fusions. Comparing two groups of unequal sample sizes are accounted for within the statistical test, which incorporates both groups equitably and the estimate of standard errors of the effect size is calculated using all samples irrespective of the condition. Thus, unequal sample sizes between the two groups does not require any further normalisation. For further statistical explanation, please see the nice explanation by Jan VanHove <https://janhove.github.io/design/2015/11/02/unequal-sample-sized>

Comment 9

There is not discussion of sex as a variable

- Thank you. To remove unwanted variation due to sex, we included gender as a covariate in the statistical design, which helps ensure that variation in gene expression due to gender is adjusted for.

Methods text:

Gender was confirmed by examining the expression of the X chromosome gene *XIST* (female) and Y chromosome genes *KDM5D*, *DDX37*, *RP54Y1* and *EIFAY* (male).

The **integrated** analysis results of ALS iPSMNs were generated by comparing the ALS versus control groups using the Wald test, controlling for sex differences and dataset variation with the design formula \sim gender + dataset + condition.

Author Changes

- We have now added to the results text that we adjust for gender.

To identify pan-ALS transcriptomic changes we performed an **integrated** analysis comparing all 323 ALS versus 106 control iPSMNs, accounting for batch effects between datasets **and gender** (see Methods).

Reviewer 3

Comment 1

Ziff et al. combined the huge datasets of ALS, unifying them to investigate the pathological mechanisms of ALS. The research direction of integrating patient samples and data sets to explore the ALS pathology is an interesting approach that should coincide with current trends. The problem with such an approach is the difficulty in integrating various data-sets, obtained by various methods, and with appropriate correction for bias. In addition, this reviewer is also concerned that there is no axis to evaluate the appropriateness for correcting bias, and there are limited clues to justify the results of integrated analysis. In this study, the authors pointed out the activation of p53 in ALS samples from different datasets, and confirmed a similar trend of p53 activation by using autopsy specimens of spinal cords. However, the reviewers and the readership will face a considerable difficulty in judging the scientific validity of results from the integrated data-sets of this paper.

- We thank the reviewer for these comments and for acknowledging the challenges of dataset integration. The decision to combine datasets needs careful assessment of the benefits and risks. Whilst the key risks have been outlined by the reviewer the benefits are twofold:
 1. Increase the power to detect true positive changes and avoid false negative changes (i.e. increased sensitivity).
 2. Increase the variety of cases (and controls) to avoid false positive changes (and detect true negative changes; i.e. increased specificity).
- Because of the cost of generating iPSMN lines, iPSMN studies have been greatly underpowered with many limited to only 2-3 patients, meaning that the sensitivity and specificity of their findings are limited. This is a major problem in ALS research because of the substantial genetic, pathological, and clinical heterogeneity. Combining RNAseq from 300+ ALS and 100+ control iPSMNs addresses this. Hence, the benefits of integrating the datasets largely outweigh the risks, many of which can be minimised with detailed QC and appropriately designed statistical tests. Indeed, we have gone to great depths to attempt to minimise these risks by:
 1. Quality control checks: sequencing metrics (Table S2), unbiased transcriptome-wide PCA clustering (Ext Data Fig 2-3), and cell type identity and differentiation markers (Ext Dat Fig 4-7).
 2. Using a statistical design in generalised linear models that includes covariates to remove unwanted variation due to dataset batch effects, including variations in differentiation protocols, library preparation, and sequencing instrument.
 3. Sensitivity analyses examining potential sources of unwanted variation, including RNA library preparations (total and polyA), datasets independently (see below), and different ALS subgroups.
- There is precedence for integration of next-generation sequencing genome and transcriptome datasets from cancer, where multiple distinct cancer type datasets have been successfully

integrated in pan-cancer analyses, for example, Drews et al Nature 2022¹⁴, Liu et al Nature 2022¹⁵, Litchfield et al Cell 2021¹⁶, PCAWG Group Nature 2020¹⁷. Furthermore, we have previously published this approach of integrating RNA-seq datasets of iPSC-derived astrocytes: Ziff et al. Genome Research 2021¹⁸

Author Changes

- To further address the concern of integrating datasets, we have now added an analysis of each dataset independently.

We further investigated changes between ALS and control samples in each dataset separately. This revealed substantial heterogeneity between datasets in ALS versus control gene expression changes (Extended Data Fig. 10a-b). Despite this, both TP53 transcription factor and p53 signalling were independently upregulated in ALS iPSMNs in 11 of 17 datasets (Extended Data Fig. 10c-d). This indicates that **neither library preparation nor dataset batch effects were** responsible for the DNA damage response gene expression changes observed in ALS iPSMNs.

Extended Data Figure 10 Analysis of each iPSMN dataset separately

a: Volcano plots showing \log_2 fold change in differential gene expression in ALS compared to control iPSMNs in each dataset. Red genes are significantly (FDR < 0.05) increased and blue is decreased in ALS. b: Heatmap showing the Pearson's correlation coefficient for transcriptome-wide changes between each dataset. c p53 signalling pathway activity showing ALS versus control iPSMNs normalised enrichment scores (y-axis) in each dataset. Pathways that are increased in ALS are coloured red whilst pathways decreased are coloured blue. * represents enrichment p-value < 0.05, ** p < 0.01, *** p < 0.001, **** p < 0.0001. d: TP3 transcription factor activity in ALS versus control iPSMNs in each dataset. Normalised enrichment in ALS versus control (x-axis) is plotted for each dataset (y-axis).

Comment 2

In particular, the activation of p53, which the authors have pointed out, is a phenomenon commonly observed in a wide variety of pathological conditions, and is unfortunately not new. Although the meta-analysis with multiple data sets may be novel, the p53 pathway found as a result of this study has already been reported in ALS and is not novel.

- We thank the reviewer for these comments. Whilst we agree that p53 activation has been previously described in ALS, these have been **limited to mutant forms of ALS**, predominantly with *C9orf72* mutations¹⁻⁵. Previous reports have also been **underpowered**, for example, ALS iPSMN studies examining DNA damage have been restricted to very small sample sizes and mutant forms of ALS, e.g. Lopez-Gonzalez et al 2016 studied only 3 *C9orf72* mutant iPSMNs³. Likewise, post-mortem studies reporting p53 upregulation have been limited to only a handful of patients, e.g. Farg et al 2016 in 10 *C9orf72* patients².
- Conversely, this is the first study with sufficient power to demonstrate genome instability across ALS subtypes in both iPSMNs and post-mortem. This is especially noteworthy in sporadic ALS, which accounts for 90% of cases but has the least prior evidence for increased p53^{19,20}. The scale of this resource also enables us to compare the magnitude of p53 activation between ALS subgroups for the first time, exposing that TDP-43 pathology cases show substantially greater p53 activation than non-TDP-43 ALS cases - see response to Reviewer 3, Comment 6. Furthermore, this is the first study to show increased somatic mutations and gene fusions in both ALS iPSMNs and post-mortem.
- The novelty of this resource is that it provides the largest catalogue of human ALS associated gene alterations to date, obtained by characterising 429 human ALS iPSMN and 271 post-mortem transcriptomes. The unbiased genome-wide approach used reveals multiple novel targets in ALS beyond p53 e.g. RNase L expression, MAPK signalling, CNOT3 splicing. It also provides the largest study of splicing changes in ALS iPSMNs, not only revealing TDP-43 regulated events (e.g. *POLDIP3*, *CAMK2B*, *CEP290*) but also revealing new splicing events shared between ALS subgroups (e.g. *CNOT3*, *SRSF10*, *DNAJC17*). Thus, this paper offers substantial advances in the understanding of changes across ALS subtypes, which is crucial for identifying therapeutic targets.

Author Changes

- We discuss other ALS studies of DNA damage, and clarify the novelty regarding DNA damage in this study:

Our findings support previous, smaller studies showing p53 activation in ALS, particularly with *C9orf72* repeat expansions^{1-5,20-23}. However, the novelty in this study is the finding of p53 activation, together with increased somatic mutations and gene fusions, across diverse ALS subgroups, not only in iPSMNs but also in post-mortem spinal cord tissue. We identified the greatest increase in p53 activity in *C9orf72* mutants, consistent with reports that the *C9orf72* repeat expansion induces

DNA damage, likely mediated by dipeptide repeat proteins and the formation of R loops and G quadruplexes¹⁻⁵. However, we also found p53 to be strongly and significantly activated in *TARDBP* and sporadic subgroups. This finding in sporadic cases is particularly striking since they represent 90% of ALS but have the least prior evidence for p53 activation.

Comment 3

This [p53] pathway also includes many genes that tend to be identified as a “statistically significant” pathway. This reviewer believes that more specific gene sets and specific signaling pathways may have hidden molecular clues that will lead to understanding of the pathology and the development of therapeutic methods. Therefore, it would be more important to use the presented methodology for finding a specific pathway such as a screening step, and to verify the identified results by another modality. This reviewer thinks that the presented methodology in the paper is extremely important for the scientific field, and believes that the following points, as described below, need to be revised in order to maximize the value of this paper and ensure that the method will be widely accepted in the future.

- We thank the reviewer for the positive comments and the suggestion to investigate more specific gene sets and use orthogonal modalities.
- We have addressed the point regarding the screening step followed by orthogonal validation. The screening step is the over-representation analysis (ORA) of differentially expressed genes (using gprofiler2 to screen Gene Ontology, KEGG, Reactome, and WikiPathways). We then use Gene Set Enrichment Analysis (GSEA) for validation to identify changes across the entire gene set. Combining ORA with GSEA of functionally enriched pathways is an effective dual-pronged approach to expose individual genes changing where the difference is large (ORA of differentially expressed genes) and sets of related genes that change in a coordinated way but each gene changes by a small amount (GSEA of gene set) - explained in more detail at <https://yulab-smu.top/biomedical-knowledge-mining-book/enrichment-overview.html#gsea-algorithm>. We also further investigate transcription factor and signalling pathway activities using a distinct method with decoupleR (PROGENy and DoRothEA), which allows us to account for the sign and weight of each gene within the network. For example with p53, we expose the individual genes with the greatest positive weighting, driving the p53 term enrichment in ALS - CDKN1A, SESN1, RRM2B:

Author Changes

- To assess more specific gene sets, we have performed a GSEA on the daughter terms of the p53 signalling pathway and DNA damage response:

Since p53 signalling and DNA damage response are large pathways, we next explored more specific gene sets by examining their daughter pathways. This revealed strong upregulation of genes involved with the mitotic G1 DNA damage checkpoint and intrinsic apoptosis signalling (Extended Data Fig. 8).

Extended Data Figure 8 Gene set enrichment analysis of DNA damage daughter terms

Daughter terms of DNA damage pathway upregulated in ALS are red (to the right) and those decreased in ALS iPSMNs are blue (left). **** $p < 0.0001$, *** $p < 0.001$, ** $p < 0.01$, * adjusted $p < 0.05$ from GSEA enrichment test.

- As with iPSMNs, we have added PROGENy weights for each gene in the p53 pathway in ALS postmortem spinal cord (Fig. 4e):

Exploring individual genes in the p53 pathway according to their PROGENy weights revealed that the top genes driving p53 activity in ALS post-mortem include *FAS*, *RRM2B*, *CSTA*, *ZMAT3*, *TP53I3*, and *CDKN1A* (Fig. 4e). *CDKN1A* is a cyclin-dependent kinase inhibitor and its upregulation in both ALS iPSMNs and post-mortem tissue may represent a possible cell cycle block through cyclin-dependent kinase inhibition.

Fig. 4e: Expression changes of p53 signalling pathway genes in postmortem spinal cord ALS versus control according to their PROGENy weights. Genes increasing p53 activity in ALS are red whilst genes decreasing p53 activity in ALS are blue.

Comment 4

The authors combined the deposited datasets from several studies for metaanalysis. However, the differentiation characteristics of iPSC-derived motor neurons, evaluated by the expression of motor neuron markers like *CHAT*, *MNX1*, *ISL1*, *FOXP1*, etc., vary among different studies in extended data figures 4 and 5. The homogeneity of the differentiation quality is the most important factor in the discussion of disease phenotypes. Can the authors correct the gene expression status, or set a stricter threshold for selecting the datasets? These steps might help to identify a more specific pathway that is related to the motor neurons of ALS.

Variation in the proportion of astrocytes in the culture dish can also be expected to cause noise in the analysis. The datasets used were iPSC-derived samples obtained from different differentiation stages and different culture conditions, and were too crude to be analyzed on the same platform. Correction with appropriate intrinsic controls is also challenging. A meta-analysis of the results found in each dataset might be acceptable.

As commented as major points, the correction among different batches of RNAseq and/or single-cell RNAseq analysis is important and must be quite challenging even in the unified

experimental method or condition. The authors should show how the authors corrected the noise among different platforms and evaluated the corrected quality to justify the usage of these datasets.

- We thank the reviewer for these astute points regarding dataset integration. We agree that the expression of motor neuron markers between iPSMN datasets is variable - this is a key aspect of this study that is made possible by the systematic QC and comparison of iPSMN datasets.
- We have corrected for the variation in gene expression between datasets by including dataset as a covariate in the DESeq2 differential expression design model: $\sim \text{gender} + \text{dataset} + \text{condition}$. When contrasting ALS versus control, gender and dataset are treated as unwanted batch effects, which effectively ensures that variation in gene expression due to gender or dataset differences is corrected for, increasing the specificity for finding differences due to ALS. This model favours genes that have a homogeneous and consistent magnitude of ALS vs. CTRL effect across the datasets, whilst allowing for differences in baseline expression between datasets. This approach safeguards samples within datasets, helping to ensure that controls and ALS are treated similarly. A detailed explanation of how this is performed statistically is explained at <http://bioconductor.org/packages/devel/bioc/vignettes/DESeq2/inst/doc/DESeq2.html> (search for “batch” within this webpage).
- To visualise the effect of removing dataset batch effects on motor neuron marker expression, here we have replotted the Ext Dat Fig 4 heatmap after applying the removeBatchEffect function from limma:

- To clarify, single-cell RNAseq is not used in this study, which is based on bulk RNAseq. This is an important distinction since the integration of single-cell RNAseq datasets is a substantially more challenging problem than with bulk RNAseq - primarily due to the shallow sequencing depth and large variance between individual cells in single-cell RNAseq.

Author Changes

- Describing this study as a meta-analysis is a mistake on our part. Whilst a meta-analysis does combine multiple studies, it is typically a synthesis of their pre-analysed results, for example, combining each individual study p-values using the metaRNAseq package. However, the approach we have taken of integrating the raw sequencing data from the component datasets, and performing de-novo differential expression analysis, is substantially more robust at being able to identify and remove batch effects between datasets. To make this clearer we replace the use of “meta-analysis” with “integrative analysis”.
- We have noted the variability in motor neuron marker expression in the discussion:

In our compendium of iPSMNs, we highlight how the expression of rostro-caudal and dorso-ventral spinal cord markers vary between differentiation protocols (Table S1, Extended Data Fig. 4-7). While there were no clear links between differentiation protocols and the expression of MN domain markers, these findings should guide the field in optimising *in vitro* culturing strategies to promote the maturation of iPSMNs⁶³.

- To ensure readers are aware that only bulk RNAseq datasets were included, and that single-cell RNAseq and single-nuclear RNAseq were excluded, we have made this explicit in the text:

Our search strategy of databases identified 16 ALS iPSMN datasets that had undergone **bulk** RNA-sequencing (RNA-seq; Fig. 1, Extended Data Fig. 1).

Single-cell and single-nuclear RNA-seq datasets were excluded.

- To address the point regarding thresholds for selection of datasets, we have made clearer the inclusion and exclusion criteria - please see the response to Reviewer 3, Comment 10.
- We have now added an analysis of each dataset independently as noted in the response to Reviewer 3, Comment 1.

Comment 5

As mentioned in the above comments, this reviewer believes that the reproducibility of the results of such meta-analyses, based on datasets collected from different platforms, should be confirmed by other methods. Since the relationship between p53 and ALS is a known phenomenon that has already been reported elsewhere, this reviewer believes that this point has been clarified. On the flip side, the identified results can be considered as a result without novelty. Can the authors direct more focus toward new pathways, identified only by the present meta-analysis?

- We thank the reviewer for these comments. We have highlighted the novelty of this study in the response to Reviewer 3, Comment 2 (sample size, across ALS subtypes, iPSMN and post-mortem, TDP-43 contribution, somatic mutations and gene fusions).
- As the reviewer points out, the unbiased transcriptome-wide approach used reveals multiple novel targets in ALS beyond p53. Among these are the RNase L gene expression change, MAPK signalling pathway, CNOT3 splicing event. It also provides the largest study of splicing changes in ALS iPSMNs, not only revealing TDP-43 regulated events (e.g. *POLDIP3*, *CAMK2B*, *CEP290*) but also revealing new splicing events shared between ALS subgroups (e.g. *CNOT3*, *SRSF10*, *DNAJC17*).

Author Changes

- We have now added an analysis of each dataset independently - please see the response to Reviewer 3, Comment 1.
- We have now added TDP-43 subgroup analyses and an integrative analysis of TDP-43 knockdown datasets - please see the response to Reviewer 3, Comment 6.

Comment 6

In figure 3, the authors point out that the p53 pathology is common in different ALS subtypes. This reviewer believes that it is also important to take advantage of the different ALS subtypes that are being addressed in this study. Can the authors classify the characteristics of the pathology and the signaling pathways for each ALS subtype? For example, searching for ALS with TDP43 pathology and others, and their characteristic pathways, would be a new focus for this meta-analysis?

- We greatly thank the reviewer for this suggestion to compare the different ALS subtypes, particularly based on TDP-43 pathology.
- Examining each ALS subtype for iPSMNs are shown in Extended Data Figures 12-13.

Author Changes

- We have now added a more extensive comparison of each ALS subtype for postmortem tissue:

In contrast to iPSMNs, comparing genetic subgroups from post-mortem spinal cord tissue revealed strongly correlated gene expression changes (R range +0.68 to +0.9) with 1,750 overlapping differentially expressed genes between sporadic, *C9orf72*, *SOD1*, and *FUS* subgroups (Fig. 4h, Extended Data Fig. 14a-j). In each subgroup, upregulated differentially expressed genes were consistently over-represented by the stress response, cell death, and protein metabolism, while downregulated genes were enriched in protein binding and neuronal terms (Extended Data Fig. 14e-h). Examining signalling pathways and transcription factors in each genetic subgroup revealed that p53 signalling and TP53 TF activity were **significantly** increased in both in sporadic (n = 161; p53: NES +5.0, p < 0.001; TP53 NES +2.9, p = 0.05) and *C9orf72* (n = 36; NES +5.3, p < 0.001; TP53 NES +4.7, p < 0.001) and **non significantly increased in *SOD1*** (n = 5; NES +3.3, p = 0.05; TP53 NES +1.2, p = 0.1) and *FUS* (n = 2; NES +2.9, p = 0.28; TP53 NES +2.9, p = 0.29; Fig. 4i, Extended Data Fig. 14l,m).

Extended Data Figure 14 Postmortem spinal cord gene expression changes between genetic backgrounds

a-d, Volcano plots comparing ALS to control post-mortem tissue in each ALS genetic background. Genes coloured red are significantly increased in the ALS subgroup, and genes coloured blue are decreased in the ALS subgroup. **e-f**, Upset plots showing overlapping (j) upregulated and (k) downregulated differentially expressed genes (FDR < 0.05) between each genetic subgroup. **g-j**, Functional enrichment terms enriched in (e) Sporadic, (f) C9orf72, (g) SOD1, and (h) FUS. Upregulated terms are coloured red and downregulated are blue. **k-l**, normalised enrichment scores in the distinct ALS genetic subgroups versus controls for (l) PROGENy signalling pathways and (m) DoRoThEA transcription factor activities.

- We have added a new analysis **comparing TDP-43 ALS to non-TDP-43 ALS cases**, revealing that TDP-43 ALS shows significantly greater p53 activation than non-TDP-43 ALS in both iPSMNs and post-mortem.
- To examine for a **functional link between TDP-43 nuclear depletion and p53 activation**, we have now examined RNA-seq from **neuronal nuclei depleted of TDP-43 from patients' brains**⁶ as well as integrating 7 RNA-seq datasets from human cultured cells that have undergone **TDP-43 knockdown with shRNA, siRNA, or CRISPR/Cas9 mediated inhibition**⁷⁻¹³. Both neuronal nuclei depleted of TDP-43 and cells undergoing TDP-43 knockdown show significant upregulation of p53 signaling, indicating that TDP-43 nuclear loss contributes to p53 activation.

Results text:

TDP-43 pathology contributes to the DNA damage response

Although TDP-43 pathology (characterised by neuronal nuclear depletion and cytoplasmic accumulation) is observed in 97% of ALS, it is absent in SOD1 and FUS mutant cases (non-TDP-43 ALS)^{24,25}. Interestingly, SOD1 and FUS mutants displayed the weakest p53 upregulation of the genetic subtypes in both iPSMNs and post-mortem. To identify the degree to which genotypes linked to TDP-43 pathology contribute to p53 upregulation in the pan-ALS analyses, we classified ALS samples based on whether their genetic background is associated with TDP-43 pathology. In iPSMNs, whilst non-TDP-43 ALS (SOD1 and FUS mutant) iPSMNs showed only a modest, non-significant increase in p53 (NES = +2.0, p = 0.25), TDP-43 ALS iPSMNs exhibited strong and significant p53 upregulation (p53 NES = +14.2, p < 0.001). Likewise, in non-TDP-43 ALS iPSMNs the TP53 TF was mildly decreased in activity (NES = -1.6, p = 0.12), whereas TDP-43 ALS iPSMNs showed TP53 was the most strongly upregulated TF (NES + 7.4, p < 0.001; Extended Data Fig. 15a,b). We found a similar pattern in post-mortem samples, with non-TDP-43 ALS showing smaller increases in p53 signaling and TP53 TF activity (p53: NES +3.6, p = 0.03; TP53: NES +3.3, p = 0.01) as compared to TDP-43 ALS (p53: NES +5.2, p < 0.001; TP53: NES +2.0, p = 0.5; Extended Data Fig. 15c,d). These findings suggest that the p53 signature from the pan-ALS analysis is largely driven by genetic backgrounds associated with TDP-43 proteinopathy.

To discover whether p53 signalling changes are regulated by TDP-43, we next examined RNA-seq from FACS sorted neuronal nuclei into those with and without TDP-43 pathology from FTD-ALS postmortem brain tissue⁶. We found that neuronal nuclei depleted of TDP-43 showed significant upregulation of p53 signalling (NES +0.4, p = 0.02) and non-significant upregulation of TP53 TF as

compared to neuronal nuclei retaining TDP-43 (NES +0.7, $p = 0.26$; Extended Data Fig. 15e,f).

To determine whether TDP-43 depletion directly promotes p53 activation, we integrated seven RNA-seq datasets from human cells that have undergone TDP-43 knockdown with shRNA, siRNA, or CRISPR inhibition (Table S8)⁷⁻¹³. In support of a direct role for TDP-43 regulation of p53 signalling, we discovered significant upregulation of both p53 signalling (NES +5.5, $p = 0.02$) and TP53 TF activity (NES +3.5, $p = 0.02$) upon TDP-43 knockdown (Extended Data Fig. 15e,f).

Extended Data Figure 15 TDP-43 loss of function contributes to p53 signaling activation

PROGENy signalling pathway barcharts (left) and DoRoThEA transcription factor activities volcano plot (right) in (a-b) non-TDP-43 ALS (i.e. SOD1 and FUS mutant) and TDP-43 ALS iPSMNs; (c-d) non-TDP-43 ALS (i.e. SOD1 and FUS mutant) and TDP-43 ALS post-mortem; (e-f) FACS sorted neuronal nuclei depleted of TDP-43 and TDP-43 knockdown cell models.

Discussion text:

Our findings support previous, smaller studies showing p53 activation in ALS, particularly with *C9orf72* repeat expansions^{1-5,20-23}. However, the novelty in this study is the finding of p53 activation, together with increased somatic mutations and gene fusions, across diverse ALS subgroups. This finding, not only in iPSMNs but also in post-mortem spinal cord tissue, indicates that the DNA damage response in ALS begins early and persists into the later stages of the disease. We identified the greatest increase in p53 activity in *C9orf72* mutants, consistent with reports that the *C9orf72* repeat expansion induces DNA damage, likely mediated by dipeptide repeat proteins and the formation of R loops and G quadruplexes¹⁻⁵. However, we also found p53 to be strongly and significantly activated in *TARDBP* and sporadic subgroups. This finding in sporadic cases is particularly striking since they represent 90% of ALS but have the least prior evidence for p53 activation.

Although we found p53 activation in non-TDP-43 ALS (*FUS* and *SOD1* mutants), the magnitude of activation was substantially weaker than in TDP-43 ALS cases, raising the possibility that TDP-43 contributes to the DNA damage response. In support of this, we found significant p53 upregulation in neuronal nuclei depleted of TDP-43, as well as TDP-43 knockdown models. TDP-43 has an established role in DNA repair and TDP-43 depletion results in the accumulation of DNA damage^{23,26}. Conversely, TDP-43 overexpression in iPSC-derived neurons led to a pro-apoptotic phenotype, which was partially rescued by p53 inhibition^{27,28}. Other studies of p53 inhibition in TDP-43^{A315T} iPSC-derived neurons has also demonstrated a partial phenotypic rescue^{27,29}. Together, this suggests that TDP-43 pathology contributes to heightened DNA damage, and the subsequent p53 activation may promote motor neuron death in ALS.

Abstract text:

The strongest p53 activation was observed with *C9orf72* repeat expansions but was also found in sporadic ALS, which accounts for 90% of cases. Conversely, non-TDP-43 ALS cases due to *SOD1* and *FUS* mutations^{24,25}, showed weaker p53 activation than TDP-43 ALS. Loss of TDP-43 was found to potentiate p53 activation, both in neuronal nuclei of patients' brains and in cultured cells, providing a functional link between p53 activation and loss of TDP-43 function.

- We now examine splicing in TDP-43 knockdown and TDP-43 FACS sorted neuronal nuclei and compare these events to ALS iPSMNs.

By examining splicing changes in neuronal nuclei depleted of TDP-43⁶, we found 12 overlapping differentially spliced genes (encompassing 17 splicing events) with ALS iPSMN (including *POLDIP3*, *PPP6R3*, *CAMK2B*, *CEP290*; Table S8)⁶. Similarly, comparing splicing changes upon TDP-43 knockdown with ALS iPSMNs revealed 4 overlapping genes containing 6 splicing events (*POLDIP3*, *CAMK2B*, *HERC2P3*, *CEP290*; Table S8). Interestingly, the multi-exon skipping splicing event in *POLDIP3* was precisely the same event that occurs in both TDP-43 neuronal nuclei depletion and TDP-43 knockdown (Extended Data Fig. 15a-c)^{9,48}. This indicates that TDP-43 nuclear loss of function may contribute to splicing changes in ALS iPSMNs.

Methods text:

TDP-43 depletion

The post-mortem brain TDP-43 FACS sorted neuronal nuclei RNA-seq dataset was acquired from accession GSE126543 and processed using the same pipeline described above. Only NeuN positive samples were utilised. Differential expression results were calculated by comparing TDP-43 negative versus TDP-43 positive samples.

For analysis of artificially TDP-43 depleted human cells, we searched RNA-seq databases for TDP-43 knockdown datasets. We found 8 datasets^{7-13,64}, of which one had low TARDBP expression in controls and did not achieve >60% TARDBP reduction with depletion and was excluded (Table S8)⁶⁴. Differential expression results were calculated by comparing TDP-43 depleted versus control samples, accounting for dataset batch effects in the design with the formula $\sim dataset + gender + condition$.

Extended Data Figure 16 POLDIP3 splicing event in ALS iPSMNs, TDP-43 depleted neuronal nuclei, and TDP43 knockdown iNeurons

MAJIQ voila view of POLDIP3 multi-exon skipping event (coordinates start 42,602,768 and end 42,603,160) in (a) ALS versus control iPSMNs (using heterogen function), (b) TDP-43

positive versus TDP-43 negative FACS sorted neuronal nuclei from Liu et al 2019, and (c) TDP-43 knockdown versus control iNeurons from Leigh-Brown et al 2022 (using deltaPSI function). In the splice graphs, it is the green exon skipping event (coordinates 42,602,056-42,602,770) that is increased in ALS iPSMN, TDP-43 neuronal nuclei depletion, and TDP-43 knockdown compared to their respective controls. Conversely, the blue exon skipping event is decreased in ALS iPSMN, TDP-43 neuronal nuclei depletion, and TDP-43 knockdown compared to their respective controls.

Comment 7

In figure 4, the authors showed the altered splicing or SNVs in ALS iPSNs. To confirm the reproducibility of the findings, the authors should show the similar alteration in the other cohort, independent of the presented meta-analysis.

- We thank the reviewer for this suggestion. We are not clear what is meant by “the other cohort” but we assume this refers to either genetic subgroups or postmortem tissue.

Author Changes

- We have added splicing analyses on each genetic cohort independently:

We next investigated alternative splicing in each ALS genetic subgroup separately. Whilst there were no sporadic ALS iPSMNs that had undergone poly(A) selection, there were 23 *C9orf72*, 9 *FUS*, 9 *TARDBP* and 4 *SOD1* mutant poly(A) samples. Compared to controls, *TARDBP* mutants showed the greatest number of differential splicing events (1,435), followed by *FUS* (1,099), *C9orf72* (429) and *SOD1* (256; Extended Data Fig. 17a-d). Functional over-representation analysis in each mutant group revealed that genes exhibiting differential splicing were involved with protein binding, neuronal structures, and RNA processing (Extended Data Fig. 17e-h). Exon skipping was the most common splicing type in *TARDBP*, *FUS* and *C9orf72* subgroups, however, in *SOD1* mutants, intron retention was the most frequent (Extended Data Fig. 17i). Correlation of alternative splicing changes between genetic subgroups revealed weak associations with the strongest correlation between *FUS* and *C9orf72* mutations ($R = +0.09$) and the weakest correlation between *SOD1* and *TARDBP* ($R = -0.1$; Extended Data Fig. 17j). Intersecting significant splicing events between genetic backgrounds revealed that none were common to each genetic group, but that 33 were shared amongst two mutant groups (Extended Data Fig. 17k-l; Table S9). Amongst these were the p53 signalling and RBP gene *CNOT3* as well as other RBPs including *SRSF10*, *DNAJC17*, *SRRM1*, *SIDT2*, *SREK1*.

Extended Data Figure 17 Splicing changes in distinct ALS genetic backgrounds

a-d: Volcano plots showing splicing changes in ALS versus control iPSCs (delta PSI (x-axis) against $-\log_{10}$ test statistic for (a) TARDBBP mutant, (b) FUS mutant, (c) SOD1 mutant, and (d) C9orf72 mutant iPSCs. Splice events significantly increased in ALS are coloured red, and those significantly decreased are coloured blue.

e-h: Functionally enriched terms amongst differential splice events in (e) TARDBBP mutant, (f) FUS mutant, (g) SOD1 mutant, and (h) C9orf72 mutant iPSCs.

i: Barchart showing proportions of each splicing type in significant splice events in SOD1, TARDBBP, FUS, and C9orf72 mutant iPSCs. Labels depict the numbers and percent of splice events for the most common splicing types.

j: Heatmap showing the Pearson's correlation coefficient for transcriptome-wide splicing changes between each mutant group.

k-l: UpSet plots showing the numbers of overlapping splice events (k) increased and (l) decreased between mutations.

- Regarding splicing in postmortem tissue, NYGC uses total ribo-zero libraries (rather than polyA) and so we are unable to examine and compare splicing like for like between iPSMNs and postmortem.
- We have now added analysis of somatic mutations and gene fusions in postmortem tissue - please see our response to Reviewer 1, Comment 10.

Comment 8

In the extended data of figures 4 and 5, it is unclear what “variance stabilized gene counts” mean, and what the range of variance is as described by the blue/red color?

- We thank the reviewer for raising this. Variance-stabilised gene counts are gene counts that have been normalised using variance stabilizing transformation (VST), which is a log-like transformation that helps to ensure that lowly and highly expressed genes have similar expected variances. VST is the preferred choice of count data transformation as it avoids dependence on the variance of the mean. VST normalises counts for library size and other normalisation factors on a \log_2 scale. Further details can be found at: <http://bioconductor.org/packages/devel/bioc/vignettes/DESeq2/inst/doc/DESeq2.html#count-data-transformations>
- In all the heatmaps blue represents low expression and red represents high expression, where 0 (dark blue) represents undetectable gene expression, and the maximum (dark red) represents the greatest gene expression detected across all genes and samples plotted. The actual numbers are somewhat irrelevant as these vary depending on the samples and genes included - more important are the relative differences between genes and samples. There is no centering or scaling used, so normalised gene expression can be directly compared both between samples (rows) and genes (columns).

Author Changes

- For simplicity and to avoid confusing readers not familiar with VST, in the heatmap legends we refer to variance stabilised gene counts as normalised gene counts e.g.:

Heatmap of normalised gene counts for astrocytes, endothelial cells, ...

- The methods section still explains that gene counts were normalised using VST:

Gene counts were normalised **for library size and other normalisation factors and transformed on a \log_2 scale** using the variance stabilizing transformation function in DESeq2.

- We have explained the heatmap range of colour in the figure legends:

Blue represents low gene expression and red represents high expression, where 0 (dark blue) represents undetectable gene expression, and the maximum (dark red) represents the greatest gene expression detected across all genes and samples plotted.

Comment 9

The authors used “iPSNs” as the abbreviation for induced pluripotent stem cell-derived motor neurons. iPSC-derived motor neurons or iPSMNs etc. might be more appropriate and helpful for the readership in order to understand that this study focuses on the pathology of motor neurons with ALS.

Author Changes

- Thank you. We have changed iPSN to iPSMN throughout the manuscript text and figures.

Comment 10

The authors excluded the samples and datasets of Lee et al. and NeuroLINCS. This reviewer can intuitively understand that these samples are outliers when compared with other datasets, but this reviewer believes that the authors should clarify the threshold criteria to maintain reproducibility.

Author Changes

- We thank the reviewer for raising this important point and acknowledge that the inclusion and exclusion criteria could be clearer. To address this, we have now clarified these criteria in a new dedicated methods section:

Eligibility criteria

We evaluated all human iPSMN datasets that had undergone **short-read** bulk RNA-sequencing (RNA-seq) that examined **iPSMNs from individuals with ALS** and non-ALS controls (healthy individuals or isogenic correction), regardless of the **RNA extraction kit** (e.g. Qiagen RNeasy mini kit, Invitrogen TRIZol), library preparation (polyA or ribo-zero), **short-read lengths** (range 50-150 base pairs), **read sequencing** (single or paired-end), **sequencing instrument** (e.g. Illumina NovoSeq 6000, HiSeq 2500), or **sequencing depth** (Table S1). All ALS subtypes were included, and the definition of ALS used by each dataset was accepted. In datasets with multiple time points through motor neuron differentiation only the final most terminally differentiated time point was utilised⁶⁵.

We excluded datasets that (i) had not undergone an accepted spinal motor neuron differentiation protocol using the steps detailed in Sances et al ⁶³, (ii) failed RNA-seq quality control measures (Table S2), (iii) failed spinal motor neuron identity based on the expression of established spinal cord dorso-ventral and rostro-caudal markers (Extended Data Fig. 4-7), or (iv) exhibited unadjustable batch effects between ALS and control samples (e.g. different RNA library strategies or sequencing platforms). Long-read sequencing as well as single-cell and single-nuclear RNA-seq datasets were excluded.

Comment 11

The title seems slightly exaggerated, and it should be revised by focusing on the methodology.

Author Changes

- We thank the reviewer for this suggestion and have changed the title to address the dataset integration. We also remove iPSC-MN from the title as we have now shown genome instability in ALS postmortem tissue. We now add TDP-43 pathology to the title, given the new findings outlined in Reviewer 3, Comment 6.

Integrated analysis of ALS links genome instability to TDP-43 pathology

Reviewer 4

Comment 1

I have read with interest the work by Ziff et al., titled “Genome instability underlies an augmented DNA damage response in familial and sporadic ALS human iPSC- derived motor neurons”, dealing with the bioinformatics analysis of deep transcriptome data from ALS human induced pluripotent stem cell-derived motor neurons (iPSNs). The authors collected RNAseq data from 429 donors spanning 10 ALS mutations and sporadic ALS. They found an increase in the DNA damage response, which is characterized by the activation of p53 signalling.

All data were initially checked and low-quality samples were filtered out, leaving 323 ALS samples and 106 controls. Comparing these two groups revealed only 43 DE genes. In my opinion, this specific comparison is biologically questionable. Indeed, it has been performed by mixing heterogeneous RNAseq data from heterogeneous genetic backgrounds. Comparing transcriptomes from different genetic backgrounds should provide misleading results.

- We thank the reviewer for these comments and for raising the important issues around dataset integration. The decision to combine datasets needs careful assessment of the benefits and risks. Whilst the key risks have been outlined by the reviewer the benefits are twofold:
 - a. Increase the power to detect true positive changes and avoid false negative changes (i.e. increased sensitivity).
 - b. Increase the variety of cases (and controls) to avoid false positive changes (and detect true negative changes; i.e. increased specificity).
- Because of the cost of generating iPSMN lines, iPSMN studies have been greatly underpowered with many limited to only 2-3 patients, meaning that the sensitivity and specificity of their findings are limited. This is a major problem in ALS research because of the substantial genetic, pathological, and clinical heterogeneity. Combining RNAseq from 300+ ALS and 100+ control iPSMNs addresses this. Hence, the benefits of integrating the datasets largely outweigh the risks, many of which can be minimised with detailed QC and appropriately designed statistical tests. Indeed, we have gone to great depths to attempt to minimise these risks by:
 - a. Quality control checks: sequencing metrics (Table S2), unbiased transcriptome-wide PCA clustering (Ext Data Fig 2-3), and cell type identity and differentiation markers (Ext Dat Fig 4-7).
 - b. Using a statistical design in generalised linear models that includes covariates to remove unwanted variation due to dataset batch effects, including variations in differentiation protocols, library preparation, and sequencing instrument.
 - c. Sensitivity analyses examining potential sources of unwanted variation, including RNA library preparations (total and polyA), datasets independently (see below), and **different ALS subgroups**.
- There is precedence for integration of next-generation sequencing genome and transcriptome datasets from cancer, where multiple distinct cancer type datasets have been successfully integrated in pan-cancer analyses, for example, Drews et al Nature 2022¹⁴, Liu et al Nature 2022¹⁵, Litchfield et al Cell 2021¹⁶, PCAWG Group Nature 2020¹⁷. Furthermore, we have previously published this approach of integrating RNA-seq datasets of iPSC-derived astrocytes: Ziff et al. Genome Research 2021¹⁸

Author Changes

- To further address the concern of integrating datasets, we have now added an analysis of each dataset independently.

We further investigated changes between ALS and control samples in each dataset separately. This revealed substantial heterogeneity between datasets in ALS versus control gene expression changes (Extended Data Fig. 10a-b). Despite this, both TP53 transcription factor and p53 signalling were independently upregulated in ALS iPSMNs in 11 of 17 datasets (Extended Data Fig. 10c-d). This indicates that neither library preparation nor dataset batch effects were responsible for the DNA damage response gene expression changes observed in ALS iPSMNs.

Extended Data Figure 10 Analysis of each iPSMN dataset separately

a: Volcano plots showing log₂ fold change in differential gene expression in ALS compared to control iPSMNs in each dataset. Red genes are significantly (FDR < 0.05) increased and blue is decreased in ALS.

b: Heatmap showing the Pearson's correlation coefficient for transcriptome-wide changes between each dataset.

c p53 signalling pathway activity showing ALS versus control iPSMNs normalised enrichment

scores (y-axis) in each dataset. Pathways that are increased in ALS are coloured red whilst pathways decreased are coloured blue. * represents enrichment p-value < 0.05, ** p < 0.01, *** p < 0.001, **** p < 0.0001. d: TP3 transcription factor activity in ALS versus control iPSMNs in each dataset. Normalised enrichment in ALS versus control (x-axis) is plotted for each dataset (y-axis).

Comment 2

Additionally, the effect of the differentiation protocol has not been taken into account. It could really impact the detection of DE genes.

- We thank the reviewer for raising this point. We have corrected for the variation in gene expression between differentiation protocols by including dataset as a covariate in the DESeq2 differential expression design model: $\sim \text{gender} + \text{dataset} + \text{condition}$. The differentiation protocol is nested within the dataset covariate, and so the differentiation protocol has been taken into account in the differential expression analyses. Also nested within dataset, are other covariates e.g. RNA library and sequencing technology, that are also taken into account.
- When contrasting ALS versus control, gender and dataset are treated as unwanted batch effects, which effectively ensures that variation in gene expression due to gender or differentiation protocol differences is corrected for, increasing the specificity for finding differences due to ALS. This model favours genes that have a homogeneous and consistent magnitude of ALS vs. CTRL effect across the datasets, whilst allowing for differences in baseline expression between datasets. This approach safeguards samples within datasets, helping to ensure that controls and ALS are treated similarly. A detailed explanation of how this is performed statistically is explained at <http://bioconductor.org/packages/devel/bioc/vignettes/DESeq2/inst/doc/DESeq2.html> (search for “batch” within this webpage).
- To visualise the effect of removing dataset batch effects on motor neuron marker expression, here we have replotted the Ext Dat Fig 4 heatmap after applying the removeBatchEffect function from limma:

Author Changes

- We have now added an analysis of each dataset independently as noted in the response to Reviewer 4, Comment 1.

Comment 3

Regarding the identification of aberrant splicing, the authors compared again all ALS samples versus control iPSNs. I don't think different genetic backgrounds should be mixed.

- We thank the reviewer for raising this suggestion to evaluate splicing in each genetic background separately. We agree that, as for gene expression, a subgroup analysis of each ALS genetic subgroup for alternative splicing may expose therapeutically relevant heterogeneity. Nonetheless, because of the possibility of identifying common signatures across ALS genetic

backgrounds combined with greater statistical power, we feel the benefits outweigh the potential risks of integrating genetic backgrounds.

Author Changes

- We have added splicing analyses on each genetic cohort independently:

We next investigated alternative splicing in each ALS genetic subgroup separately. Whilst there were no sporadic ALS iPSMNs that had undergone poly(A) selection, there were 23 *C9orf72*, 9 *FUS*, 9 *TARDBP* and 4 *SOD1* mutant poly(A) samples. Compared to controls, *TARDBP* mutants showed the greatest number of differential splicing events (1,435), followed by *FUS* (1,099), *C9orf72* (429) and *SOD1* (256; Extended Data Fig. 17a-d). Functional over-representation analysis in each mutant group revealed that genes exhibiting differential splicing were involved with protein binding, neuronal structures, and RNA processing (Extended Data Fig. 17e-h). Exon skipping was the most common splicing type in *TARDBP*, *FUS* and *C9orf72* subgroups, however, in *SOD1* mutants, intron retention was the most frequent (Extended Data Fig. 17i). Correlation of alternative splicing changes between genetic subgroups revealed weak associations with the strongest correlation between *FUS* and *C9orf72* mutations ($R = +0.09$) and the weakest correlation between *SOD1* and *TARDBP* ($R = -0.1$; Extended Data Fig. 17j). Intersecting significant splicing events between genetic backgrounds revealed that none were common to each genetic group, but that 33 were shared amongst two mutant groups (Extended Data Fig. 17k-l; Table S9). Amongst these were the p53 signalling and RBP gene *CNOT3* as well as other RBPs including *SRSF10*, *DNAJC17*, *SRRM1*, *SIDT2*, *SREK1*.

Extended Data Figure 17 Splicing changes in distinct ALS genetic backgrounds

a-d: Volcano plots showing splicing changes in ALS versus control iPSMNs (delta PSI (x-axis) against $-\log_{10}$ test statistic for (a) TARDBP mutant, (b) FUS mutant, (c) SOD1 mutant, and (d) C9orf72 mutant iPSMNs. Splice events significantly increased in ALS are coloured red, and those significantly decreased are coloured blue.

e-h: Functionally enriched terms amongst differential splice events in (e) TARDBP mutant, (f) FUS mutant, (g) SOD1 mutant, and (h) C9orf72 mutant iPSMNs.

i: Bar chart showing proportions of each splicing type in significant splice events in SOD1, TARDBP, FUS, and C9orf72 mutant iPSMNs. Labels depict the numbers and percent of splice events for the most common splicing types.

j: Heatmap showing the Pearson’s correlation coefficient for transcriptome-wide splicing changes between each mutant group.

k-l: UpSet plots showing the numbers of overlapping splice events (k) increased and (l) decreased between mutations.

- Regarding splicing in postmortem tissue, NYGC uses total ribo-zero libraries (rather than polyA) and so we are unable to examine and compare splicing like for like between iPSMNs and postmortem.

Comment 4

About SNVs, statistics are based on the number of variants. It is not correct because the number of variants per sample is strongly connected with the coverage depth.

- We thank the reviewer for this point and agree that there is a correlation between the number of variants detected and read depth. However, we have already addressed this issue by (i) restricting the variant analysis to only the AnswerALS dataset (the largest iPSMN dataset, representing over half of all samples) and (ii) adjusting for the read coverage in the generalised linear model statistical design. Thus, we are confident that the variant results are not confounded by read depth.

Author Changes

- We have now examined somatic mutations and gene fusions in ALS postmortem spinal cord - please see our response to Reviewer 1, Comment 10.
- For the violin plots in Fig 6, instead of displaying the raw observed number of somatic mutations and fusion (which can be misleading), we have now plotted the normalised counts for read depth and age using partial residuals as per the jtools vignette. The partial residuals allow us to remove the variation due to read depth and age, but leave in the residual variation due to ALS vs control, thus offering the best visual representation of the modelled data.

Methods text:

In the violin plots, the observed number of variants and fusions were adjusted for read depth and donor age by using the partial residuals from the generalised linear model, which was performed using the `partialize` function from the `jtools` package.

- For genetic subgroups, forest plots were generated, which depict the regression coefficient results from the generalised linear model showing the point estimate and 95% confidence interval.

Fig. 6: ALS iPSMNs and post-mortem tissue accumulate somatic mutations and gene fusions.

a, Violin plots showing the numbers of somatic mutations, adjusted for age and read depth, identified in Answer ALS iPSMNs in ALS (red) and CTRL (blue) samples for all variant types, single-nucleotide Variant (SNV), Insertions, and Deletions. **b**, Forest plot showing the point estimate and 95% confidence interval of changes in variant types (SNV, blue; Insertion, red; Deletion, green) in ALS genetic subgroups versus controls. Vertical dashed line indicates no difference, to the right of the dashed line indicates increase in ALS. **c-d**, As for a-b except in post-mortem spinal cord. **e**, Violin plots showing the adjusted number of gene fusions in CTRL (blue) and ALS (red) in iPSMNs. **f**, Forest plot showing the point estimate and 95% confidence interval changes in each genetic subtype versus controls. To the right of the dashed line indicates an increase in ALS subtypes. **g-h**, As for e-f except in post-mortem. **** represents Wald test $P < 0.0001$, *** $P < 0.001$, ** $P < 0.01$, * $P < 0.05$, after adjusting for read coverage, age, and dataset covariates.

- The numbers of SNVs was recently reported in Nature in neurons from individuals with Alzheimer's Disease⁵⁶. In their Fig 1d-h, age was shown to influence the number of variants and so we have now adjusted for age in both the variant and fusion generalised linear models and for both iPSMNs and postmortem:

After adjusting for sequencing depth and donor age, across all filtered variant types, we found significantly greater numbers of somatic mutations per iPSMN in ALS compared to control (Wald test $p < 2 \times 10^{-16}$; Fig. 6a).

Since gene fusion discovery is also sensitive to differences in read coverage and age, we adjusted for the read coverage, the donor age, and dataset batch effects in a generalised linear model.

To compare the number of somatic mutations per sample in ALS versus CTRL groups, a generalised linear model was fit specifying a Poisson distribution adjusting for differences in read coverage per sample and donor age using a spline: $variant\ count \sim condition + rcs(read\ depth, 3) + rcs(age, 3)$. For post-mortem tissue we included an additional term to adjust for the sequencing batch: $variant\ count \sim condition + batch + rcs(read\ depth, 3) + rcs(age, 3)$.

To compare the number of fusions per sample in ALS versus CTRL groups, a generalised linear model was fit specifying a Poisson distribution adjusting for differences in read coverage per sample and donor age and batch effects between datasets: $fusion\ count \sim condition + dataset + rcs(read\ depth, 3) + rcs(age, 3)$. For post-mortem tissue we included an additional term to adjust for the sequencing batch: $fusion\ count \sim condition + batch + rcs(read\ depth, 3) + rcs(age, 3)$.

Comment 5

Although the idea is quite interesting and the results are promising, I see methodological issues that need to be addressed. Additionally, the p53 signalling pathway includes many genes. A punctual study of this pathway has to be performed and validated in independent cohorts.

- We thank the reviewer for the positive comments and the suggestion to investigate the p53 pathway in independent cohorts. We have investigated the individual genes within the p53 signalling pathway using decoupleR (PROGENy), which allows us to account for the sign and weight of each gene within the network. We expose the individual genes with the greatest positive weighting, driving the p53 term enrichment in ALS - CDKN1A, SESN1, RRM2B:

- We have examined p53 signalling in independent cohorts:
 - a. poly(A) & total RNA samples separately
 - b. Genetic backgrounds separately: C9orf72, TARDBP, Sporadic ALS, and FUS separately
 - c. iPSMN & postmortem spinal cord separately
 - d. TDP-43 ALS and non-TDP-43 ALS pathology subgroups separately

Author Changes

- To assess more specific gene sets, we have performed a GSEA on the daughter terms of the p53 signalling pathway and DNA damage response:

Since p53 signalling and DNA damage response are large pathways, we next explored more specific gene sets by examining their daughter pathways. This revealed strong upregulation of genes involved with the mitotic G1 DNA damage checkpoint and intrinsic apoptosis signalling (Extended Data Fig. 8).

Extended Data Figure 8 Gene set enrichment analysis of DNA damage daughter terms

Daughter terms of DNA damage pathway upregulated in ALS are red (to the right) and those decreased in ALS iPSMN are blue (left). **** $p < 0.0001$, *** $p < 0.001$, ** $p < 0.01$, * adjusted $p < 0.05$ from GSEA enrichment test.

- As with iPSMNs, we have added PROGENy weights for each gene in the p53 pathway in ALS postmortem spinal cord (Fig. 4e):

Exploring individual genes in the p53 pathway according to their PROGENy weights revealed that the top genes driving p53 activity in ALS post-mortem include *FAS*, *RRM2B*, *CSTA*, *ZMAT3*, *TP53I3*, and *CDKN1A* (Fig. 4e). *CDKN1A* is a cyclin-dependent kinase inhibitor and its upregulation in both ALS iPSMNs and post-mortem tissue may represent a possible cell cycle block through cyclin-dependent kinase inhibition.

Fig. 4e: Expression changes of p53 signalling pathway genes in postmortem spinal cord ALS versus control according to their PROGENy weights. Genes increasing p53 activity in ALS are red whilst genes decreasing p53 activity in ALS are blue.

- We have now added an analysis of p53 signalling each dataset independently - please see the response to Reviewer 4 Comment 1.

References

1. Maor-Nof M, Shipony Z, Lopez-Gonzalez R, Nakayama L, Zhang YJ, Couthouis J, Blum JA, Castruita PA, Linares GR, Ruan K, Ramaswami G, Simon DJ, Nof A, Santana M, Han K, Sinnott-Armstrong N, Bassik MC, Geschwind DH, Tessier-Lavigne M, Attardi LD, Lloyd TE, Ichida JK, Gao FB, Greenleaf WJ, Yokoyama JS, Petrucelli L, Gitler AD. p53 is a central regulator driving neurodegeneration caused by C9orf72 poly(PR). *Cell*. 2021 Feb 4;184(3):689–708.e20. PMID: 33886018
2. Farg MA, Konopka A, Soo KY, Ito D, Atkin JD. The DNA damage response (DDR) is induced by the C9orf72 repeat expansion in amyotrophic lateral sclerosis. *Hum Mol Genet*. 2017 Aug 1;26(15):2882–2896. PMID: 28481984
3. Lopez-Gonzalez R, Lu Y, Gendron TF, Karydas A, Tran H, Yang D, Petrucelli L, Miller BL, Almeida S, Gao FB. Poly(GR) in C9ORF72 -Related ALS/FTD Compromises Mitochondrial Function and Increases Oxidative Stress and DNA Damage in iPSC-Derived Motor Neurons [Internet]. *Neuron*. 2016. p. 383–391. Available from:

<http://dx.doi.org/10.1016/j.neuron.2016.09.015>

4. Walker C, Herranz-Martin S, Karyka E, Liao C, Lewis K, Elsayed W, Lukashchuk V, Chiang SC, Ray S, Mulcahy PJ, Jurga M, Tsagakis I, Iannitti T, Chandran J, Coldicott I, De Vos KJ, Hassan MK, Higginbottom A, Shaw PJ, Hautbergue GM, Azzouz M, El-Khamisy SF. C9orf72 expansion disrupts ATM-mediated chromosomal break repair. *Nat Neurosci*. 2017 Sep;20(9):1225–1235. PMID: PMC5578434
5. Nihei Y, Mori K, Werner G, Arzberger T, Zhou Q, Khosravi B, Japok J, Hermann A, Sommacal A, Weber M, Kamp F, Nuscher B, Edbauer D, Haass C, German Consortium for Frontotemporal Lobar Degeneration, Bavarian Brain Banking Alliance. Poly-glycine–alanine exacerbates C9orf72 repeat expansion-mediated DNA damage via sequestration of phosphorylated ATM and loss of nuclear hnRNP A3. *Acta Neuropathol*. 2020 Jan 1;139(1):99–118.
6. Liu EY, Russ J, Cali CP, Phan JM, Amlie-Wolf A, Lee EB. Loss of Nuclear TDP-43 Is Associated with Decondensation of LINE Retrotransposons. *Cell Rep*. 2019 Apr 30;27(5):1409–1421.e6. PMID: PMC6508629
7. Tam OH, Rozhkov NV, Shaw R, Kim D, Hubbard I, Fennessey S, Propp N, NYGC ALS Consortium, Fagegaltier D, Harris BT, Ostrow LW, Phatnani H, Ravits J, Dubnau J, Gale Hammell M. Postmortem Cortex Samples Identify Distinct Molecular Subtypes of ALS: Retrotransposon Activation, Oxidative Stress, and Activated Glia. *Cell Rep*. 2019 Oct 29;29(5):1164–1177.e5. PMID: PMC6866666
8. Appocher C, Mohagheghi F, Cappelli S, Stuani C, Romano M, Feiguin F, Buratti E. Major hnRNP proteins act as general TDP-43 functional modifiers both in Drosophila and human neuronal cells. *Nucleic Acids Res*. 2017 Jul 27;45(13):8026–8045. PMID: PMC5570092
9. Brown AL, Wilkins OG, Keuss MJ, Hill SE, Zanovello M, Lee WC, Bampton A, Lee FCY, Masino L, Qi YA, Bryce-Smith S, Gatt A, Hallegger M, Fagegaltier D, Phatnani H, NYGC ALS Consortium, Newcombe J, Gustavsson EK, Seddighi S, Reyes JF, Coon SL, Ramos D, Schiavo G, Fisher EMC, Raj T, Secrier M, Lashley T, Ule J, Buratti E, Humphrey J, Ward ME, Fratta P. TDP-43 loss and ALS-risk SNPs drive mis-splicing and depletion of UNC13A. *Nature [Internet]*. 2022 Feb 23; Available from: <http://dx.doi.org/10.1038/s41586-022-04436-3> PMID: 35197628
10. Klim JR, Williams LA, Limone F, Guerra San Juan I, Davis-Dusenbery BN, Mordes DA, Burberry A, Steinbaugh MJ, Gamage KK, Kirchner R, Moccia R, Cassel SH, Chen K, Wainger BJ, Woolf CJ, Egan K. ALS-implicated protein TDP-43 sustains levels of STMN2, a mediator of motor neuron growth and repair. *Nat Neurosci*. 2019 Feb;22(2):167–179. PMID: PMC7153761
11. Melamed Z 'ev, López-Erauskin J, Baughn MW, Zhang O, Drenner K, Sun Y, Freyermuth F, McMahon MA, Beccari MS, Artates JW, Ohkubo T, Rodriguez M, Lin N, Wu D, Bennett CF, Rigo F, Da Cruz S, Ravits J, Lagier-Tourenne C, Cleveland DW. Premature polyadenylation-mediated loss of stathmin-2 is a hallmark of TDP-43-dependent neurodegeneration. *Nat Neurosci*. 2019 Feb;22(2):180–190. PMID: PMC6348009
12. Rocznik-Ferguson A, Ferguson SM. Pleiotropic requirements for human TDP-43 in the regulation of cell and organelle homeostasis. *Life Sci Alliance [Internet]*. 2019 Oct;2(5). Available from: <http://dx.doi.org/10.26508/lsa.201900358> PMID: PMC6749094
13. Dunker W, Ye X, Zhao Y, Liu L, Richardson A, Karijolic J. TDP-43 prevents endogenous RNAs from triggering a lethal RIG-I-dependent interferon response. *Cell Rep*. 2021 Apr 13;35(2):108976. PMID: PMC8109599
14. Drews RM, Hernando B, Tarabichi M, Haase K, Lesluyes T, Smith PS, Morrill Gavarró L,

- Couturier DL, Liu L, Schneider M, Brenton JD, Van Loo P, Macintyre G, Markowitz F. A pan-cancer compendium of chromosomal instability. *Nature*. 2022 Jun;606(7916):976–983. PMID: PMC7613102
15. Liu R, Rizzo S, Waliyany S, Garmhausen MR, Pal N, Huang Z, Chaudhary N, Wang L, Harbron C, Neal J, Copping R, Zou J. Systematic pan-cancer analysis of mutation-treatment interactions using large real-world clinicogenomics data. *Nat Med*. 2022 Aug;28(8):1656–1661. PMID: 35773542
 16. Litchfield K, Reading JL, Puttick C, Thakkar K, Abbosh C, Bentham R, Watkins TBK, Rosenthal R, Biswas D, Rowan A, Lim E, Al Bakir M, Turati V, Guerra-Assunção JA, Conde L, Furness AJS, Saini SK, Hadrup SR, Herrero J, Lee SH, Van Loo P, Enver T, Larkin J, Hellmann MD, Turajlic S, Quezada SA, McGranahan N, Swanton C. Meta-analysis of tumor- and T cell-intrinsic mechanisms of sensitization to checkpoint inhibition. *Cell*. 2021 Feb 4;184(3):596–614.e14. PMID: PMC7933824
 17. PCAWG Transcriptome Core Group, Calabrese C, Davidson NR, Demircioğlu D, Fonseca NA, He Y, Kahles A, Lehmann KV, Liu F, Shiraiishi Y, Soulette CM, Urban L, Greger L, Li S, Liu D, Perry MD, Xiang Q, Zhang F, Zhang J, Bailey P, Erkek S, Hoadley KA, Hou Y, Huska MR, Kilpinen H, Korbel JO, Marin MG, Markowski J, Nandi T, Pan-Hammarström Q, Pedamallu CS, Siebert R, Stark SG, Su H, Tan P, Waszak SM, Yung C, Zhu S, Awadalla P, Creighton CJ, Meyerson M, Ouellette BFF, Wu K, Yang H, PCAWG Transcriptome Working Group, Brazma A, Brooks AN, Göke J, Rätsch G, Schwarz RF, Stegle O, Zhang Z, PCAWG Consortium. Genomic basis for RNA alterations in cancer. *Nature*. 2020 Feb;578(7793):129–136. PMID: PMC7054216
 18. Ziff OJ, Clarke BE, Taha DM, Crerar H, Luscombe NM, Patani R. Meta-analysis of human and mouse ALS astrocytes reveals multi-omic signatures of inflammatory reactive states. *Genome Res [Internet]*. 2021 Dec 28; Available from: <http://dx.doi.org/10.1101/gr.275939.121> PMID: 34963663
 19. Kok JR, Palminha NM, Dos Santos Souza C, El-Khamisy SF, Ferraiuolo L. DNA damage as a mechanism of neurodegeneration in ALS and a contributor to astrocyte toxicity. *Cell Mol Life Sci*. 2021 Aug;78(15):5707–5729. PMID: PMC8316199
 20. Sun Y, Curle AJ, Haider AM, Balmus G. The role of DNA damage response in amyotrophic lateral sclerosis. *Essays Biochem*. 2020 Oct 26;64(5):847–861. PMID: PMC7588667
 21. Martin LJ. p53 is abnormally elevated and active in the CNS of patients with amyotrophic lateral sclerosis. *Neurobiol Dis*. 2000 Dec;7(6 Pt B):613–622. PMID: 11114260
 22. Ranganathan S, Bowser R. p53 and Cell Cycle Proteins Participate in Spinal Motor Neuron Cell Death in ALS. *Open Pathol J*. 2010 Jan 1;4:11–22. PMID: PMC3092395
 23. Mitra J, Guerrero EN, Hegde PM, Liachko NF, Wang H, Vasquez V, Gao J, Pandey A, Taylor JP, Kraemer BC, Wu P, Boldogh I, Garruto RM, Mitra S, Rao KS, Hegde ML. Motor neuron disease-associated loss of nuclear TDP-43 is linked to DNA double-strand break repair defects. *Proc Natl Acad Sci U S A*. 2019 Mar 5;116(10):4696–4705. PMID: PMC6410842
 24. Vance C, Rogelj B, Hortobágyi T, De Vos KJ, Nishimura AL, Sreedharan J, Hu X, Smith B, Ruddy D, Wright P, Ganesalingam J, Williams KL, Tripathi V, Al-Saraj S, Al-Chalabi A, Leigh PN, Blair IP, Nicholson G, de Belleruche J, Gallo JM, Miller CC, Shaw CE. Mutations in FUS, an RNA processing protein, cause familial amyotrophic lateral sclerosis type 6. *Science*. 2009 Feb 27;323(5918):1208–1211. PMID: PMC4516382
 25. Mackenzie IRA, Bigio EH, Ince PG, Geser F, Neumann M, Cairns NJ, Kwong LK, Forman MS,

- Ravits J, Stewart H, Eisen A, McClusky L, Kretzschmar HA, Monoranu CM, Highley JR, Kirby J, Siddique T, Shaw PJ, Lee VMY, Trojanowski JQ. Pathological TDP-43 distinguishes sporadic amyotrophic lateral sclerosis from amyotrophic lateral sclerosis with SOD1 mutations. *Ann Neurol*. 2007 May;61(5):427–434. PMID: 17469116
26. Hill SJ, Mordes DA, Cameron LA, Neuberger DS, Landini S, Eggan K, Livingston DM. Two familial ALS proteins function in prevention/repair of transcription-associated DNA damage. *Proc Natl Acad Sci U S A*. 2016 Nov 29;113(48):E7701–E7709. PMID: PMC5137757
 27. Vogt MA, Ehsaei Z, Knuckles P, Higginbottom A, Helmbrecht MS, Kunath T, Eggan K, Williams LA, Shaw PJ, Wurst W, Floss T, Huber AB, Taylor V. TDP-43 induces p53-mediated cell death of cortical progenitors and immature neurons. *Sci Rep*. Nature Publishing Group; 2018 May 25;8(1):1–13.
 28. Lee K, Suzuki H, Aiso S, Matsuoka M. Overexpression of TDP-43 causes partially p53-dependent G2/M arrest and p53-independent cell death in HeLa cells. *Neurosci Lett*. 2012 Jan 11;506(2):271–276. PMID: 22133803
 29. Mc Guire C, Beyaert R, van Loo G. Death receptor signalling in central nervous system inflammation and demyelination. *Trends Neurosci*. 2011 Dec;34(12):619–628. PMID: 21999927
 30. Wang H, Kodavati M, Britz GW, Hegde ML. DNA Damage and Repair Deficiency in ALS/FTD-Associated Neurodegeneration: From Molecular Mechanisms to Therapeutic Implication. *Front Mol Neurosci*. 2021 Dec 16;14:784361. PMID: PMC8716463
 31. Wang H, Dharmalingam P, Vasquez V, Mitra J, Boldogh I, Rao KS, Kent TA, Mitra S, Hegde ML. Chronic oxidative damage together with genome repair deficiency in the neurons is a double whammy for neurodegeneration: Is damage response signaling a potential therapeutic target? *Mech Ageing Dev*. 2017 Jan 1;161:163–176.
 32. Schubert M, Klinger B, Klünemann M, Sieber A, Uhlitz F, Sauer S, Garnett MJ, Blüthgen N, Saez-Rodriguez J. Perturbation-response genes reveal signaling footprints in cancer gene expression. *Nat Commun*. 2018 Jan 2;9(1):20.
 33. Mertens J, Herdy JR, Traxler L, Schafer ST, Schlachetzki JCM, Böhnke L, Reid DA, Lee H, Zangwill D, Fernandes DP, Agarwal RK, Lucciola R, Zhou-Yang L, Karbacher L, Edenhofer F, Stern S, Horvath S, Paquola ACM, Glass CK, Yuan SH, Ku M, Szücs A, Goldstein LSB, Galasko D, Gage FH. Age-dependent instability of mature neuronal fate in induced neurons from Alzheimer's patients. *Cell Stem Cell*. 2021 Sep 2;28(9):1533–1548.e6. PMID: PMC8423435
 34. Al-Chalabi A, van den Berg LH, Veldink J. Gene discovery in amyotrophic lateral sclerosis: implications for clinical management. *Nat Rev Neurol*. 2017 Feb;13(2):96–104. PMID: 27982040
 35. Pamphlett R. Exposure to environmental toxins and the risk of sporadic motor neuron disease: an expanded Australian case-control study. *Eur J Neurol*. Wiley; 2012 Oct;19(10):1343–1348. PMID: 22642256
 36. Veldink JH, Kalmijn S, Groeneveld GJ, Titulaer MJ, Wokke JHJ, van den Berg LH. Physical activity and the association with sporadic ALS. *Neurology*. 2005 Jan 25;64(2):241–245. PMID: 15668420
 37. Al-Chalabi A, Hardiman O. The epidemiology of ALS: a conspiracy of genes, environment and time. *Nat Rev Neurol*. 2013 Nov;9(11):617–628. PMID: 24126629
 38. Armon C. An evidence-based medicine approach to the evaluation of the role of exogenous risk

factors in sporadic amyotrophic lateral sclerosis. *Neuroepidemiology*. 2003 Jul;22(4):217–228. PMID: 12792141

39. Andrew AS, Bradley WG, Peipert D, Butt T, Amoako K, Pioro EP, Tandan R, Novak J, Quick A, Pugar KD, Sawlani K, Katirji B, Hayes TA, Cazzolli P, Gui J, Mehta P, Horton DK, Stommel EW. Risk factors for amyotrophic lateral sclerosis: A regional United States case-control study. *Muscle Nerve*. 2021 Jan;63(1):52–59. PMID: PMC7821307
40. Humphrey J, Venkatesh S, Hasan R, Herb JT, de Paiva Lopes K, Kucukali F, Byrska-Bishop M, Evani US, Narzisi G, Fagegaltier D, NYGC ALS Consortium, Slegers K, Phatnani H, Knowles DA, Fratta P, Raj T. Integrative genetic analysis of the amyotrophic lateral sclerosis spinal cord implicates glial activation and suggests new risk genes. *medRxiv*. Cold Spring Harbor Laboratory Press; 2021 Sep 2;2021.08.31.21262682.
41. Schiffer D, Cordera S, Cavalla P, Migheli A. Reactive astrogliosis of the spinal cord in amyotrophic lateral sclerosis. *J Neurol Sci*. 1996 Aug;139 Suppl:27–33. PMID: 8899654
42. D’Erchia AM, Gallo A, Manzari C, Raho S, Horner DS, Chiara M, Valletti A, Aiello I, Mastropasqua F, Ciaccia L, Locatelli F, Pisani F, Nicchia GP, Svelto M, Pesole G, Picardi E. Massive transcriptome sequencing of human spinal cord tissues provides new insights into motor neuron degeneration in ALS. *Sci Rep*. 2017 Aug 30;7(1):10046. PMID: PMC5577269
43. Maniatis S, Äijö T, Vickovic S, Braine C, Kang K, Mollbrink A, Fagegaltier D, Andrusivová Ž, Saarenpää S, Saiz-Castro G, Cuevas M, Watters A, Lundeberg J, Bonneau R, Phatnani H. Spatiotemporal dynamics of molecular pathology in amyotrophic lateral sclerosis. *Science*. 2019 Apr 5;364(6435):89–93. PMID: 30948552
44. Saez-Atienzar S, Bandres-Ciga S, Langston RG, Kim JJ, Choi SW, Reynolds RH, International ALS Genomics Consortium, ITALSGEN, Abramzon Y, Dewan R, Ahmed S, Landers JE, Chia R, Ryten M, Cookson MR, Nalls MA, Chiò A, Traynor BJ. Genetic analysis of amyotrophic lateral sclerosis identifies contributing pathways and cell types. *Sci Adv* [Internet]. 2021 Jan;7(3). Available from: <http://dx.doi.org/10.1126/sciadv.abd9036> PMID: PMC7810371
45. Dols-Icardo O, Montal V, Sirisi S, López-Pernas G, Cervera-Carles L, Querol-Vilaseca M, Muñoz L, Belbin O, Alcolea D, Molina-Porcel L, Pegueroles J, Turón-Sans J, Blesa R, Lleó A, Fortea J, Rojas-García R, Clarimón J. Motor cortex transcriptome reveals microglial key events in amyotrophic lateral sclerosis. *Neurol Neuroimmunol Neuroinflamm* [Internet]. 2020 Sep;7(5). Available from: <http://dx.doi.org/10.1212/NXI.0000000000000829> PMID: PMC7371375
46. Hagemann C, Tyzack GE, Taha DM, Devine H, Greensmith L, Newcombe J, Patani R, Serio A, Luisier R. Automated and unbiased discrimination of ALS from control tissue at single cell resolution. *Brain Pathol*. 2021 Feb 11;e12937. PMID: 33576079
47. Jiang YM, Yamamoto M, Kobayashi Y, Yoshihara T, Liang Y, Terao S, Takeuchi H, Ishigaki S, Katsuno M, Adachi H, Niwa JI, Tanaka F, Doyu M, Yoshida M, Hashizume Y, Sobue G. Gene expression profile of spinal motor neurons in sporadic amyotrophic lateral sclerosis. *Ann Neurol*. 2005 Feb;57(2):236–251. PMID: 15668976
48. Shiga A, Ishihara T, Miyashita A, Kuwabara M, Kato T, Watanabe N, Yamahira A, Kondo C, Yokoseki A, Takahashi M, Kuwano R, Kakita A, Nishizawa M, Takahashi H, Onodera O. Alteration of POLDIP3 splicing associated with loss of function of TDP-43 in tissues affected with ALS. *PLoS One*. 2012 Aug 10;7(8):e43120. PMID: PMC3416794
49. Jangi M, Fleet C, Cullen P, Gupta SV, Mekhoubad S, Chiao E, Allaire N, Bennett CF, Rigo F, Krainer AR, Hurt JA, Carulli JP, Staropoli JF. SMN deficiency in severe models of spinal muscular atrophy causes widespread intron retention and DNA damage. *Proc Natl Acad Sci U S*

A. 2017 Mar 21;114(12):E2347–E2356. PMID: PMC5373344

50. Perego MGL, Taiana M, Bresolin N, Comi GP, Corti S. R-Loops in Motor Neuron Diseases. *Mol Neurobiol.* 2019 Apr;56(4):2579–2589. PMID: 30047099
51. Milek M, Imami K, Mukherjee N, Bortoli FD, Zinnall U, Hazapis O, Trahan C, Oeffinger M, Heyd F, Ohler U, Selbach M, Landthaler M. DDX54 regulates transcriptome dynamics during DNA damage response. *Genome Res.* 2017 Aug;27(8):1344–1359. PMID: PMC5538551
52. Tan DQ, Li Y, Yang C, Li J, Tan SH, Chin DWL, Nakamura-Ishizu A, Yang H, Suda T. PRMT5 Modulates Splicing for Genome Integrity and Preserves Proteostasis of Hematopoietic Stem Cells. *Cell Rep.* 2019 Feb 26;26(9):2316–2328.e6. PMID: 30811983
53. Boutz PL, Bhutkar A, Sharp PA. Detained introns are a novel, widespread class of post-transcriptionally spliced introns. *Genes Dev.* 2015 Jan 1;29(1):63–80. PMID: PMC4281565
54. Crossley MP, Bocek MJ, Hamperl S, Swigut T, Cimprich KA. qDRIP: a method to quantitatively assess RNA-DNA hybrid formation genome-wide. *Nucleic Acids Res.* 2020 Aug 20;48(14):e84. PMID: PMC7641308
55. Gore A, Li Z, Fung HL, Young JE, Agarwal S, Antosiewicz-Bourget J, Canto I, Giorgetti A, Israel MA, Kiskinis E, Lee JH, Loh YH, Manos PD, Montserrat N, Panopoulos AD, Ruiz S, Wilbert ML, Yu J, Kirkness EF, Izpisua Belmonte JC, Rossi DJ, Thomson JA, Eggan K, Daley GQ, Goldstein LSB, Zhang K. Somatic coding mutations in human induced pluripotent stem cells. *Nature.* 2011 Mar 3;471(7336):63–67. PMID: PMC3074107
56. Miller MB, Huang AY, Kim J, Zhou Z, Kirkham SL, Maury EA, Ziegenfuss JS, Reed HC, Neil JE, Rento L, Ryu SC, Ma CC, Luquette LJ, Ames HM, Oakley DH, Frosch MP, Hyman BT, Lodato MA, Lee EA, Walsh CA. Somatic genomic changes in single Alzheimer’s disease neurons. *Nature.* 2022 Apr;604(7907):714–722. PMID: PMC9357465
57. Risso D, Ngai J, Speed TP, Dudoit S. Normalization of RNA-seq data using factor analysis of control genes or samples. *Nat Biotechnol.* 2014 Sep;32(9):896–902. PMID: PMC4404308
58. Vogel C, Marcotte EM. Insights into the regulation of protein abundance from proteomic and transcriptomic analyses. *Nat Rev Genet.* 2012 Mar 13;13(4):227–232. PMID: PMC3654667
59. de Sousa Abreu R, Penalva LO, Marcotte EM, Vogel C. Global signatures of protein and mRNA expression levels. *Mol Biosyst.* 2009 Dec;5(12):1512–1526. PMID: PMC4089977
60. Baxi EG, Thompson T, Li J, Kaye JA, Lim RG, Wu J, Ramamoorthy D, Lima L, Vaibhav V, Matlock A, Frank A, Coyne AN, Landin B, Ornelas L, Mosmiller E, Thrower S, Farr SM, Panther L, Gomez E, Galvez E, Perez D, Meepe I, Lei S, Mandefro B, Trost H, Pinedo L, Banuelos MG, Liu C, Moran R, Garcia V, Workman M, Ho R, Wyman S, Roggenbuck J, Harms MB, Stocksdales J, Miramontes R, Wang K, Venkatraman V, Holewinski R, Sundararaman N, Pandey R, Manalo DM, Donde A, Huynh N, Adam M, Wassie BT, Vertudes E, Amirani N, Raja K, Thomas R, Hayes L, Lenail A, Cerezo A, Luppino S, Farrar A, Pothier L, Prina C, Morgan T, Jamil A, Heintzman S, Jockel-Balsarotti J, Karanja E, Markway J, McCallum M, Joslin B, Alibazoglu D, Kolb S, Ajroud-Driss S, Baloh R, Heitzman D, Miller T, Glass JD, Patel-Murray NL, Yu H, Sinani E, Vigneswaran P, Sherman AV, Ahmad O, Roy P, Beavers JC, Zeiler S, Krakauer JW, Agurto C, Cecchi G, Bellard M, Raghav Y, Sachs K, Ehrenberger T, Bruce E, Cudkowicz ME, Maragakis N, Norel R, Van Eyk JE, Finkbeiner S, Berry J, Sareen D, Thompson LM, Fraenkel E, Svendsen CN, Rothstein JD. Answer ALS, a large-scale resource for sporadic and familial ALS combining clinical and multi-omics data from induced pluripotent cell lines. *Nat Neurosci [Internet].* 2022 Feb 3; Available from: <http://dx.doi.org/10.1038/s41593-021-01006-0> PMID: 35115730

61. Smyth GK. Linear models and empirical bayes methods for assessing differential expression in microarray experiments. *Stat Appl Genet Mol Biol*. 2004 Feb 12;3:Article3. PMID: 16646809
62. Peterson KA, Nishi Y, Ma W, Vedenko A, Shokri L, Zhang X, McFarlane M, Baizabal JM, Junker JP, van Oudenaarden A, Mikkelsen T, Bernstein BE, Bailey TL, Bulyk ML, Wong WH, McMahon AP. Neural-specific Sox2 input and differential Gli-binding affinity provide context and positional information in Shh-directed neural patterning. *Genes Dev*. 2012 Dec 15;26(24):2802–2816. PMID: PMC3533082
63. Sances S, Bruijn LI, Chandran S, Eggen K, Ho R, Klim JR, Livesey MR, Lowry E, Macklis JD, Rushton D, Sadegh C, Sareen D, Wichterle H, Zhang SC, Svendsen CN. Modeling ALS with motor neurons derived from human induced pluripotent stem cells. *Nat Neurosci*. 2016 Apr;19(4):542–553. PMID: PMC5015775
64. Kapeli K, Pratt GA, Vu AQ, Hutt KR, Martinez FJ, Sundararaman B, Batra R, Freese P, Lambert NJ, Huelga SC, Chun SJ, Liang TY, Chang J, Donohue JP, Shiue L, Zhang J, Zhu H, Cambi F, Kasarskis E, Hoon S, Ares M Jr, Burge CB, Ravits J, Rigo F, Yeo GW. Distinct and shared functions of ALS-associated proteins TDP-43, FUS and TAF15 revealed by multisystem analyses. *Nat Commun*. 2016 Jul 5;7:12143. PMID: PMC4935974
65. Luisier R, Tyzack GE, Hall CE, Mitchell JS, Devine H, Taha DM, Malik B, Meyer I, Greensmith L, Newcombe J, Ule J, Luscombe NM, Patani R. Intron retention and nuclear loss of SFPQ are molecular hallmarks of ALS. *Nat Commun*. 2018 May 22;9(1):2010. PMID: PMC5964114

Reviewers' comments:

Reviewer #1 (Remarks to the Author):

In response to Reviewer comments the authors have added significant new analyses however the work is still largely descriptive which (in my opinion) makes the impact below Nature Communications.

The major issue is whether or not the expression, splicing and somatic mutations identified by the authors are causal in the pathogenesis of ALS or whether they represent downstream consequences of disease. The authors attempt to address this in 2 ways in the revised manuscript:

1. The authors claim that somatic mutations present within RNA derived from iPSN and post mortem tissue (proposed as a consequence of DNA damage) must be upstream of the transcriptome changes linked to the DNA damage response.

The issues with this are (i) that this does not address causality with respect to the pathogenesis of ALS, only with respect to the DNA damage response itself. And (ii) the new data still do not conclusively demonstrate somatic mutations linked to DNA damage: the addition of analysis of post mortem tissue is a good step but measurements of somatic heterogeneity are not consistent between iPSN and tissue, and actually SNPs reminiscent of failure to repair oxidative breaks (C>T and C>A) are reduced in the ALS postmortem tissue compared to controls which is opposite to what would be expected with excessive ALS-associated DNA damage.

2. The authors examine several TDP-43 models to try and associate changes in p53 signalling with TDP-43 mislocalisation which is an upstream event in the development of neurotoxicity. However the data does not convincingly demonstrate that the p53 signalling changes are a direct result of TDP-43 mislocalisation (versus a downstream marker of toxicity). In reality this would require a rescue experiment; however, in the analysis of sorted nuclei with and without TDP-43, it would be useful to know that p53 signalling was not perturbed in nuclei isolated from ALS patients but without TDP-43 mislocalisation.

Reviewer #2 (Remarks to the Author):

The authors have done a good job in their rebuttal in terms of the biological relevance and novelty of the work. The additional analyses, including evaluating the individual data sets as well as having the integration of data is strong. Particularly strengthening the manuscript is the inclusion of TDP43 depleted nuclei and inclusion of mass spec data. As a resource, this provides additional value to the field.

Reviewer #3 (Remarks to the Author):

Ziff et al. revised their manuscript and provided a detailed description of their method for analysis. Combining data of iPSMNs generated by different methods and at different differentiation stages using in silico analysis is a major infeasibility in actual biology, and the results of the analysis need to be validated on real samples. However, this issue has not been addressed in the revised manuscript. Therefore, it should be described as a paper on methodology and its validation in silico. Furthermore, the DNA instability of ALS claimed to be found in this study has already been reported and is not novel.

Major points:

1. The authors changed the title of the revised manuscript. However, the title should be revised to claim the analytical method as the focus of this manuscript, since it is not examined in a biological manner, especially using TDP-43 pathology.

2. The limitations of combining and analyzing data from iPSMNs samples at different differentiation stages should be described in detail.

Reviewer #4 (Remarks to the Author):

The manuscript has been dramatically improved and now the aims and results are pretty clear. My only concern is about SNV statistics. No doubt about the generalized linear model but my question is related to the relationship between the number of variants and the coverage depth that, in turn, depends on the number of generated reads per experiment. Having 10 experiments, each one with a different number of reads, I could experience a different number of variants. I expect that the number of such variants increase with the increase of the number of reads per experiment. This is the reason why I suggested revising this statistic.

Reviewer #1

Comment 1

In response to Reviewer comments the authors have added significant new analyses however the work is still largely descriptive which (in my opinion) makes the impact below Nature Communications. The major issue is whether or not the expression, splicing and somatic mutations identified by the authors are causal in the pathogenesis of ALS or whether they represent downstream consequences of disease.

- We greatly thank the reviewer for reviewing our updated manuscript and for this comment. However, we respectfully disagree that our substantial findings are below the impact of Nature Communications. The findings from TDP-43 sorted nuclei, TDP-43 knockdown, and newly added TDP-43 overexpression data enable us to establish a functional link between DNA damage and TDP-43 mislocalisation and are thus not solely descriptive. Furthermore, expression and splicing events shown to be a causal mechanism of ALS pathogenesis tend to be published in Nature, for example, with *UNC13A* splicing^{2,3} and *ATXN2* expression⁴. Conversely, Nature Neuroscience and Nature Communications have indeed published large resource-based findings, which may not fully establish causation, e.g.: Postmortem clusters and prognosis in ALS; SOD1 variants and disease duration; AnswerALS resource; NYGC ALS resource.

Author Changes

- A key strength of this study is the orthogonal examination of ALS transcriptomic changes in iPSMNs and post-mortem tissue. To further establish splicing changes in ALS, we have now examined splicing in postmortem spinal cord tissue and compared these events with ALS iPSMNs and the TDP-43 models:

We next sought to identify the splicing changes in the ALS postmortem spinal cord from the NYGC cohort (214 ALS patients and 57 controls). Because of RNA degradation in postmortem tissue, the NYGC samples were generated using ribosomal depletion instead of poly(A) library selection⁵³. Comparing splicing in post-mortem ALS versus control samples revealed 842 significant local splice events in 445 unique genes (Δ PSI > 0.1, tnom $p < 0.05$; Fig. 5f; Table S11). Amongst the differential splicing events in ALS post-mortem were 4 established ALS genes (*CAMTA1*, *NEK1*, *ATXN1*, and *GRIN1*), 19 genes with altered splicing in neuronal

nuclei depleted of TDP-43 (including *KALRN*, *PRUNE2*, *DNMI1*) and 11 genes with altered splicing in TDP-43 knockdown (e.g. *KALRN*, *DNMI1*, *NEK1*; Fig. 5i, Table S11). We also found a significant number of genes that encode DNA damage repair factors (e.g. *APTX*, *CENPX*, *RIF1*, *CNOT3*; Fisher $p = 6.7 \times 10^{-16}$) and RBPs (e.g. *EIF4E3*, *HNRNPUL1*, *ATXN1*, *SRSF5*; Fisher $p = 1.3 \times 10^{-26}$).

Functional enrichment analysis confirmed that the 445 genes with altered splicing were involved with protein binding (FDR = 2.5×10^{-10}) and neuron compartments (FDR = 1.6×10^{-6} , Fig. 5g). Of the 842 differential splicing events, 178 (21.1%) involved de novo splice junctions and of these, 21 were cryptic exons of which EP400, PLEKHA1, BMP2K, and KMT2C overlapped with TDP-43 depletion⁴². The majority of differential splice events harboured IR (462 / 842, 55%) and IR was the second most common splicing type, accounting for 27.7% of all postmortem splicing events, behind skipped exons (28.2%; Fig. 5h). Intersecting genes exhibiting altered splicing in postmortem tissue with iPSMNs revealed 12 overlapping genes including synaptojanin 1 (*SYNJ1*), kinesin 1B (*KIF1B*), dynamin 2 (*DNM2*), and polyA ribonuclease 3 (*PAN3*) as well as others involved with cytoskeletal functions (e.g. *AGAPI*, *Cytohesin 1*; Fisher exact test $p = 3.3 \times 10^{-9}$; Table S11).

Fig 5f, Differential alternative splicing in poly(A) selected ALS versus control (b) iPSMNs and (f) post-mortem tissue. Y-axis is $-\log_{10}$ of TNOM p-value with events < 0.05 coloured. X-axis is Δ PSI (ALS - CTRL) with events > 0.1 coloured red (increased) and < -0.1 blue (decreased). **c, g**, Functionally enriched terms amongst genes with differential alternative splicing in (c) iPSMNs and (g) post-mortem. **d, h**, Pie chart showing the categorisation of differential local splice variants into each of the basic splicing event types using the MAJIQ modulizer in (d) iPSMNs and (h) post-mortem. **e, i**, Violin plots showing PSI values (y-axis) for each ALS (red) and control samples (blue) for the representative splicing events in (e) iPSMNs and (i) post-mortem.

Comment 2

The authors attempt to address this in 2 ways in the revised manuscript:

The authors claim that somatic mutations present within RNA derived from iPSN and post mortem tissue (proposed as a consequence of DNA damage) must be upstream of the transcriptome changes linked to the DNA damage response. The issues with this are (i) that this does not address causality with respect to the pathogenesis of ALS, only with respect to the DNA damage response itself.

- We thank the reviewer for this comment and we acknowledge that the heightened somatic mutations, and the DNA damage response, do not equate to causality with respect to motor neuron degeneration. Nonetheless, there are novel aspects of this finding and, as noted by Reviewer 2, is important for the field as it exposes a disease signature that reveals potential DNA repair-based therapeutic targets.

Author Changes

- To make clearer that the heightened somatic mutation and subsequent DNA damage response do not necessarily indicate causality with respect to ALS motor neuron degeneration, we have now added the following discussion:

Whether somatic mutations and p53 activation in ALS causes motor neuron degeneration remains to be established. It has been shown that the acquisition of DNA damage in post-mitotic neurons promotes cell cycle re-entry, attempting to activate cell cycle-associated DNA repair pathways, but in doing so triggers an apoptotic outcome⁵. Furthermore, studies of p53 ablation and inhibition in *C9orf72* and TDP-43 mutant iPSMNs, mouse and fly models have demonstrated a phenotypic rescue, supporting a pathogenic role of the DNA damage response^{1,6,7}. Another possibility is whether DNA damage indirectly causes neuronal dysfunction, for

example, through invoking the neuroinflammatory cGAS-STING pathway^{8,9}. Thus, DNA damage may contribute directly to neuronal death, or perturb other mechanisms that maintain healthy neuronal functions.

Comment 3

And (ii) the new data still do not conclusively demonstrate somatic mutations linked to DNA damage: the addition of analysis of post mortem tissue is a good step but measurements of somatic heterogeneity are not consistent between iPSN and tissue, and actually SNPs reminiscent of failure to repair oxidative breaks (C>T and C>A) are reduced in the ALS postmortem tissue compared to controls which is opposite to what would be expected with excessive ALS-associated DNA damage.

- We thank the reviewer for this comment. Given the violin plots in Extended Data Fig 18 a-b showing the numbers of each base substitution type, we understand how the reviewer has come to this conclusion. However, these violin plots are somewhat misleading as although they show the numbers of each type of base substitution separately, they do not provide a sense of the **relative contribution** of each base substitution type. The overall differences in the numbers of each base substitution type in ALS vs CTRL are modest and showing only the total numbers without showing the relative contributions of each has the potential for misleading readers that there are large differences when the relative differences are tiny. We had originally included these relative proportion plots in the first submission but removed them due to confusion by Reviewer 2 Comment 7 in the initial rebuttal between relative contribution and the number of variants.
- Irrespective of the modest differences for SNVs, the effect sizes for the other mutation types (insertions, deletions and gene fusions) in ALS vs CTRL are consistently large for both iPSMNs and post-mortem tissue, which indeed suggests excessive ALS-associated DNA damage.

Author Changes

- As in the MutationalPatterns vignette, we have now replaced the numbers of each base substitution with the more traditional **relative contribution** of each base substitution. These 96 trinucleotide mutation profiles are utilised extensively in the cancer field for depicting single base substitution signatures e.g. <https://www.nature.com/articles/s41586-020-1943-3#Fig2>. Using this approach we confirm consistent SNV patterns between iPSMNs and post-mortem tissue contexts:

Assessing the relative contributions of each base substitution type revealed largely similar SNV spectrum profiles in ALS and CTRL iPSMNs, predominantly composed of C>T and T>C substitutions (Extended Data Fig. 18a).

As with iPSMNs, base substitutions in post-mortem tissue were largely composed of C>T and T>C substitutions with similar SNV spectrum profiles between ALS and CTRL post-mortem tissue (Extended Data Fig. 18b).

Extended Data Fig 18a-b, Relative contribution of each base substitution type (96 trinucleotide mutation profile) in CTRL (top) and ALS (bottom) (a) iPSMNs and (b) postmortem tissue.

Comment 4

The authors examine several TDP-43 models to try and associate changes in p53 signalling with TDP-43 mislocalisation which is an upstream event in the development of neurotoxicity. However the data does not convincingly demonstrate that the p53 signalling changes are a direct result of TDP-43 mislocalisation (versus a downstream marker of toxicity). In reality this would require a rescue experiment; however, in the analysis of sorted nuclei with and without TDP-43, it would be useful to know that p53 signalling was not perturbed in nuclei isolated from ALS patients but without TDP-43 mislocalisation.

- We thank the reviewer for these points. The suggested rescue experiment examining restoration of TDP-43 expression on the DNA damage response has already been performed - please see Vogt et al 2018⁷; Lee et al 2012¹⁰; Mitra et al 2019¹¹; Maor-Nof et al 2021¹. We have outlined these previous results in paragraph 3 of the discussion:

Although we found p53 activation in ALS cases that lack TDP-43 pathology (FUS and SOD1 mutants), the magnitude of activation was substantially weaker than in

TDP-43 ALS cases, raising the possibility that TDP-43 contributes to the DNA damage response. In support of this, we found significant p53 upregulation in neuronal nuclei depleted of TDP-43, as well as TDP-43 knockdown **and overexpression** models. **This is consistent with the established role of TDP-43 in DNA repair and p53 activation** ^{1,7,10-12}. TDP-43 depletion results in the accumulation of DNA damage whilst TDP-43 overexpression **leads to cell cycle arrest at the G2/M phase and a** pro-apoptotic phenotype, which can be partially rescued by p53 inhibition ^{1,7,10-12}. Together, this suggests that TDP-43 pathology exacerbates the DNA damage response, and the subsequent p53 activation may promote motor neuron death in ALS.

- We are unable to compare nuclei isolated from ALS patients without TDP-43 depletion with non-ALS samples, as there are no non-ALS samples in the Liu et al 2019¹³ dataset. However, by comparing the non-TDP43 ALS with non-ALS controls in both iPSMN and postmortem tissue, we have already confirmed that p53 is weakly activated in non-TDP43 ALS cases. Therefore, TDP-43 mislocalisation is unlikely to be responsible for the entirety of the DNA damage signature in ALS.

Author Changes

- To further establish a functional link between TDP-43 and the DNA damage response, in addition to the TDP-43 depletion data, we have now examined mouse neurons **overexpressing TDP-43**:

We next examined RNA-seq from mouse primary neurons overexpressing TDP-43 ¹. Compared to controls, neurons overexpressing TDP-43 showed significant upregulation of the p53 pathway (NES +4.1, p = 0.002) and TP53 transcription factor (NES +4.1, p = 0.002 Extended Data Fig. 15g,h). Together, these results indicate that both TDP-43 depletion and accumulation augment p53 activity, suggesting that tight regulation of TDP-43 levels is required to ensure an appropriate DNA damage response.

Ext Data Fig 15g,h: PROGENy signalling pathway bar charts (left) and DoRothEA transcription factor activities volcano plot (right) in TDP-43 overexpressing mouse neurons.

Methods

Primary mouse neurons overexpressing TDP-43 were utilised from GSE162048. Neurons transduced with lentivirus overexpressing TDP-43 for 20 hours were compared to control neurons.

Reviewer #2

Comment 1

The authors have done a good job in their rebuttal in terms of the biological relevance and novelty of the work. The additional analyses, including evaluating the individual data sets as well as having the integration of data is strong. Particularly strengthening the manuscript is the inclusion of TDP43 depleted nuclei and inclusion of mass spec data. As a resource, this provides additional value to the field.

- We greatly thank the reviewer for reviewing the revised manuscript and for these positive comments.

Reviewer #3

Comment 1

Ziff et al. revised their manuscript and provided a detailed description of their method for analysis. Combining data of iPSMNs generated by different methods and at different differentiation stages using in silico analysis is a major infeasibility in actual biology, and the results of the analysis need to be validated on real samples. However, this issue has not been addressed in the revised manuscript. Therefore, it should be described as a paper on methodology and its validation in silico.

- We greatly thank the reviewer for reviewing the updated manuscript. As outlined in our previous rebuttal, and acknowledged by Reviewer 2, we have addressed the concern of dataset integration by examining each dataset individually. We further addressed this concern with extensive QC checks, detailed statistical design in the generalised linear model that comprehensively accounts for confounding variables, and numerous subgroup analyses.
- Whilst there are differences in iPSMN differentiation protocols, we disagree with the suggestion that iPSMNs are of “different differentiation stages”. By examining markers of motor neuron differentiation state (pluripotency; neural precursors; progenitors; and post-mitotic) we showed in the previous rebuttal the homogenous expression between datasets (Extended Data Fig 5).
- The comment that “the results of the analysis need to be validated on real samples” implies that the iPSMNs and post-mortem tissue used in this study are not real. This is not true - 429 iPSMN samples and 271 post-mortem samples were used in this study. We would respectfully argue that *there are no models that faithfully recapitulate ALS beyond the actual patient tissue and enriched cultures of their own motor neurons*. Nevertheless, we have additionally examined 49 TDP-43 knockdown samples and 30 FACS-sorted postmortem nuclei, which further validate our core findings.

Comment 2

Furthermore, the DNA instability of ALS claimed to be found in this study has already been reported and is not novel.

- We thank the reviewer for this comment. We explained the novelty of this study in our previous rebuttal and it is unclear as to what aspects of our novelty explanation the reviewer disagrees with. We repeat our explanation here:
- Whilst we agree that p53 activation has been previously described in ALS, these have been **limited to mutant forms of ALS**, predominantly with C9orf72 mutations^{1,14-17}. Previous reports have also been **underpowered**, for example, ALS iPSMN studies examining DNA damage have been restricted to very small sample sizes and mutant forms of ALS, e.g. Lopez-Gonzalez et al 2016 studied only 3 C9orf72 mutant iPSMNs¹⁵. Likewise, post-mortem studies reporting p53 upregulation have been limited to only a handful of patients, e.g. Farg et al 2016 in 10 C9orf72 patients¹⁴.
- Conversely, this is the first study with sufficient power to demonstrate genome instability across ALS subtypes in both iPSMNs and post-mortem. This is especially noteworthy in sporadic ALS, which accounts for 90% of cases but has the least prior evidence for increased p53^{18,19}. The scale of this resource also enables us to compare the magnitude of p53 activation between ALS subgroups for the first time, exposing that TDP-43 pathology cases show substantially greater p53 activation than non-TDP-43 ALS cases. Furthermore, this is the first study to show increased somatic mutations and gene fusions in both ALS iPSMNs and post-mortem tissue.
- The novelty of this resource is that it provides the largest catalogue of human ALS-associated gene alterations to date, obtained by characterising 429 human ALS iPSMN and 271 post-

mortem transcriptomes. The unbiased genome-wide approach used reveals multiple novel targets in ALS beyond p53 e.g. RNase L expression, MAPK signalling, CNOT3 splicing. It also provides the largest study of splicing changes in ALS iPSMNs and post-mortem tissue, not only revealing TDP-43 regulated events (e.g. POLDIP3, CAMK2B, CEP290) but also revealing new splicing events shared between ALS subgroups (e.g. CNOT3, SRSF10, DNAJC17). Thus, this paper offers substantial advances in the understanding of changes across ALS subtypes, which is crucial for identifying therapeutic targets.

Author Changes

- We discuss other ALS studies of DNA damage, and clarify the novelty regarding DNA damage in this study:

Our findings support previous, smaller studies showing p53 activation in ALS, particularly with *C9orf72* repeat expansions^{1,11,14-17,19-21}. However, the novelty in this study is the finding of p53 activation, together with increased somatic mutations and gene fusions, across diverse ALS subgroups, not only in iPSMNs but also in post-mortem spinal cord tissue. We identified the greatest increase in p53 activity in *C9orf72* mutants, consistent with reports that the *C9orf72* repeat expansion induces DNA damage, likely mediated by dipeptide repeat proteins and the formation of R loops and G quadruplexes^{1,14-17}. However, we also found p53 to be strongly and significantly activated in *TARDBP* and sporadic subgroups. This finding in sporadic cases is particularly striking since they represent 90% of ALS but have the least prior evidence for p53 activation.

- In this rebuttal, we have further added splicing in ALS post-mortem tissue exposing further novel splicing changes (e.g. CAMTA1, NEK1, ATXN1) - please see Reviewer 1, Comment 1.

Comment 3

The authors changed the title of the revised manuscript. However, the title should be revised to claim the analytical method as the focus of this manuscript, since it is not examined in a biological manner, especially using TDP-43 pathology.

- We thank the reviewer for this comment. We respectfully disagree that the manuscript has not been examined in a biological manner - we have used well-established accepted gold-standard reproducible bioinformatic approaches to integrate and examine gene expression, alternative splicing, and somatic mutations of 429 iPSMN and 271 post-mortem samples.
- Our findings are based on iPSMNs and post-mortem samples from NYGC and as advised by Reviewer 3, Comment 6 in the previous review, we subgrouped samples based on their TDP-43 pathological status. Furthermore, we validate our findings utilising FACS-sorted post-mortem nuclei (TDP-43 positive vs TDP-43 negative); TDP-43 knockdown samples; as well

as TDP-43 overexpression samples. We are therefore of the opinion that the use of TDP-43 pathology in the title is justified.

Comment 4

The limitations of combining and analyzing data from iPSMN samples at different differentiation stages should be described in detail.

- We thank the reviewer for this comment. As noted in our response to Reviewer 3, Comment 1, examination of differentiation markers confirmed a homogeneous differentiation state between the included iPSMN datasets, indicating that the iPSMNs are not of different differentiation states. Furthermore, differences due to variation in iPSMN differentiation protocols were removed by including *dataset* as a covariate in the DESeq2 design.

Author Changes

- We now further acknowledge the limitations of dataset integration in the discussion:

Although we safeguard against dataset batch effects by using a generalised linear model that comprehensively adjusts for known confounding variables, an inherent limitation of integrative studies is that it is possible that confounders remain unknown or masked. We minimised these risks by utilising extensive quality control and validation subgroup analyses.

Reviewer #4

Comment 1

The manuscript has been dramatically improved and now the aims and results are pretty clear.

My only concern is about SNV statistics. No doubt about the generalized linear model but my question is related to the relationship between the number of variants and the coverage depth that, in turn, depends on the number of generated reads per experiment. Having 10 experiments, each one with a different number of reads, I could experience a different number of variants. I expect that the number of such variants increase with the increase of the number of reads per experiment. This is the reason why I suggested revising this statistic.

- We greatly thank the reviewer for reviewing our updated manuscript and for the positive comments. Regarding the SNV statistics, we fully agree with the reviewer that variation in sample-to-sample coverage depth confounds the number of variants detected. In this plot you can see how increasing coverage depth is associated with an increasing raw number of variants detected in the AnswerALS iPSMN dataset:

However, by including coverage as a variable in the statistical design of the generalised linear model, and by having a well-powered sample of 300 samples, we are able to examine the number of variants **adjusted for coverage** differences:

By examining the partial residuals from the GLM, we remove the unwanted variation due to coverage (i.e. we get the number of variants detected adjusted for sample-to-sample variation in coverage, rather than the **raw** number of variants). We are then able to examine the **coverage-adjusted** number of variant differences due to ALS versus CTRL:

This same issue applies equally to age, whereby increasing age is associated with increasing numbers of variants detected. Therefore, we have also adjusted for age in the model and so the numbers of variants examined in the manuscript are age and coverage adjusted.

There is an eloquent explanation and example of how the GLM adjusts for covariates at: https://www.middleprofessor.com/files/applied-biostatistics_bookdown/book/adding-covariates-to-a-linear-model.html#understanding-a-linear-model-with-an-added-covariate-heart-necrosis-data

Furthermore, the reliability of RNAseq variant detection and how coverage differences influence variant calling are examined using subsampling at:

<https://www.ncbi.nlm.nih.gov/pmc/articles/PMC3791257/> (see Fig 5).

References

1. Maor-Nof M, Shipony Z, Lopez-Gonzalez R, Nakayama L, Zhang YJ, Couthouis J, Blum JA, Castruita PA, Linares GR, Ruan K, Ramaswami G, Simon DJ, Nof A, Santana M, Han K, Sinnott-Armstrong N, Bassik MC, Geschwind DH, Tessier-Lavigne M, Attardi LD, Lloyd TE, Ichida JK, Gao FB, Greenleaf WJ, Yokoyama JS, Petrucelli L, Gitler AD. p53 is a central regulator driving neurodegeneration caused by C9orf72 poly(PR). *Cell*. 2021 Feb 4;184(3):689–708.e20. PMID: PMC7886018
2. Brown AL, Wilkins OG, Keuss MJ, Hill SE, Zanovello M, Lee WC, Bampton A, Lee FCY, Masino L, Qi YA, Bryce-Smith S, Gatt A, Hallegger M, Fagegaltier D, Phatnani H, NYGC ALS Consortium, Newcombe J, Gustavsson EK, Seddighi S, Reyes JF, Coon SL, Ramos D, Schiavo G, Fisher EMC, Raj T, Secrier M, Lashley T, Ule J, Buratti E, Humphrey J, Ward ME, Fratta P. TDP-43 loss and ALS-risk SNPs drive mis-splicing and depletion of UNC13A. *Nature* [Internet]. 2022 Feb 23; Available from: <http://dx.doi.org/10.1038/s41586-022-04436-3> PMID: 35197628
3. Ma XR, Prudencio M, Koike Y, Vatsavayai SC, Kim G, Harbinski F, Briner A, Rodriguez CM, Guo C, Akiyama T, Schmidt HB, Cummings BB, Wyatt DW, Kurylo K, Miller G, Mekhoubad S, Sallee N, Mekonnen G, Ganser L, Rubien JD, Jansen-West K, Cook CN, Pickles S, Oskarsson B, Graff-Radford NR, Boeve BF, Knopman DS, Petersen RC, Dickson DW, Shorter J, Myong S, Green EM, Seeley WW, Petrucelli L, Gitler AD. TDP-43 represses cryptic exon inclusion in the FTD-ALS gene UNC13A. *Nature* [Internet]. 2022 Feb 23; Available from: <http://dx.doi.org/10.1038/s41586-022-04424-7> PMID: 35197626
4. Becker LA, Huang B, Bieri G, Ma R, Knowles DA, Jafar-Nejad P, Messing J, Kim HJ, Soriano

- A, Auburger G, Pulst SM, Taylor JP, Rigo F, Gitler AD. Therapeutic reduction of ataxin-2 extends lifespan and reduces pathology in TDP-43 mice. *Nature*. 2017 Apr 20;544(7650):367–371. PMID: PMC5642042
5. Folch J, Junyent F, Verdaguer E, Auladell C, Pizarro JG, Beas-Zarate C, Pallàs M, Camins A. Role of cell cycle re-entry in neurons: a common apoptotic mechanism of neuronal cell death. *Neurotox Res*. 2012 Oct;22(3):195–207. PMID: 21965004
 6. Mc Guire C, Beyaert R, van Loo G. Death receptor signalling in central nervous system inflammation and demyelination. *Trends Neurosci*. 2011 Dec;34(12):619–628. PMID: 21999927
 7. Vogt MA, Ehsaei Z, Knuckles P, Higginbottom A, Helmbrecht MS, Kunath T, Eggan K, Williams LA, Shaw PJ, Wurst W, Floss T, Huber AB, Taylor V. TDP-43 induces p53-mediated cell death of cortical progenitors and immature neurons. *Sci Rep*. Nature Publishing Group; 2018 May 25;8(1):1–13.
 8. Yu CH, Davidson S, Harapas CR, Hilton JB, Mlodzianoski MJ, Laohamonthonkul P, Louis C, Low RRJ, Moecking J, De Nardo D, Balka KR, Calleja DJ, Moghaddas F, Ni E, McLean CA, Samson AL, Tyebji S, Tonkin CJ, Bye CR, Turner BJ, Pepin G, Gantier MP, Rogers KL, McArthur K, Crouch PJ, Masters SL. TDP-43 Triggers Mitochondrial DNA Release via mPTP to Activate cGAS/STING in ALS. *Cell* [Internet]. 2020 Oct 3; Available from: <http://dx.doi.org/10.1016/j.cell.2020.09.020> PMID: 33031745
 9. Wang H, Kodavati M, Britz GW, Hegde ML. DNA Damage and Repair Deficiency in ALS/FTD-Associated Neurodegeneration: From Molecular Mechanisms to Therapeutic Implication. *Front Mol Neurosci*. 2021 Dec 16;14:784361. PMID: PMC8716463
 10. Lee K, Suzuki H, Aiso S, Matsuoka M. Overexpression of TDP-43 causes partially p53-dependent G2/M arrest and p53-independent cell death in HeLa cells. *Neurosci Lett*. 2012 Jan 11;506(2):271–276. PMID: 22133803
 11. Mitra J, Guerrero EN, Hegde PM, Liachko NF, Wang H, Vasquez V, Gao J, Pandey A, Taylor JP, Kraemer BC, Wu P, Boldogh I, Garruto RM, Mitra S, Rao KS, Hegde ML. Motor neuron disease-associated loss of nuclear TDP-43 is linked to DNA double-strand break repair defects. *Proc Natl Acad Sci U S A*. 2019 Mar 5;116(10):4696–4705. PMID: PMC6410842
 12. Hill SJ, Mordes DA, Cameron LA, Neuberg DS, Landini S, Eggan K, Livingston DM. Two familial ALS proteins function in prevention/repair of transcription-associated DNA damage. *Proc Natl Acad Sci U S A*. 2016 Nov 29;113(48):E7701–E7709. PMID: PMC5137757
 13. Liu EY, Russ J, Cali CP, Phan JM, Amlie-Wolf A, Lee EB. Loss of Nuclear TDP-43 Is Associated with Decondensation of LINE Retrotransposons. *Cell Rep*. 2019 Apr 30;27(5):1409–1421.e6. PMID: PMC6508629
 14. Farg MA, Konopka A, Soo KY, Ito D, Atkin JD. The DNA damage response (DDR) is induced by the C9orf72 repeat expansion in amyotrophic lateral sclerosis. *Hum Mol Genet*. 2017 Aug 1;26(15):2882–2896. PMID: 28481984
 15. Lopez-Gonzalez R, Lu Y, Gendron TF, Karydas A, Tran H, Yang D, Petrucelli L, Miller BL, Almeida S, Gao FB. Poly(GR) in C9ORF72 -Related ALS/FTD Compromises Mitochondrial Function and Increases Oxidative Stress and DNA Damage in iPSC-Derived Motor Neurons [Internet]. *Neuron*. 2016. p. 383–391. Available from: <http://dx.doi.org/10.1016/j.neuron.2016.09.015>
 16. Walker C, Herranz-Martin S, Karyka E, Liao C, Lewis K, Elsayed W, Lukashchuk V, Chiang SC, Ray S, Mulcahy PJ, Jurga M, Tsagakis I, Iannitti T, Chandran J, Coldicott I, De Vos KJ,

Hassan MK, Higginbottom A, Shaw PJ, Hautbergue GM, Azzouz M, El-Khamisy SF. C9orf72 expansion disrupts ATM-mediated chromosomal break repair. *Nat Neurosci.* 2017 Sep;20(9):1225–1235. PMID: 28714434

17. Nihei Y, Mori K, Werner G, Arzberger T, Zhou Q, Khosravi B, Japok J, Hermann A, Sommacal A, Weber M, Kamp F, Nuscher B, Edbauer D, Haass C, German Consortium for Frontotemporal Lobar Degeneration, Bavarian Brain Banking Alliance. Poly-glycine–alanine exacerbates C9orf72 repeat expansion-mediated DNA damage via sequestration of phosphorylated ATM and loss of nuclear hnRNP A3. *Acta Neuropathol.* 2020 Jan 1;139(1):99–118.
18. Kok JR, Palminha NM, Dos Santos Souza C, El-Khamisy SF, Ferraiuolo L. DNA damage as a mechanism of neurodegeneration in ALS and a contributor to astrocyte toxicity. *Cell Mol Life Sci.* 2021 Aug;78(15):5707–5729. PMID: 34116199
19. Sun Y, Curle AJ, Haider AM, Balmus G. The role of DNA damage response in amyotrophic lateral sclerosis. *Essays Biochem.* 2020 Oct 26;64(5):847–861. PMID: 33088667
20. Martin LJ. p53 is abnormally elevated and active in the CNS of patients with amyotrophic lateral sclerosis. *Neurobiol Dis.* 2000 Dec;7(6 Pt B):613–622. PMID: 11114260
21. Ranganathan S, Bowser R. p53 and Cell Cycle Proteins Participate in Spinal Motor Neuron Cell Death in ALS. *Open Pathol J.* 2010 Jan 1;4:11–22. PMID: 20392395

REVIEWERS' COMMENTS

Reviewer #1 (Remarks to the Author):

I am happy for publication at this stage.

Reviewer #4 (Remarks to the Author):

The manuscript has been improved and the authors took into account my comments.